# BPOP-v1 model: exploring the impact of changes in the biological pump on the shelf sea and ocean nutrient and redox state

Elisa Lovecchio[1] and Timothy M. Lenton[1]

[1]Global Systems Institute, University of Exeter, Exeter, EX4 4QE, United Kingdom

*Correspondence to*: Elisa Lovecchio (e.lovecchio@exeter.ac.uk)

**Abstract.** The ocean's biological pump has changed over Earth history from one dominated by prokaryotes, to one involving a mixture of prokaryotes and eukaryotes with trophic structure. Changes in the biological pump are in turn hypothesised to have caused important changes in the ocean's nutrient and redox properties. To explore

these hypotheses, we present here a new box model including oxygen (O), phosphorus (P) and a dynamical biological pump. Our Biological Pump, Oxygen and Phosphorus (BPOP) model accounts for two – small and large – organic matter species generated by production and coagulation, respectively. Export and burial of these particles are regulated by a remineralization length ($z_{rem}$) scheme. We independently vary $z_{rem}$ of small and large particles in order to study how changes in sinking speeds and remineralization rates affect the major

biogeochemical fluxes, and O and P ocean concentrations. Modelled O and P budgets and fluxes lie reasonably close to present estimates for $z_{rem}$ in the range of currently measured values. Our results highlight that relatively small changes in $z_{rem}$ of the large particles can have important impacts on the O and P ocean availability and support the idea that an early ocean dominated by small particles was nutrient rich due to inefficient removal to sediments. The results also suggest that extremely low oxygen concentrations in the shelf can coexist with an

oxygenated deep open ocean for realistic values of $z_{rem}$, especially for large values of the small particle $z_{rem}$. This could challenge conventional interpretations that the Proterozoic deep ocean was anoxic, which are derived from shelf and slope sediment redox data. This simple and computationally inexpensive model is a promising tool to investigate the impact of changes in the organic matter sinking and remineralization rates as well as changes in physical processes coupled to the biological pump in a variety of case studies.

## 1 Introduction

The 'biological pump' describes the production of organic matter at the ocean's surface (an oxygen source), its downward export/sinking flux, remineralisation at depth (an oxygen sink), and burial. This set of processes acts against the homogenization of tracer concentrations by the ocean's circulation, maintaining large-scale tracer gradients (Sarmiento and Gruber, 2006). In today's world, the biological pump plays a key role in transferring

carbon from the atmosphere/surface ocean to the deep ocean and in so doing lowers atmospheric $CO_2$ and creates oxygen demand in deeper waters (Lam et al., 2011;Kwon et al., 2009). Those deeper waters with the greatest oxygen demand relative to oxygen supply can be driven hypoxic ($O_2 < 60$ mmol m$^{-3}$), suboxic ($O_2 < 5$ mmol m$^{-3}$) or even anoxic – as is being seen in parts of the ocean today (Keeling et al., 2010). By combining surface oxygen production and organic carbon burial, the biological pump plays a role in determining the long-term source of

oxygen to the atmosphere. The biological pump also provides a means of efficiently transferring organic matter

and the nutrients it contains to marine sediments, if sinking through the water column happens fast enough compared to remineralization for the material to hit the bottom (Sarmiento and Gruber, 2006). Hence the biological pump plays a key part in balancing the input of phosphorus to the ocean with a corresponding output flux of phosphorus buried in marine sediments.

Through Earth's history, the characteristics, efficiency and impact of the biological pump are thought to have changed dramatically due to the evolution of increasingly large and complex marine organisms (Ridgwell, 2011;Logan et al., 1995;Boyle et al., 2018). Life in the ocean began as just prokaryotes, presumably attacked by viruses, with slow sinking of the resulting tiny particles. Now the marine ecosystem is a mix of prokaryotic cyanobacteria and heterotrophs, and size-structured eukaryotic algae, mixotrophs and heterotrophs all the way up

to large jellyfish, fish and whales. Some of the resulting particles sink very fast (McDonnell and Buesseler, 2010). How changes in the biological pump have affected ocean nutrient and redox state at different times in Earth history is a subject of active research and hypothesis generation. Previous work has highlighted the Neoproterozoic Era, spanning from 1,000 to 541 million years ago, as of particular interest because it saw a shift of dominance from prokaryotes to eukaryotes and a series of dramatic shifts in the climate, biogeochemical cycling and ocean redox

state (Katz et al., 2007;Brocks et al., 2017). A common paradigm has been to assume that a progressive rise of oxygen in the atmosphere (of uncertain cause) drove the oxygenation of the deep ocean at this time through air-sea gas exchange and mixing, but equally increases in the efficiency of the biological pump could have lowered ocean phosphorus concentration and thus oxygenated the ocean (Lenton et al., 2014). Recent data show a series of transient ocean oxygenation events $\sim$660-520 Ma, which get more frequent over time, suggesting a complex

interplay of processes on multiple timescales, including changes in the biological pump and ocean phosphorus inventory (Lenton and Daines, 2018).

During the Phanerozoic Eon there have been further changes to the biological pump. In particular, a rise of eukaryotic algae from the early Jurassic onwards is hypothesised to have increased the efficiency of the biological pump and thus oxygenated shallow waters (Lu et al., 2018), but presumably deoxygenated deeper waters, at least

25 in the short term. In the oceanic anoxic events (OAEs) that occurred during the Mesozoic Era there were major increases in prokaryotic nitrogen fixation yet evidence for a eukaryote-dominated biological pump (Higgins et al., 2012), raising interesting questions as to whether this reinforced anoxia at depth.

Previous modelling work has examined the impact of changes in the organic matter remineralisation length/depth ($z_{rem}$) in the 3D GENIE intermediate complexity model (Meyer et al., 2016;Lu et al., 2018). Both studies clearly

demonstrated the important control of the $z_{rem}$ on ocean oxygen concentrations – as it gets larger the oxygen minimum zone shifts to greater depths. Furthermore, Lu et al. (2018) showed that an increase in $z_{rem}$ can explain an observed deepening of the oxycline from the Paleozoic to Meso-Cenozoic in the ocean redox proxy I/Ca. However, coarse 3D models such as GENIE do not really resolve shelf seas and their dynamics, which are distinct from those of the open ocean. Furthermore, GENIE only accounts for one organic carbon species, overlooking

processes of transformation of organic material, such as coagulation and fragmentation, which contribute to modulate the efficiency of the organic matter vertical export and burial (Wilson et al., 2008;Karakaş et al., 2009;Boyd and Trull, 2007).

In this study, we take a more idealised approach, exploring how changes in the properties of the biological pump may have affected the shelf sea and open ocean nutrient and redox state using a new Biological Pump, Oxygen

and Phosphorus (BPOP) box model. This model combines a box representation of the marine O and P cycles with

an intermediate complexity representation of the biological pump transformations, including two classes of particulate organic matter (POM). BPOP allows us to modify the properties of two POM pools, whose abundance is regulated by the processes of production and coagulation. We focus on changes in the characteristic depths at which the two POM pools are remineralized, i.e., the particle remineralization length scale ($z_{rem}$), and study the resulting equilibrium budgets and fluxes. The model has a deliberately simplified treatment of redox carriers and is designed to focus on ocean P and ocean redox steady states, not on longer-term controls on atmospheric oxygen. In the following sections we describe the model, we provide an evaluation of its performance in the context of modern observations and flux estimates, and finally present and discuss our model results.

## 2 Model description

Here we describe the Biological Pump, Oxygen and Phosphorus (BPOP) model. The model was implemented using Matlab and the equations are solved by the built-in ode15s solver. BPOP can easily run on a single core, integrating 50 million years of time in less than a minute on an ordinary machine, and is therefore computationally efficient. We refer to the user's manual (see the supplementary material) for further information on how to run the model.

### 2.1 Variables and circulation

The box model resolves explicitly for each relevant box the local concentrations of three types of tracers: molecular oxygen $O_2$ (O), inorganic dissolved phosphorus (P) and sediment organic phosphorus (SedP$_{org}$). The total budgets of P and O, respectively P$^{TOT}$ and O$^{TOT}$, are also independently integrated from the net sources and sinks of the two tracers over the entire model domain, for the purpose of checking mass conservation. The entire set of the model's state and diagnostic variables and their units are listed in Table 1. In the following subsections we describe the box model's geometry and discuss the physical and geochemical fluxes that drive the tracers' dynamics. Box properties are listed in Table 2, while the set of parameters adopted for the modelled physical and geochemical fluxes can be found in Table 3.

### 2.1.1 Box properties and physical fluxes of inorganic tracers

The box model includes 4 ocean boxes, 1 atmospheric box and 2 sediment boxes (Figure 1a). The ocean and sediment boxes are equally split between shelf sea and open ocean, both including one surface ocean box and one deep ocean box.

O and P are exchanged between the 4 ocean boxes through advection and mixing, including an upwelling recirculation between shelf sea and open ocean (Wollast, 1998). For a generic tracer concentration C and in the $i^{th}$ box, the physical exchange flux [mmol m$^{-3}$ yr$^{-1}$] is represented by

$$\boldsymbol{AdvMix(C)^i = \sum_j MassFlux_{ij}/V^i \cdot (C_j - C_i)} \tag{1}$$

where MassFlux$_{ij}$ represents the volumetric flow between the $i^{th}$ box and any adjacent box j, while $V^i$ is the volume of the $i^{th}$ box.

For each surface box $i$, air-sea gas exchange allows O fluxes between the ocean and the atmosphere (at). The flux [mmolO$_2$ m$^{-3}$ yr$^{-1}$] is positive when directed into the ocean and depends on the gas transfer velocity K$_w$, atmospheric pressure p$_{at}$ (here assumed constant), and Henry's constant K$_{Henry}$, as in:

$$AirSea^i = K_W \cdot (O^{at} \cdot p_{at}/K_{Henry} - O^i) \cdot A^i/V^i \qquad (2)$$

where $K_W$ [m yr$^{-1}$] is a function of the prescribed mean temperature $T_{mean}$ and wind speed $W_{speed}$ (Sarmiento and Gruber, 2006).

### 2.1.2 Initialization and boundary fluxes

The model is initialized with an even concentration of P ($P_{ini}$) in all the ocean boxes, zero oxygen and zero sediment $P_{org}$. A constant input of P from rivers ($P_{in}$) into the surface ocean replenishes the P ocean reservoir despite the burial flux (net sink of $P_{org}$) into the sediments. $P_{in}$ is in part delivered directly to the surface open ocean (Sharples et al., 2017). At equilibrium, the $P_{org}$ burial flux balances $P_{in}$. Oxidative weathering determined by atmospheric oxygen $O^{at}$ constitutes a net sink flux for O. The weathering flux [yr$^{-1}$] depends on a constant a baseline flux $W_0$ and it scales like the square root of the oxygen mixing ratio normalised to present values $Omix_0$ (Lenton et al., 2018), following:

$$OxyWeath = W_0 \cdot \sqrt{O^{at}/Omix_0} \qquad (3)$$

### 2.2 Biological pump details

The modelled tracer cycles are coupled by a set of biological transformations, i.e., the biological pump, governing the cycle of production, remineralization and burial of $P_{org}$ in the water column and in the sediments. $P_{org}$ in the water column is resolved implicitly: at each time step all the produced $P_{org}$ that does not reach the sediments is instantaneously remineralized. In this sense, in our model no $P_{org}$ can accumulate in the ocean's water column and we only calculate fluxes of watercolumn $P_{org}$ without treating $SP_{org}$ and $LP_{org}$ as state variables. This scheme is similar to the one used to represent detrital POM in some modern ocean biogeochemical models (Moore et al., 2004). P and O biological fluxes are coupled by a fixed Redfield ratio $OP_{Red}$. The next few paragraphs describe the cycle of production, coagulation, export, remineralization and burial that constitute the biological pump representation. The full set of parameters used to resolve the $P_{org}$ cycle is provided in Table 4.

### 2.2.1 Particle classes, production and coagulation

The model includes two $P_{org}$ classes, which get produced, exported and remineralized in the ocean's water column: small $P_{org}$ ($SP_{org}$) and large $P_{org}$ ($LP_{org}$). The use of two Porg classes is in line with modern ocean in situ observations, which reveal a bimodal distribution of the particle sizes and sinking speeds (Riley et al., 2012;Alonso-González et al., 2010). Moreover, it allows to better reproduce the commonly observed Martin power-law decay of the particle export flux with the use of a remineralization length scheme of export and burial fluxes (Boyd and Trull, 2007).

Organic matter production happens only in the surface ocean boxes through the uptake of P. This is regulated by a maximum rate $P_{eff}$ and a Michaelis-Menten kinetics with constant $K_P$. Production [mmolP m$^{-3}$ yr$^{-1}$] in each $i$th box only generates $SP_{org}$, according to:

$$Prod^i = P_{eff} \cdot (P^i/(P^i + K_P)) \cdot P^i \qquad (4)$$

$LP_{org}$ is generated via the coagulation of $SP_{org}$ at the surface after production. As we do not explicitly solve for the concentrations of $SP_{org}$ and $LP_{org}$, we assume that the coagulation [mmolP m$^{-3}$ yr$^{-1}$] of $SP_{org}$ into $LP_{org}$ in each box $i$ is proportional to the rate of production of small particles:

$$Coag^i = cg_f \cdot Prod^i \tag{5}$$

This is a necessary simplifying assumption compared to the usual coagulation models which define the flux as the square of the particle concentration (Boyd and Trull, 2007;Gruber et al., 2006), given the fact that our model does not resolve this variable. Coagulation impacts the relative contribution of small and large particles to the export and burial fluxes by subtracting from the local $SP_{org}$ pool and adding to the $LP_{org}$ pool.

### 2.2.2 Physical fluxes of organic material

The implicit representation of the organic matter in the water column implies that no organic matter is accumulated in the ocean. In our baseline version of the model, corresponding to the results presented in this manuscript, $SP_{org}$ and $LP_{org}$ are redistributed throughout the watercolumn exclusively by implicitly modelled gravitational sinking before being either buried, accumulated in the sediments or remineralized. Even though the vertical export by downwelling and mixing (Stukel and Ducklow, 2017), and the lateral organic matter redistribution (Lovecchio et al., 2017;Inthorn et al., 2006) may be important when working with suspended $SP_{org}$ ($z_{rem}^S = 0$), these fluxes are not currently accounted for in the model.

### 2.2.3 Remineralization length scheme

The export and sedimentation fluxes of $P_{org}$ through the water column are represented by a remineralization length scheme. In this representation, the vertical fluxes of organic matter f(z) vary exponentially with depth. The shape of the exponential depends on the value of the remineralization length ($z_{rem}$) of each organic matter species:

$$f^k(z) = f_0^k \cdot e^{-\frac{z-z_0}{z_{rem}^k}}, \tag{6}$$

where $f_0^k$ is the flux [mmolP m$^{-2}$ yr$^{-1}$] at the reference depth $z_0$, and the index $k$ indicates the organic matter pool of reference, either small (S) or large (L). This representation of the export flux is convenient, as it does not depend on the specific choice of $z_0$ (Boyd and Trull, 2007).

The remineralization length $z_{rem}$ indicates the distance through which the particle flux becomes 1/e times (about 36 %) the flux at the reference depth (Buesseler and Boyd, 2009;Marsay et al., 2015). This quantity is expressed in metres and can be calculated as the ratio between the particle sinking speed and the particle's remineralization rate (Cavan et al., 2017). Consequently, $z_{rem}$ implicitly contains information on several particle inherent properties (among which density, size, shape, organic matter liability) as well as information about the surrounding environment, e.g., the type of heterotrophs which feed upon the organic material (McDonnell and Buesseler, 2010;Baker et al., 2017). For simplicity, we assume that the remineralization length of small and large particles does not vary between shelf sea and open ocean boxes. We examine the potential impact of this limitation in the discussion section of the paper.

### 2.2.4 Sediments and burial

SPorg and LPorg accumulate in the sediments as SedP$_{org}$, which is calculated as a density per unit of area. The flux [mmolP m$^{-2}$ yr$^{-1}$] into the sediment box $i$ depends on the organic matter fluxes into the overlaying deep ocean box $j$ and on the remineralization length of the two pools as in:

$$SedFlx^i = (Flx\_SP_{org}{}^j \cdot exp(-\Delta Z_j / z_{rem}^S) + Flx\_LP_{org}{}^j \cdot exp(-\Delta Z_j / z_{rem}^L)) \tag{7}$$

The accumulated SedP$_{org}$ is partially slowly remineralized and partially irreversibly buried in a mineral form. Phosphorus burial as mineral Ca-P is modelled as a function of the square of SedP$_{org}$ that accumulates in the sediments and is regulated by a constant rate coefficient CaP$_r$. Ca-P formation happens at a lower rate under low oxygen conditions (CaP$_r^*$ = CaP$_r$ · fs$_{an}$ with fs$_{an}$ < 1), in agreement with observations and previous models (Slomp and Van Cappellen, 2006). The transition from aerobic and anaerobic conditions is controlled by a Michaelis-Menten type of function of the oxygen concentration in the deep ocean box j overlaying the sediment box i. The oxic and anoxic terms sum to the total formation term [mmol m$^{-2}$ yr$^{-1}$] as in:

$$CaPform^i = \left(SedP_{org}{}^i\right)^2 \cdot [CaP_r \cdot O^j/(O^j + K_O^s) + (CaP_r \cdot fs_{an}) \cdot (1 - O^j/(O^j + K_O^s))] \qquad (8)$$

This flux is essential to balance the continuous P river input, therefore preventing the ocean from overflowing with nutrients.

### 2.2.5 Remineralization in the water column and sediments

At each time step, remineralization in the water column completely depletes the P$_{org}$ that has not reached the sediments. In the two surface boxes, remineralization of P$_{org}$ that is not exported below the euphotic layer uses up part of the oxygen that was released by production. For this reason, net oxygen production in each surface box is proportional to the export of P$_{org}$ below the euphotic layer. The overall loss of P due to export [mmolP m$^{-3}$ yr$^{-1}$] from a surface box $i$ to a deep box $j$ via gravitational sinking, is calculated as:

$$VExp^i = (Flx\_SP_{org}{}^i \cdot exp(-(\Delta Z_{eu}/2)/z_{rem}^S) + Flx\_LP_{org}{}^i \cdot exp(-(\Delta Z_{eu}/2)/z_{rem}^L)/\Delta Z_i \qquad (9)$$

where the fluxes per unit of area of SPorg and LPorg in the surface boxes depend on production and coagulation as described in subsection 2.2.1.

At depth, the remineralization of P$_{org}$ that does not reach the sediments happens through both aerobic and anaerobic processes, completely depleting the remaining P$_{org}$. The amount of inorganic P released in each deep box $j$ by water-column remineralization [mmolP m$^{-3}$ yr$^{-1}$] of P$_{org}$ is therefore calculated as:

$$WcRem^j = (Flx\_SP_{org}{}^j \cdot (1 - exp(-\Delta Z_j/z_{rem}^S)) + Flx\_LP_{org}{}^i \cdot (1 - exp(-\Delta Z_j/z_{rem}^L)))/\Delta Z_j \qquad (10)$$

In each deep ocean box $i$, aerobic remineralization uses some of the available oxygen and is therefore limited by a Michaelis-Menten kinetics with a half-saturation constant K$^w_O$ (DeVries and Weber, 2017). Anaerobic remineralization takes up the entire remaining P$_{org}$ that is not remineralized aerobically and releases a product which "bubbles up" to the atmosphere, reacting with atmospheric oxygen. In our model, the reducing agent produced by anaerobic remineralisation is methane gas and it is only produced when the sediments and the deep shelf water column have gone anoxic. As we do not track other oxidising agents such as SO$_4$, there is nothing for the methane to be oxidised by until it reaches the surface ocean, and as the surface ocean is equilibrated with the atmosphere, the fact that we assume oxidation in the atmosphere is a reasonable approximation. In each sediment box $i$, remineralization of SedP$_{org}$ happens in a similar way to remineralization in the water column, with an aerobic and an anaerobic component. However, remineralization in the sediments is not instantaneous, but happens at a fixed rate which depends on the oxygenation state of the overlaying water column. Aerobic remineralization takes up oxygen from the overlaying deep-water box $j$ and happens at a rate rm$_r$, while being limited by a Michaelis-Menten coefficient. Anaerobic remineralization releases its product to the atmosphere and happens at a faster rate rm$_r^*$ = rm$_r$ ·fe$_{an}$ with fe$_{an}$ >1, in agreement with recent observations and previous models (Slomp and Van Cappellen, 2006). The release of P$_{org}$ from a sediment box $i$ into the overlaying ocean box due to sediment remineralization [mmolP m$^{-3}$ yr$^{-1}$] is therefore the sum of the two terms as in:

$$SedRem^i = (rm_r \cdot SedP_{org}^i \cdot (O^j/(O^j + K_O^s))) + (rm_r \cdot fe_{an}) \cdot SedP_{org}^i \cdot (1 - O^j/(O^j + K_O^s)))/\Delta Z_d$$
(11)

## 2.3 Equations summary

The dynamics of the model's 11 state variables is regulated by just as many equations. We summarize here the major terms for P, O and SedP$_{org}$ in the surface ocean (s), deep ocean (d), atmosphere (at) and sediments, without distinguishing between coastal and open ocean boxes and assuming that all terms have been scaled with the reference box' dimensions or number of moles (atmosphere). A full set of equations including the explicit formulation of all the flux terms for each box can be found in the paper's Appendix.

$$\frac{dP^s}{dt} = P_{in} + AdvMix(P)^s - VExp$$
(12)

$$\frac{dP^d}{dt} = AdvMix(P)^d + WcRem + SedRem$$
(13)

$$\frac{dO^s}{dt} = AdvMix(O)^s + VExp \cdot OP_{Red} + AirSea$$
(14)

$$\frac{dO^d}{dt} = AdvMix(O)^d - WcRem_{Aer} \cdot OP_{Red} - SedRem_{Aer} \cdot OP_{Red}$$
(15)

$$\frac{dSedP_{org}}{dt} = SedFlx - CaPform - SedRem \cdot \Delta Z_d$$
(16)

$$\frac{dO^{at}}{dt} = -\sum AirSea - WcRem_{Ana} \cdot OP_{Red} - SedRem_{Ana} \cdot OP_{Red} - OxyWeath$$
(17)

Where: P$_{in}$ is the river input of P to the ocean's surface, AdvMix indicates the advective and mixing physical fluxes of the variable of interest (which differ for each box according to the circulation scheme); Exp is the export flux of P$_{org}$ in P units; WcRem indicates the water column complete remineralization of the organic material in P units, which is split into an anaerobic (Ana) and aerobic (Aer) component; SedRem indicates the sediment remineralization of SedP$_{org}$ in P units (also aerobic and anaerobic); AirSea represents the air-sea flux exchange of O; OxyWeath is the O weathering flux sink; SedFlx is the SedP$_{org}$ accumulation flux as regulated by the remineralization length scheme at the bottom of the water column; and finally CaPform represents the sediment burial flux of P in mineral form. For each box, flux terms are rescaled with the appropriate box geometry.

## 2.4 Strategy: sensitivity studies for varying z$_{rem}$

In order to characterize the model, we analyse the equilibrium budgets and fluxes of the state variables for varying z$_{rem}$ values separately for SP$_{org}$ and LP$_{org}$, respectively z$_{rem}^S$ and z$_{rem}^L$. We adopt a range of zrem values that fall close to modern observations (Cavan et al., 2017;Buesseler and Boyd, 2009;Marsay et al., 2015) and keeps into consideration our future aim to apply the model to simulate the impact of the time evolution of the early biological pump (at the Neoproterozoic-Palaeozoic transition). For this reason, we don't push the range as far as what would be needed to consider the impact of fast sinking rates typical of silicified or calcified small phytoplankton (McDonnell and Buesseler, 2010;Lam et al., 2011). In our sensitivity simulations, z$_{rem}^S$ is in the range of [0, 40 m], while z$_{rem}^L$ varies in the range of [50 m, 450 m].

**3 Evaluation**

**3.1 Timescales**

Starting from the initial values listed in Table 3, the modelled state variables evolve towards equilibrium for any couple of values of $z_{rem}^S$ and $z_{rem}^L$ in the explored interval. Simple mass conservation checks show no hidden source or sink of tracers in the model's boxes. Figure 3 illustrates an example of evolution of the variables for $z_{rem}^S$ and $z_{rem}^L$ in the middle of the interval of explored values for both particle types. In all the ocean boxes, P shows an initial oscillation that evolves on timescales of tens of thousands of years (Figure 3a,b), as expected by the typical timescale of evolution of the tracer (Lenton and Watson, 2000). This is followed by a slower drift which depends on the dynamics of the deep water oxygen content, as the release and burial of P in the sediments depends on the level of oxygenation of the deep ocean and especially of the deep shelf sea. P reaches complete equilibrium as soon as the deep ocean boxes become stably oxygenated. The timescales of evolution of O are slower and lay on the order of tens of millions of years (Lenton and Watson, 2000). Oxygen in the deep shelf overcomes hypoxia after the first few millions of years and then slowly evolves towards equilibrium on the same timescale of O in the other ocean boxes. The dynamics of SedPorg is also strongly driven by level of oxygenation of the deep shelf sea. The model's dynamical response to changes in the biological pump is rapid, subsequent to the model equilibrating considering the given initial conditions. For example, step changes in the particles' $z_{rem}$ result in a transition time to a new equilibrium that is of the order of a few tens of thousands of years, which is the typical timescale of the P cycle.

**3.2 Modern ocean budgets and fluxes**

Modern estimates of the $z_{rem}^S$ and $z_{rem}^L$ vary depending on the region of sampling and on the local community structure, with most of the measurements focusing on large or heavy particles and most studies focusing on the open ocean (Iversen and Ploug, 2010;Cavan et al., 2017;Lam et al., 2011). Furthermore, only a very limited number of measurements accounts for both microbial and zooplankton remineralization, the latter disregarded by lab measurements of $z_{rem}$ (Cavan et al., 2017). Considering the fundamental role of the shelf sea in our model (always accounting for > 98 % of the total burial), we evaluate modelled tracer budgets and fluxes for values of $z_{rem}^L$ that lay around 76 m, as measured in situ by Cavan et al. (2017) for a modern shelf sea. We pose no restrictions on $z_{rem}^S$ due to the lack of precise measurements. A summary of our evaluation is provided in Table 5.

In the above mentioned range of $z_{rem}$, our model predicts equilibrium budgets of between 2250 TmolP and 2970 TmolP for phosphorus, and an oxygen budget of between 100 $PmolO_2$ and 107 $PmolO_2$ in the entire ocean, compared to the estimated total P reservoir of 3100 TmolP (Watson et al., 2017) and estimated ocean $O_2$ reservoir of between 225 $PmolO_2$ and 310 $PmolO_2$ (Keeling et al., 1993;Duursma and Boisson, 1994). Due to the relative size of the ocean boxes, it is important to underline that total budgets are strongly driven by the deep open ocean budget, and that the low oxygen reservoir of our model may be connected to an underestimation of the deep open ocean oxygenation.

Deep shelf P and O concentrations lay in the ranges of [3.9 mmol m$^{-3}$, 4.9 mmol m$^{-3}$] and [3.8 mmol m$^{-3}$, 9.2 mmol m$^{-3}$] respectively (Figure 5,6). Deep shelf nutrient concentrations are higher than expected by about a factor of two compared to modern values, possibly due to the fact that our model does not store any $P_{org}$ in the water column or due to an underestimation of the vertical supply of nutrients to the surface shelf (e.g., via mixing).

Limiting deep P concentrations via lower remineralization or higher burial rates, however, also results in sensibly lower production rates. In the deep open ocean, P and O concentrations fall in the ranges of [1.9 mmol m$^{-3}$, 2.5 mmol m$^{-3}$] and [76 mmol m$^{-3}$, 83 mmol m$^{-3}$] respectively. For any combination of $z_{rem}^S$ and $z_{rem}^L$, O levels in surface ocean boxes lay between 273 mmol m$^{-3}$ and 274 mmol m$^{-3}$, a good approximation of average modern surface values (Garcia et al., 2018b). In general, the deep shelf always shows the highest P values and lowest O concentrations compared to the other ocean regions, while, as expected, the surface shelf sea is richer in P compared to the surface open ocean.

In order to compare the modelled fluxes to modern estimates, we converted our results into carbon (C) units assuming a C:P Redfield ratio of 106. However, recent studies found a substantially higher mean C:P ratio for the modern ocean (Martiny et al., 2014), therefore our derived C fluxes may be a conservative estimate. Modelled biological fluxes in C units, such as production and export, fall just below the low end of present estimates (Figure 7). Our model predicts a total primary production of between 11 GtC yr$^{-1}$ and 30 GtC yr$^{-1}$, and an export below the euphotic layer ranges between 3.4 GtC yr$^{-1}$ and 3.8 GtC yr$^{-1}$. These must be compared to an expected value of production of between 35 GtC yr$^{-1}$ and 80 GtC yr$^{-1}$ (Carr et al., 2006) and an estimated export flux of at least 4 TmolC yr$^{-1}$ (Henson et al., 2011). Despite the absolute fluxes being at the low end of the present estimates, modelled export production (the export to production ratio) and the burial to production ratio compare well to range of present estimates. The modelled export corresponds to between 11 % and 33 % of total production, strongly depending on $z_{rem}^S$, compared to an expected range of 2 % - 20 % (Boyd and Trull, 2007). Buried P$_{org}$ corresponds to between 0.3 % and 1 % of total production, compared to an expected 0.4 % (Sarmiento and Gruber, 2006).

In terms of the shelf contribution to the total fluxes, model results also fall close to present estimates. Modelled production in the surface shelf sea represents between 16 % and 27 % of total production (expected 20%) (Barrón and Duarte, 2015;Wollast, 1998). The fraction of modelled export and burial that happens in the shelf region represent, respectively, [16 %, 27 %] and nearly 100 % of the total ocean fluxes, compared to estimated modern values of 29 % and 91 % (Sarmiento and Gruber, 2006). Our overestimation of the shelf contribution to the burial fluxes may be due to the underestimation of the open ocean particles $z_{rem}$ compared to observations (Cavan et al., 2017;Lam et al., 2011), i.e. our choice of using the same value of $z_{rem}^S$ and $z_{rem}^L$ for both the coastal and the open ocean box. This simplifying assumption limits the capacity of P$_{org}$ to reach the deep sediment layer in the open ocean. We explore potential limitations of this choice in the Discussion section.

**4 Results**

**4.1 Budgets and fluxes sensitivity to changes in $z_{rem}$**

Around the lowest values of $z_{rem}^L$ adopted in the present study, i.e., in the range of [50 m, 100 m], our model shows a strong sensitivity of the total and local ocean P and O budgets for small changes of $z_{rem}^L$ (Figure 4). This is true for any $z_{rem}^S$, with minor differences between low and high $z_{rem}^S$ values. For smaller $z_{rem}^L$, the model shows a sharp increase in P concentrations in all the ocean boxes and a substantial decrease of O levels at depth (Figures 5,6), which are coupled to high levels of production and remineralization and low rates of sedimentation (Figure 7). Essentially slow sinking and/or rapid remineralization results in inefficient removal of P to shelf sea sediments, requiring the ocean concentration of P to rise considerably for P output to balance (fixed) P input to the ocean.

Our model results show that for any couple of values of $z_{rem}^S$ and $z_{rem}^L$ in the entire explored range, the biological pump is able to oxygenate the surface ocean (surface O levels lay close to 273 mmol m$^{-3}$) and, for most values, also to maintain the deep ocean above the level of hypoxia (Figure 6). The model shows a substantial difference between the deep shelf and the deep open ocean: while the latter is substantially oxygenated (O > 50 mmol m$^{-3}$) for nearly any value of $z_{rem}^S$ and $z_{rem}^L$, the deep shelf is hypoxic or even suboxic for a broad range of small values of $z_{rem}^L$, especially close to modern shelf $z_{rem}^L$ observations. Considering the wide spatial extension of our boxes, we expect these low oxygen levels to indicate the development of local anoxia in the deep shelf.

In a limited interval of small $z_{rem}^S$ values (roughly $z_{rem}^S < 6$ m), model results depend only on the LP$_{org}$ properties due to the rather irrelevant contribution of SP$_{org}$ to export and remineralization. For larger $z_{rem}$ values ($z_{rem}^S > 6$ m and $z_{rem}^L > 100$ m), model results show a strong interdependence of equilibrium budgets and absolute fluxes on both $z_{rem}^S$ and $z_{rem}^L$. Interestingly, in this range of values, export production depends very strongly on the small particle properties, ranging between 10 % for low $z_{rem}^S$ and 30 % for high $z_{rem}^S$, an overall trend that affects also the ratio of deep remineralization to surface production (Figure 7).

It is also important to notice that, for any couple of $z_{rem}^S$ and $z_{rem}^L$, modelled tracer concentrations and fluxes fall in a range of values that never exceeds by orders of magnitude the modern observed values. Considering all of the ocean boxes, P concentrations vary in the range of roughly 0.2 mmol m$^{-3}$ and 9 mmol m$^{-3}$, while O levels lay between 0.5 mmol m$^{-3}$ and 205 mmol m$^{-3}$. Production in carbon units lays in the interval [7.6 GtC yr$^{-1}$, 70.7 GtC yr$^{-1}$].

**4.2 Budgets and fluxes contribution by particle class**

The relative role of small and large particles to modelled biological and physical fluxes depends on a combination of their inherent properties ($z_{rem}$) and of coagulation. In our simple model, coagulation of SP$_{org}$ into LP$_{org}$ after production in surface boxes affects a constant fraction (cg$_f$ = 0.22) of the produced particles. This fraction was determined by model tuning to modern ocean conditions, and lays close to modern ocean observations of the large particle fraction (15% of the total particles) at export depth (Cavan et al., 2017).

For $z_{rem}^L > 100$ m, LP$_{org}$ efficiently remove P from the water column, limiting production. The contribution of SP$_{org}$ to the total export below the euphotic layer, however, is strongly dominated by the value of $z_{rem}^S$, with a null contribution to export for all values of $z_{rem}^S < 10$ m and increasing values above it. This trend is reflected in the deep-water small particle fraction (Figure 8c,d). Small particles contribute up to 73 % to export in both ocean boxes, and up to 60 % to the sediment accumulation in the shelf sea, with the highest contribution to sediment accumulation being reached for large $z_{rem}^S$ and low $z_{rem}^L$. Our model highlights therefore the different role of large and small particles in the determination of the equilibrium budgets and fluxes. Coagulation into large (fast sinking, less liable) particles is essential to maintain high enough sedimentation and burial rates, therefore allowing O accumulation in the system. At the same time, small (slow sinking, more liable) particles tune the total magnitude of export and remineralization below the euphotic layer, affecting the distribution of oxygen and nutrients throughout the water column.

**5 Discussion**

**5.1 Model limitations and robustness**

**5.1.1 General limitations**

BPOP consists in a simple box model with 4 ocean boxes, 2 sediment boxes and 1 atmospheric box. As with every
box model, BPOP only allows a very rough and fundamental representation of the ocean's topography and
circulation as well as of the exchange fluxes between ocean, atmosphere and sediments. Even though this may be
a limitation in the context of the study of the well-known modern (and future) ocean, such a computationally
inexpensive model can be a useful tool to for a first exploration of a large variety of projected conditions. In the
context of understanding past ocean changes, often characterized by a limited availability of observational data,
the use of such a simple model constitutes instead an effective and honest approach to understand global shifts in
budgets and fluxes. Furthermore, BPOP explicitly distinguishes between the well sampled shelf sea and the less
known open ocean of deep time, therefore allowing to relate shelf data with large scale open ocean conditions.
The model deliberately simplifies the redox carriers and processes represented, neglecting denitrification and iron
and sulphate reduction. Including additional oxidants and/or methane consumption in deeper water column would
be expected to intensify anoxia results at depth. However, our current results suggest that the model is overall
underestimating the ocean total oxygen budget, mostly driven by the deep open ocean reservoir. This suggests
that neglecting these additional processes in our simple box model does not lead to an overestimation of oxygen
accumulation at depth. Including additional state variables and processes could also lead to more complex
dynamical behaviours (Wallmann et al. 2019).

We include anaerobic remineralisation of $P_{org}$ being faster than aerobic degradation, but in reality this is not the
case for carbon – which is remineralised at a similar or slower rate under anoxic versus oxic conditions (Burdige,
2007;Hedges et al., 1999;Dale et al., 2015). Hence, in reality, under anoxic conditions, there is preferential
regeneration of phosphorus and organic C:P burial ratios rise considerably, altering the long-term steady state of
atmospheric oxygen (Van Cappellen and Ingall, 1996). We do not consider these aspects here, because to do so
would require adding state variables for organic carbon (as distinct from organic phosphorus), and because our
focus here is on changes in ocean phosphorus and ocean redox under an unchanged oxygen steady state. In future
work we intend to elaborate the model to explore long-term effects on atmospheric oxygen.

**5.1.2 Limitations connected to the biological pump representation**

In our model we adopt a very simplified representation of the biological pump, including two particle classes,
"small" and "large", generated by production and coagulation, assuming that, on average, $z_{rem}^S < z_{rem}^L$. This
scheme resembles the one commonly used in ocean biogeochemical models (Gruber et al., 2006;Jackson and
Burd, 2015). Our model does not include a DOM pool for reasons mostly connected to the implicit representation
of the biological pump and the complete remineralization of the non-sedimented organic material at each
integration step. For the same reason, we do not resolve particle $P_{org}$ concentrations and therefore we model the
coagulation flux as a constant fraction of production. A more physical representation of coagulation would require
this flux to scale with the square of the particle concentrations (Boyd and Trull, 2007). Such a further development
could potentially lead to increase large particle export for high surface P concentrations leading to high production

and particle concentrations (and viceversa). We reserve this improvement as our first step for further model developments, which will include an explicit $P_{org}$ representation.

Modelled particles get remineralized through the water column according to their characteristic $z_{rem}$. Even though for simplicity we do not use a continuum spectrum of $z_{rem}$, the use of two particles classes is in line with observations showing two distinct peaks in the observed distribution of particles' sinking speeds (Riley et al., 2012;Alonso-González et al., 2010). Furthermore, this simplification still allows to closely approximate the empirical particle flux curve as a function of depth, also known as Martin's curve (Boyd and Trull, 2007).

We assume that $z_{rem}^S$ and $z_{rem}^L$ do not vary between the shelf sea and the open ocean. However, modern ocean observations show cross-shore changes in the phytoplankton community structure and sinking speeds (Barton et al., 2013). Our simplifying assumption may therefore cause the overestimation of the relative contribution of the shelf sea to the total burial flux of $P_{org}$. Despite this, we believe that this choice is still convenient in the context of the current model, as it allows us to reduce the number of parameters in such a simple box model representation of the ocean's biological pump.

Observations suggest that hard shelled phytoplankton types, especially calcified cells, contribute substantially to the vertical export and burial of the organic material thanks to extremely large $z_{rem}$ despite their small size (Lam et al., 2011;Iversen and Ploug, 2010). In the present study we focus on an interval of $z_{rem}^S$ and $z_{rem}^L$ values that are most likely to resemble the biological pump conditions of the Neoproterozoic - early Paleozoic ocean, before the evolution of such phytoplankton types. However, the model allows to explore different ranges of $z_{rem}^S$ and $z_{rem}^L$ values and to tune the rate of coagulation in order to explore the influence of these phytoplankton classes.

Even though bacterial remineralization is thought to be the dominant pathway for organic matter recycling on a global scale, especially at low latitudes (Rivkin and Legendre, 2001), modern ocean coastal environments are also characterized by high grazing rates. The evolution of zooplankton and increasingly large grazers may have had a different impact on the effective $z_{rem}^S$ and low $z_{rem}^L$, given additional $P_{org}$ transformations such as particle fragmentation due to sloppy feeding (Cavan et al., 2017;Iversen and Poulsen, 2007). These processes can limit the large particle burial rates, while resulting in the deep production of small particle, s-POM and DOM. Our model does not currently account for particle fragmentation, however the process could be easily considered in future model developments. In this context, new processes such as the sedimentation and burial of large grazers should also be considered.

### 5.1.3 Sensitivity to parameter choices

We discuss here the model sensitivity to changes in a set of significant parameters adopted to describe its geometry, circulation and biological processes. Overall, none of the sensitivity experiments showed significant changes in the model results and conclusions: trends in budgets and fluxes obtained varying $z_{rem}^S$ and $z_{rem}^L$, as well as our main results regarding the relative deep shelf and open ocean oxygenation remain unchanged.

Among the geometrical box model parameters, a key value is represented by the percentage of shelf sea area ($\mathcal{P}_{shelf}$). An increase (e.g., doubling) in $\mathcal{P}_{shelf}$ results in an overall decrease in the total budget of P and increase in O due to the larger ratio of burial to production, which is facilitated by a larger extension of the surface of shallow water. Interestingly, deep shelf anoxia is enhanced for larger $\mathcal{P}_{shelf}$, i.e., anoxia is observed for a wider range of $z_{rem}^S$ and $z_{rem}^L$ values, while the deep ocean tends to be more oxygenated. Despite a doubling of $\mathcal{P}_{shelf}$, however, model results largely remain in the same range of those found for modern $\mathcal{P}_{shelf}$.

We explored the effect of varying the physical circulation parameters. Changes in upwelling (Upw), have an important impact on the modelled ocean's budgets. An increase in Upw induces a lowering of P levels, especially in the deep shelf, due to their recirculation towards the surface and consequent uptake by production. This is coupled to an overall larger equilibrium O budget due to higher storage in the deep open ocean, and consequent recirculation into the deep shelf. Deep shelf suboxia is still possible, but for a more limited range of $z_{rem}^{L}$ values. Changes in vertical mixing in the open ocean ($Mix_{vo}$) affect the overall P and O budgets mostly for high $z_{rem}^{L}$. For lower $Mix_{vo}$, the O budget decreases due to lower O storage at depth, while P increases. Changes in vertical mixing on the shelf ($Mix_{vs}$), instead, have a minor impact on the model's total budgets and fluxes, while locally modulating shelf oxygen and nutrient concentrations. Lateral mixing fluxes ($Mix_{ls}$, $Mix_{ld}$) were included in our model for means of generalization and in order to account for the influence of non-upwelling margins, with a lower value than in previous studies (Fennel et al., 2005). Changes in $Mix_{ls}$ and $Mix_{ld}$ result in significant changes in the deep ocean storage of tracers and on open ocean production, with little impact on the budget of the other ocean boxes. However, also in this case, our main conclusions remain unaffected.

We explored the impact of changing the portion of nutrients delivered directly to the open ocean, $\mathcal{P}_{open}$. Even large changes in this parameter do not significantly affect the model's results, indicating that the relative levels of P and O at equilibrium are determined by the internal physical and biogeochemical dynamics of the model, rather than by boundary conditions.

Lastly, we explored the model sensitivity to the choice of key biogeochemical parameters representing rates of transformation. Both increasing coagulation ($cg_f$) and the use of higher rates of formation of mineral Ca-P ($CaP_r$) result in a general increase in O levels and decrease in nutrient availability due to larger sedimentation and burial rates. However, we find again no substantial change in the model behaviour nor in the relative contribution to budgets and fluxes of each modelled ocean box.

Furthermore, we have tested the impact of having sediment remineralization rates that vary with the particles' $z_{rem}$, under the assumption that the liability of small and large particles may be different. In our experiment, we increased the remineralization $rm_r$ rate linearly with $z_{rem}$ by 40 % of our baseline value ($rm_r^{0}$), with $rm_r^{0}$ being found at the centre of the interval of explored values of $z_{rem} = [0\ m, 450\ m]$. Under these conditions, we obtained a higher decoupling between the influence of $z_{rem}^{S}$ and $z_{rem}^{L}$ on budgets and fluxes, both being more strongly driven by the small particle properties for large values of $z_{rem}^{L}$.

**5.2 Model applications**

**5.2.1 Past changes in the biological pump**

The evolution of larger and heavier cells during the Neoproterozoic and across the Neoproterozoic-Paleozoic transition is hypothesised to have caused significant changes in the ocean's nutrient and redox state (Lenton and Daines, 2018). Our new model can be used to assess the impact of this evolution in both the shelf and the open ocean. Our first model results highlight that for small $z_{rem}^{L}$, i.e., for an early biological pump with reduced capacity of export and burial, nutrient levels and production rates are particularly high. At the same time, an increase in $z_{rem}^{S}$ alone, fuelling higher remineralization rates at depth, can induce anoxia in the deep shelf while still maintaining the deep open ocean substantially oxygenated. The possibility of a coexistence of an anoxic deep shelf with an oxygenated deep open ocean has important implications for the interpretation of deep time redox

proxy data, which come almost exclusively from shelf and slope environments, yet have been widely used to infer deep ocean anoxia for most of the Proterozoic Eon (Lenton and Daines, 2017) . We plan to use our model to further explore these changes in a time-frame perspective, introducing time varying boundary conditions (such as changes in $P_{in}$) and parameter properties.

Phytoplankton evolution as well as the development of heavier and larger marine organisms continued throughout the Phanerozoic (Katz et al., 2007). BPOP can also be used to explore the role of the biological pump in the onset of OAEs in the course of the Mesozoic era, likely induced by enhanced productivity due to an upwelling intensification (Higgins et al., 2012). During the Mesozoic era, the evolution of dinoflagellates, calcareous and silica-encased phytoplankton also likely impacted the export and burial rates in a significant way (Katz et al.,

2004). By extending the range of explored values of $z_{rem}^S$ and $z_{rem}^L$, or possibly including the effect of grazing and/or an additional heavy POC class for shelled organisms, BPOP can also be used to study the consequences of such evolution.

### 5.2.2 Future changes in the biological pump

Predicted future changes connected to global warming include, among the others, changes in ocean temperature,

pH and stratification (Gruber et al., 2004), with additional repercussions on plankton community structure, production, remineralization and export rates  (Laufkötter et al., 2017;Acevedo-Trejos et al., 2014;Kwon et al., 2009). Our results show that around values of $z_{rem}^L$ measured for modern shelf environment (Cavan et al., 2017) modelled equilibrium budgets and fluxes are very sensitive to small changes in $z_{rem}$. This indicates a potentially high sensitivity of the modern ocean to small changes in the biological pump, which may be particularly important

in the deep shelf, where the boundary with suboxia is especially close (Keeling et al., 2010). Our model can be used to get a first assessment of the large-scale combined effect of predicted changes in the biological pump with expected shifts in the physical ocean properties.

### 5.2.3 Exploring past and future changes in geometry, physics and biogeochemistry

In the present study we have focused on the impact of changes of $z_{rem}^S$ and $z_{rem}^L$ on the equilibrium budget and

25 fluxes in the ocean. However, BPOP can be used to explore the effect of global changes in other physical or biogeochemical processes coupled to the biological pump dynamics. Aside from testing the robustness of our results, the sensitivity tests presented in subsection 5.1.3 serve also as a first exploration of the possibility to apply the model to these further studies. We discuss here a few examples of past changes that could be explored with the present model.

Through Earth's history, variations in the distribution of continents and in the mean sea level height likely impacted the percentage of shelf sea area ($\mathcal{P}_{shelf}$) throughout the global ocean (Katz et al., 2007). Changes in climate and therefore in the mean temperature are expected to have affected both the air-sea gas exchange of oxygen (Schmidt number, $N_{Sch}(T_{mean})$) and vertical mixing ($Mix_{vo}$) (Petit et al., 1999). Reduced vertical mixing in warm periods is also expected to be relevant in the future because of global warming (Gruber et al., 2004).

Changes in temperature are also known to impact biological activity directly, e.g,, by increasing remineralization rates ($rm_r$) (Laufkötter et al., 2017), and indirectly, e.g., affecting production and mortality rates through changes in the mixed layer depth (Polovina et al., 1995). Climatic shifts can also cause changes in the intensity of alongshore winds and therefore in the upwelling circulation (Sydeman et al., 2014). Lastly, the model can be used

to test the impact of changes in the biogeochemical cycles, including shifts in the Redfield ratio as well as global changes in the P input ($P_{in}$) to the ocean (Reinhard et al., 2017;Filippelli, 2008).

**6 Conclusions and Outlook**

This paper provides a description, evaluation and discussion of the new BPOP model. BPOP is aimed at exploring the effects of changes in the biological pump on the shelf and open ocean nutrient and redox state as well as on P and O fluxes. This model can be adopted for a large variety of studies aimed at exploring the impact of changes in the biological pump, i.e., the particle remineralization length scale $z_{rem}$, in past and future ocean settings. Furthermore, it allows to couple changes in POM properties to changes in the ocean's geometry, circulation and boundary conditions.

Despite its simple representation of the ocean circulation and of the biological pump, the model can reasonably simulate values of the current P and O tracer budgets and biological pump fluxes. The model predicts potentially large variations in these P and O budgets and fluxes for past and future changes in the POM remineralization length. Our preliminary results also indicate that the early ocean may have been nutrient rich, with high levels of production and remineralization and that a suboxic deep shelf setting may have been compatible with an oxygenated deep open ocean.

We plan to apply this model to study the time evolution of the P and O budgets in both the shelf and the open ocean environment across the Neoproterozoic-Phanerozoic transition. Further developments of the model will be aimed at accounting for successive evolutionary innovations, including particle fragmentation due to grazing.

**Code availability**

The code is available for download in the supplementary material of the present publication, which also includes the user's manual.

**Author contributions**

TL and EL conceived the study. EL conceived and implemented the model. EL and TL evaluated and improved the model. Both authors contributed to the interpretation of the results, and to the writing of the present manuscript.

**Competing interests**

The authors declare that they have no conflict of interest.

**Acknowledgements**

10 We would like to thank Dr Richard Boyle for his valuable input and suggestions. We further acknowledge the precious input of the three referees and of the editor, which substantially improved model and manuscript. This research was funded by NERC in the framework of the project Biosphere Evolution, Transitions & Resilience (BETR).

35

| Name | Description | Units |
|---|---|---|
| $P^{ss}$ | Inorganic phosphorus in surface shelf sea box | mmol m$^{-3}$ |
| $P^{ds}$ | Inorganic phosphorus in deep shelf sea box | mmol m$^{-3}$ |
| $P^{so}$ | Inorganic phosphorus in surface open ocean box | mmol m$^{-3}$ |
| $P^{do}$ | Inorganic phosphorus in deep open ocean box | mmol m$^{-3}$ |
| $O^{ss}$ | Molecular oxygen in surface shelf box | mmol m$^{-3}$ |
| $O^{ds}$ | Molecular oxygen in deep shelf box | mmol m$^{-3}$ |
| $O^{so}$ | Molecular oxygen in surface open ocean box | mmol m$^{-3}$ |
| $O^{do}$ | Molecular oxygen in deep open ocean box | mmol m$^{-3}$ |
| $O^{at}$ | Oxygen mixing ratio in atmosphere (mol mol$^{-1}$) | - |
| $SedP_{org}^{s}$ | Organic phosphorus in the sediments of the shelf sea | mmol m$^{-2}$ |
| $SedP_{org}^{o}$ | Organic phosphorus in the sediments of the open ocean | mmol m$^{-2}$ |
| $P^{TOT}$ | Diagnostic variable: total P budget from sources and sinks only | Tmol P |
| $O^{TOT}$ | Diagnostic variable: total O budget from sources and sinks only | Pmol O$_2$ |

**Table 1: List of the model's state variables and of their units**

| Name | Description | Value | Units | Source |
|---|---|---|---|---|
| $Mol_{atmo}$ | Millimoles of air in atmospheric box | $1.8 \cdot 10^{23}$ | mmol | - |
| $\Delta Z_{eu}$ | Depth of the euphotic layer in shelf and open ocean | 100 | m | [1] |
| $\Delta Z_{ds}$ | Depth of the deep shelf sea box | 100 | m | [2] |
| $\Delta Z_{do}$ | Depth of the deep open ocean box | 3500 | m | [3] |
| $A_{ocean}$ | Total area covered by the ocean | $361 \cdot 10^{12}$ | m$^2$ | - |
| $\mathcal{P}_{shelf}$ | Fraction of the total ocean area currently covered by the shelf sea ($\leq$ 200 m deep) | 0.07 | - | Barrón and Duarte (2015) |

**Table 2: Parameters set that describes the box model's geometry: [1] we assume a constant average euphotic layer**
10    **depth of 100 m in both shelf and open sea; [2] the shelf sea is assumed to be 200 m deep in total, in line with the definition of shelf sea by Barrón and Duarte (2015); [3] we assume an average open ocean depth of 3600 m (including euphotic layer).**

| Name | Description | Value | Units | Source |
|:---:|:---|:---:|:---:|:---:|
| $P_{ini}$ | Initial P concentration for all the ocean boxes | 2.2 | mmol m$^{-3}$ | Watson et al. (2017) |
| $O_{ini}$ | Initial O concentration for all the ocean & atmosphere boxes | 0 | mmol m$^{-3}$ | - |
| $(P_{org})_{ini}$ | Initial $P_{org}$ in all the sediment boxes | 0 | mmol m$^{-3}$ | - |
| $Upw$ | Upwelling cell mass fluxes | 6 | Sv | [1] |
| $Mix_{vo}$ | Vertical mixing in the open ocean | 40 | Sv | [2] |
| $Mix_{ls}$ | Lateral mixing at the surface | 0.5 | Sv | [3] |
| $Mix_{ld}$ | Lateral mixing at depth | 0.5 | Sv | [3] |
| $Mix_{vs}$ | Vertical mixing in the shelf sea | 1 | Sv | [4] |
| spy | Seconds per year conversion factor (Sv to m$^3$ yr$^{-1}$) | 31557600 | s yr$^{-1}$ | - |
| $P_{in}$ | Total P river input | $92 \cdot 10^{12}$ | mmol yr$^{-1}$ | Slomp and Van Cappellen (2006) |
| $\mathcal{P}_{open}$ | Fraction of river input delivered to the open ocean | 0.4 | - | [5] |
| $OP_{Red}$ | Oxygen to phosphorus Redfield ratio | 150 | - | Anderson and Sarmiento (1994) |
| $T_{mean}$ | Global mean temperature for oxygen's Schmidt number | 17.64 | ºC | Sarmiento and Gruber (2006) |
| $W_{speed}$ | Global mean wind speed for oxygen gas transfer velocity | 7.5 | m/s | Sarmiento and Gruber (2006) |
| $K_{Henri}$ | Henry's law constant | $770 \cdot 10^{-6}$ | m$^3$ atm mmol$^{-1}$ | - |
| $p_{at}$ | Atmospheric pressure at sea level | 1 | atm | - |
| $Omix_0$ | Today's oxygen mixing ratio in atmosphere | 0.21 | - | - |
| $W_0$ | Baseline oxidative weathering flux coefficient | $9.752 \cdot 10^{15}$ | mmol yr$^{-1}$ | [6] |

**Table 3: Parameters set pertaining to the model's initial conditions, circulation mass fluxes, boundary fluxes. Notes: [1] Chavez and Messié (2009) estimate 5.5 Sv in the four major upwelling systems alone; [2] compare to: 38 Sv (Sarmiento and Gruber, 2006), 17 Sv of mixing flux in the Southern Ocean alone (Meyer et al., 2015), estimated open ocean downwelling 38.5 Sv and upwelling 34.5 Sv (Ganachaud and Wunsch, 2000); [3] cross-shelf mass exchange due to lateral recirculation, tides and mixing aimed at including exchange processes other than upwelling (Fennel et al., 2005;Cole et al., 2015;Wollast, 1998); [4] minimal assumption for vertical mixing in nearshore regions due to seasonal and eddy mixing, see also subsection 3.2 Sensitivity to parameter choices; [5] up to 70% of river outflow reaches the open ocean, see Sharples et al. (2017); [6] calculated from the equilibrium solution given $P_{in}$.**

| Name | Description | Value | Units | Source |
|------|-------------|-------|-------|--------|
| $P_{eff}$ | P maximum uptake rate for production | 0.8 | yr$^{-1}$ | [1] |
| $K_P$ | Michaelis Menten constant for P uptake | 0.2 | mmol m$^{-3}$ | [2] |
| $K^s_O$ | Michaelis Menten constant for aerobic remineralization in the sediments | 0.2 | mmol m$^{-3}$ | [3] |
| $K^w_O$ | Michaelis Menten constant for aerobic remineralization in the watercolumn | 15 | mmol m$^{-3}$ | [4] |
| $cg_f$ | Coagulation fraction, determining the portion of small Porg production routed into large Porg | 0.22 | - | [5] |
| $rm_r$ | Remineralization rate of sedimented Porg | 0.36 | yr$^{-1}$ | [6] |
| $fe_{an}$ | Remineralization enhancement factor under anoxia | 1.25 | - | Slomp and Van Cappellen (2006) |
| $CaP_r$ | Rate of formation of Ca-P mineral from sedimented Porg | 0.5 | (mmol m$^{-2}$)$^{-1}$ yr$^{-1}$ | [7] |
| $fs_{an}$ | Ca-P formation dampening factor under anoxia | 0.5 | - | Slomp and Van Cappellen (2006) |

**Table 4: Parameters set pertaining to the model's Porg cycle and coupled biogeochemical fluxes: [1] maximum P uptake rate, meant to account for environmental limitations of phytoplankton growth rate (such as light and temperature), the magnitude of the rate keeps into account that we are not explicitly resolving phytoplankton concentrations (order of 10$^{-2}$ mmolP m$^{-3}$), see also production in Gruber et al. (2006) and Yool & Tyrrell (2003); [2] measured values vary in the range of 0.01 mmol m$^{-3}$ up to a few mmol m$^{-3}$, varying for different phytoplankton types, see Lomas et al. (2014), Tantanasarit et al. (2013), Krumhardt et al. (2013), Lin et al. (2016), Klausmeier et al. (2004); [3] measured half-saturation constant for oxygen uptake varies in the range of 0.1 - 3 mmol m$^{-3}$ (Ploug, 2001); [4] biogeochemical models commonly switch to anaerobic respiration below 4 mmol m$^{-3}$ (Paulmier et al., 2009), measurements suggest a value close to 19 mmol m$^{-3}$ (DeVries and Weber, 2017); [5] Cavan et al. (2017) shows that small particles are about 85% of the total sinking particles abundance in the coastal region at export depth, the parameter was further tuned to bring the model closer to modern ocean conditions; [6] on the same order of magnitude as Gruber et al. (2006); [7] unmeasured – given the analogous adopted functional form, we assume Ca-P formation to happen on a timescale close to that of $P_{org}$ coagulation in the water column in models with explicit particle pools (Gruber et al., 2006).**

| Quantity | Model | Modern values or estimates | Units | Source |
|---|---|---|---|---|
| Total ocean P | 2250 - 2970 | 3100 | TmolP | Watson et al. (2017) |
| Total ocean $O_2$ | 100 - 107 | 225-310 | $PmolO_2$ | Duursma and Boisson (1994);Keeling et al. (1993) |
| $P^{ss}$ | 1.4 – 2 | 1 – 1.5 | mmol m$^{-3}$ | Garcia et al. (2018a);Sarmiento and Gruber (2006) |
| $P^{ds}$ | 3.9 – 4.9 | 2.2 | mmol m$^{-3}$ | Garcia et al. (2018a);Watson et al. (2017) |
| $P^{so}$ | 0.4 – 0.9 | 0.2 - 2 | mmol m$^{-3}$ | Garcia et al. (2018a);Sarmiento and Gruber (2006) |
| $P^{do}$ | 1.9 – 2.5 | 1 - 3 | mmol m$^{-3}$ | Garcia et al. (2018a);Sarmiento and Gruber (2006) |
| $O^{ss}$ | 273 - 274 | 200 - 350 | mmol m$^{-3}$ | Garcia et al. (2018b) |
| $O^{ds}$ | 3.8 – 9.2 | 0 - 80 | mmol m$^{-3}$ | Garcia et al. (2018b) |
| $O^{so}$ | 273 | 200 - 350 | mmol m$^{-3}$ | Garcia et al. (2018b) |
| $O^{do}$ | 76 - 83 | 40 - 200 | mmol m$^{-3}$ | Garcia et al. (2018b) |
| Production (Prod) | 11 – 30 | 35 - 80 | GtC yr$^{-1}$ | Carr et al. (2006) |
| Export | 3.4 – 3.8 | 4 – 20 | GtC yr$^{-1}$ | Henson et al. (2011) |
| Export production | 11 % - 33 % | 2 % - 20 % | of total Prod | Boyd and Trull (2007) |
| Burial | 0.3 % - 1 % | 0.4 % | of total Prod | Sarmiento and Gruber (2006) |
| Shelf sea production | 16 % - 27 % | 20 % | of total Prod | Barrón and Duarte (2015);Wollast (1998) |
| Shelf sea export | 16 % - 27 % | 29 % | of total Export | Sarmiento and Gruber (2006) |
| Shelf sea burial | 100 % | 91 % | of total Burial | Sarmiento and Gruber (2006) |

**Table 5: Summary of the model evaluation provided in section 3. Modern observations and estimates are compared to model results obtained for $z_{rem}^{L}$ in the range of measured values for a modern shelf sea (Cavan et al., 2017).**

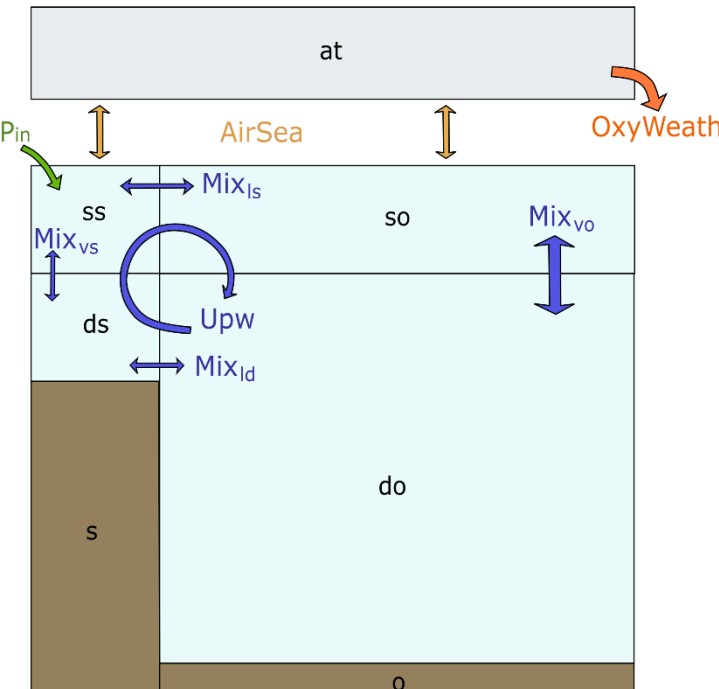

**Figure 1: Box model scheme with a representation of the physical and boundary fluxes affecting inorganic tracers in the water column and atmosphere, where blue arrows indicate advective and mixing fluxes and yellow arrows indicate air/sea gas exchange fluxes. The model includes 7 boxes: surface shelf (ss), deep shelf (ds), surface open ocean (so), deep open ocean (do), atmosphere (at), shelf sediments (s), open ocean sediments (o).**

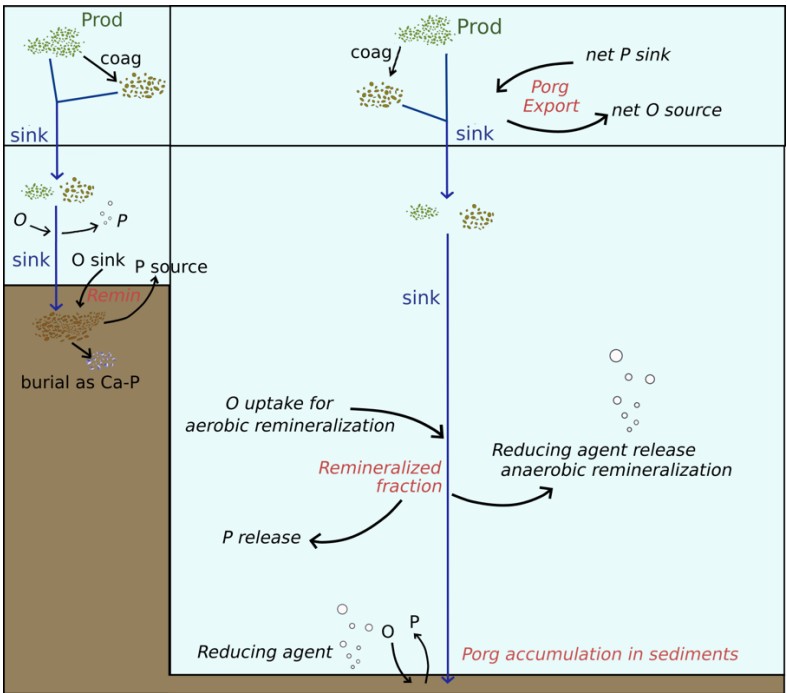

**Figure 2: Representation of the physical and biogeochemical fluxes affecting the $P_{org}$ cycling in the model. Even though some processes (such as burial as Ca-P) are here represented in detail only in one box, the set of biogeochemical processes regulating the $P_{org}$ dynamics in shelf sea and open ocean (both water column and sediments) is the same, as described in subsection 2.2.**

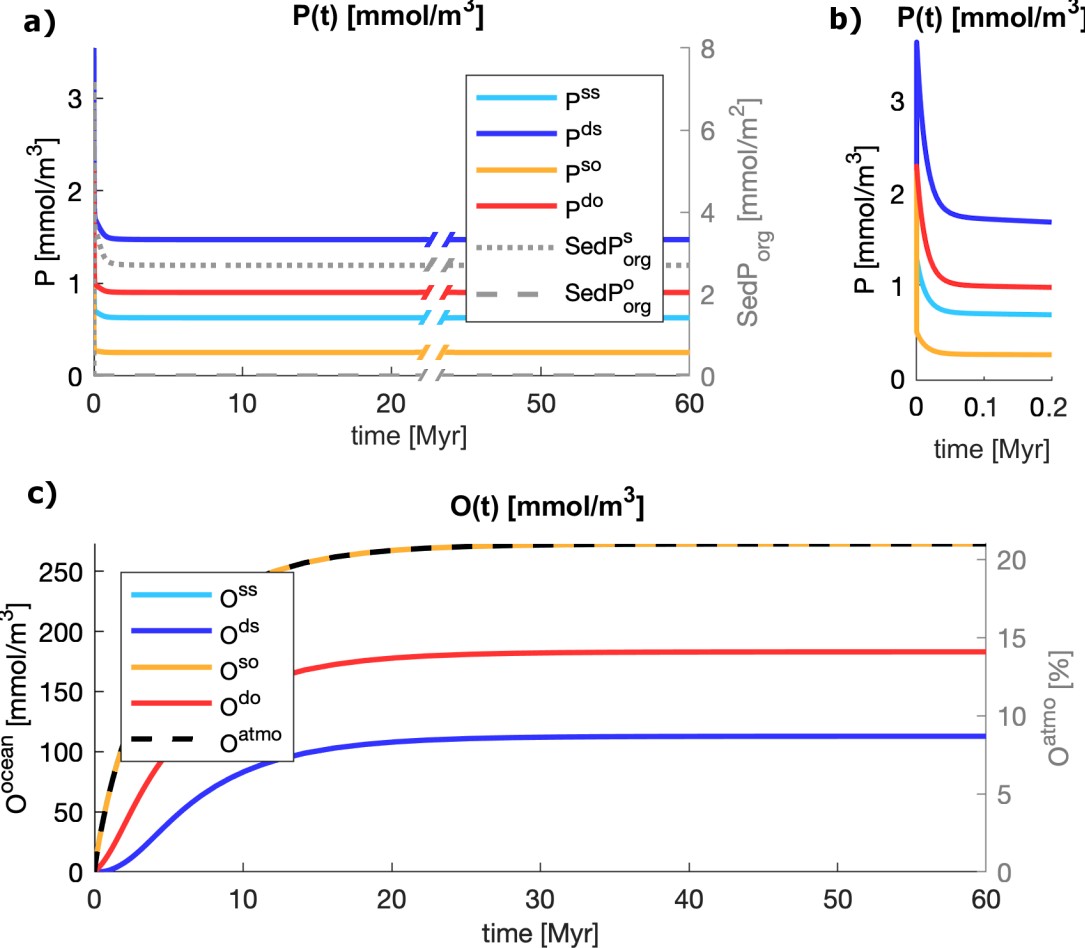

**Figure 3: Evolution of the state variables from the initial conditions listed in Table 2 and remineralization lengths roughly in the middle of the interval of explored values: $z_{rem}^S = 20$ m, $z_{rem}^L = 250$ m. (a) Evolution of inorganic phosphorus P in the water column (left axis) and of organic phosphorus in the sediments $SedP_{org}$ (right axis); (b) zoom on the dynamics of P in the first two hundred thousand years; (c) Evolution of oxygen in the water column (left axis) and atmosphere (right axis). In subplot (c) the two lines $O^{ss}$ and $O^{so}$ are overlapping: the two variables evolve closely due to the coupling of the surface ocean with the atmosphere via air-sea gas exchange.**

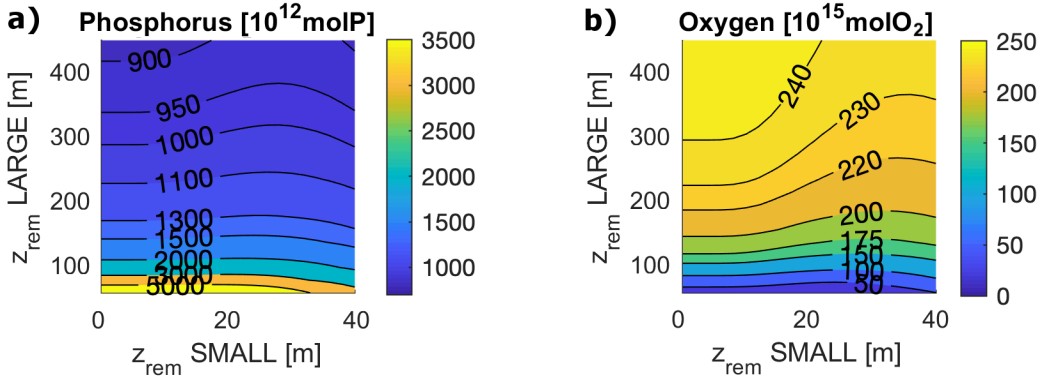

**Figure 4: Total ocean budgets of (a) P and (b) O at equilibrium for varying $z_{rem}^S$ and $z_{rem}^L$.**

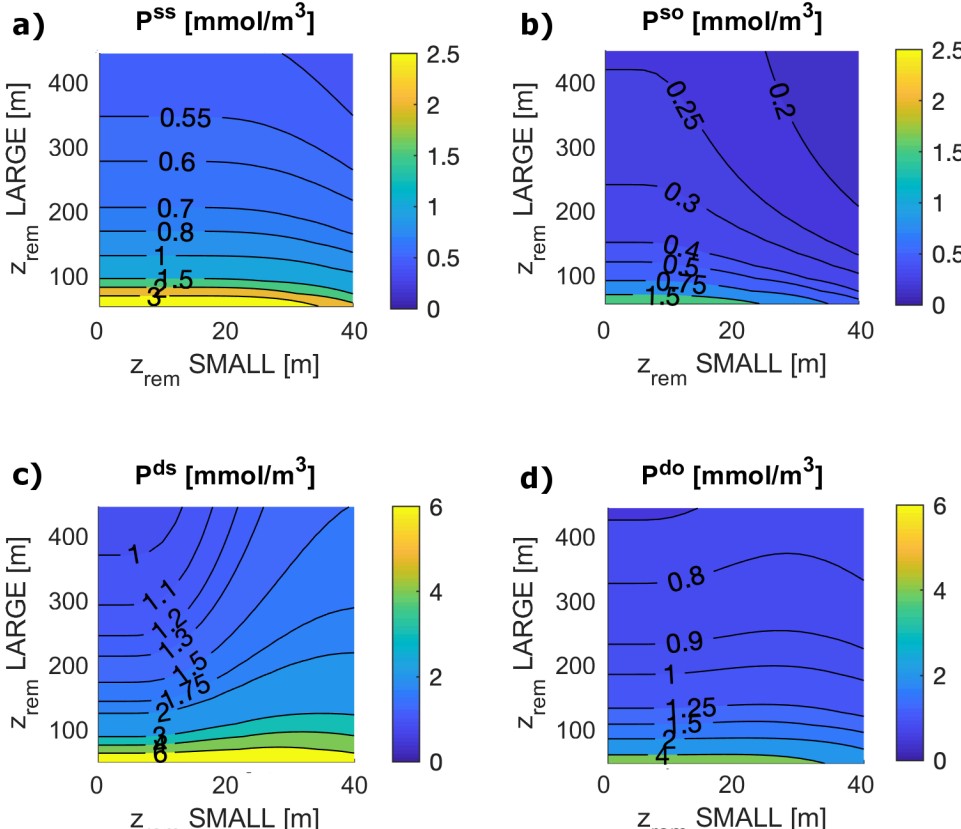

**Figure 5: Local P concentration in each ocean box for varying $z_{rem}^S$ and $z_{rem}^L$: (a) surface shelf sea, ss; (b) surface open ocean, so; (c) deep shelf sea, ds; (d) deep open ocean, do. Surface ocean boxes, as well as deep ocean boxes, are plotted on the same scale.**

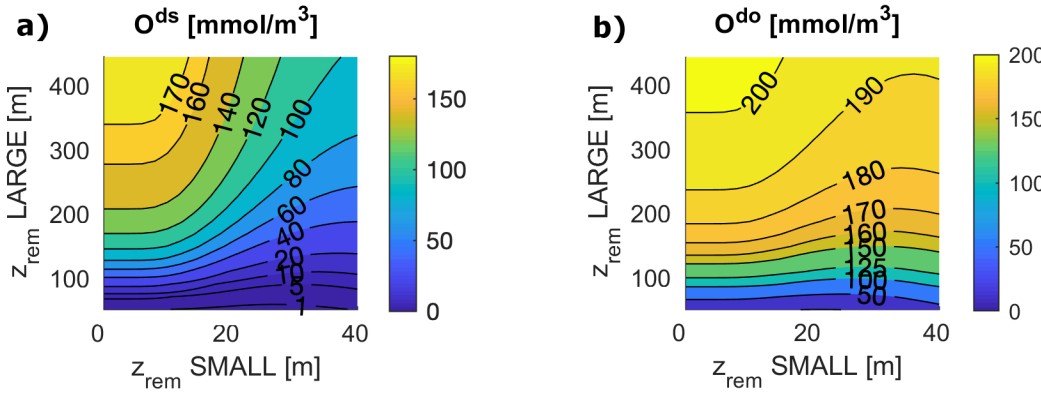

**Figure 6: O concentrations at equilibrium for varying $z_{rem}^S$ and $z_{rem}^L$: (a) deep shelf sea, ds; (b) deep open ocean, do. Surface ocean boxes (not shown) have nearly constant values of O for any set of $z_{rem}$ due to the air-sea gas exchange, which strongly couples them to the atmosphere.**

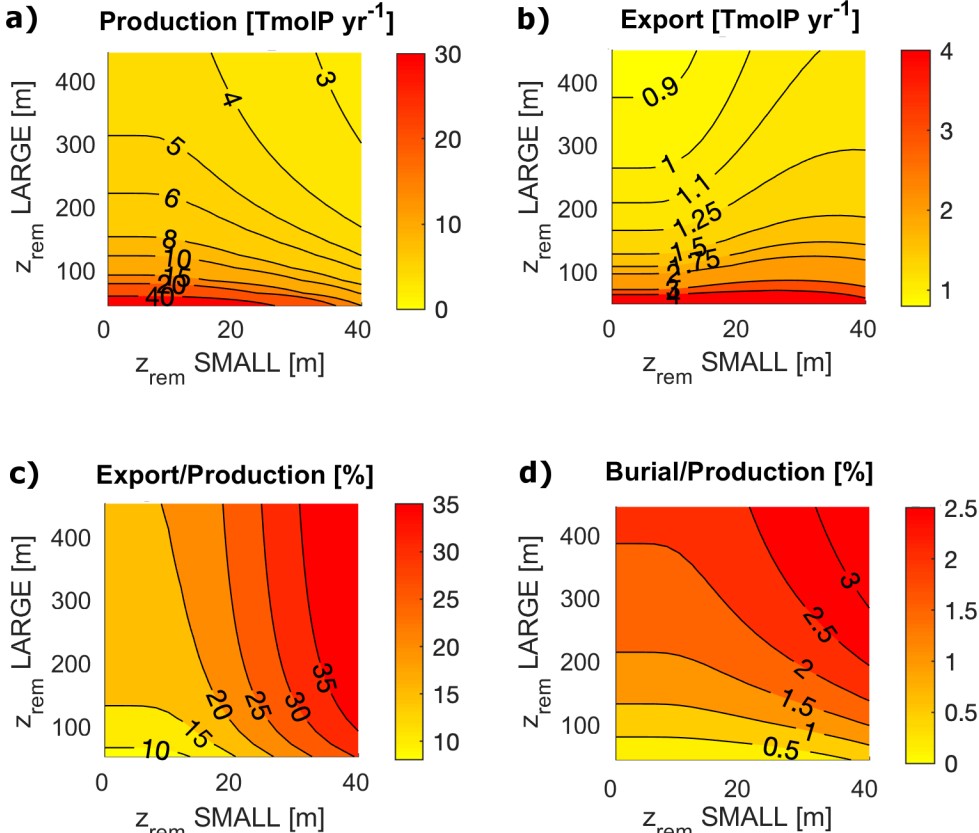

**Figure 7: Biological pump fluxes in P units for the entire ocean for varying $z_{rem}^S$ and $z_{rem}^L$: (a) $P_{org}$ surface production; (b) $P_{org}$ export through the euphotic layer depth; (c) Export production, i.e. export to production ratio (d) Burial to production ratio.**

**Appendix A: Equations**

**A.1 Air-sea gas exchange of oxygen**

$$N_{Sch} = 1638 - 81.83 \cdot T_{mean} + 1.483 \cdot T_{mean}^2 - 0.008004 \cdot T_{mean}^3 \tag{A1}$$

$$K_W = 0.31 \cdot W_{speed}^2 \cdot \sqrt{660/N_{Sch}} \cdot 10^{-2} \cdot (24 \cdot 365.25); \tag{A2}$$

**A.2 Surface shelf sea (ss)**

$$V^{ss} = \Delta Z_{eu} \cdot A_{ocean} \cdot \mathcal{P}_{shelf} \tag{A3}$$

$$Prod^{ss} = P_{eff} \cdot (P^{ss}/(P^{ss} + K_P)) \cdot P^{ss} \tag{A4}$$

$$Flx\_SP_{org}^{ss} = (Prod^{ss} - cg_f \cdot Prod^{ss}) \cdot \Delta Z_{eu} \tag{A5}$$

$$Flx\_LP_{org}^{ss} = cg_f \cdot Prod^{ss} \cdot \Delta Z_{eu} \tag{A6}$$

$$VExp\_SP_{org}^{ss} = Flx\_SP_{org}^{ss} \cdot exp(-(\Delta Z_{eu}/2)/z_{rem}^S) \tag{A7}$$

$$VExp\_LP_{org}^{ss} = Flx\_SP_{org}^{ss} \cdot exp(-(\Delta Z_{eu}/2)/z_{rem}^L) \tag{A8}$$

$$VExp\_TOTbyV^{ss} = \left(Flx\_SP_{org}^{ss} + Flx\_SP_{org}^{ss}\right)/\Delta Z_{eu} \tag{A9}$$

$$\frac{dP^{ss}}{dt} = P_{in} \cdot \left(1 - \mathcal{P}_{open}\right)/V^{ss} + (Upw \cdot (P^{ds} - P^{ss}) + Mix_{ls} \cdot (P^{so} - P^{ss}) + Mix_{vs} \cdot (P^{ds} - P^{ss})) \cdot spy/V^{ss}$$
$$- VExp\_TOTbyV^{ss}$$

$$\tag{A10}$$

$$AirSea^{ss} = K_W \cdot (O^{at} \cdot p_{at}/K_{Henri} - O^{ss}) \cdot (A_{ocean} \cdot \mathcal{P}_{shelf})/V^{ss} \tag{A11}$$

$$OProd^{ss} = OP_{Red} \cdot VExp\_TOTbyV^{ss} \tag{A12}$$

$$\frac{dO^{ss}}{dt} = \left(Upw \cdot (O^{ds} - O^{ss}) + Mix_{ls} \cdot (O^{so} - O^{ss}) + Mix_{vs} \cdot (O^{ds} - O^{ss})\right) \cdot spy/V^{ss} + AirSea^{ss}$$
$$+ OProd^{ss}$$

$$\tag{A13}$$

**A.3 Deep shelf sea (ds)**

$$V^{ds} = \Delta Z_{ds} \cdot A_{ocean} \cdot \mathcal{P}_{shelf} \tag{A14}$$

$$VIn\_SP_{org}{}^{ds} = VExp\_SP_{org}{}^{ss} \tag{A15}$$

$$VIn\_LP_{org}{}^{ds} = VExp\_LP_{org}{}^{ss} \tag{A16}$$

$$Rem\_SP_{org}{}^{ds} = VIn\_SP_{org}{}^{ds} \cdot (1 - exp(-\Delta Z_{ds}/z_{rem}^S))/\Delta Z_{ds} \tag{A17}$$

$$Rem\_LP_{org}{}^{ds} = VIn\_LP_{org}{}^{ds} \cdot (1 - exp(-\Delta Z_{ds}/z_{rem}^L))/\Delta Z_{ds} \tag{A18}$$

$$AerRem\_SedP_{org}{}^{ds} = rm_r \cdot SedP_{org}{}^s/\Delta Z_{ds} \cdot (O^{ds}/(O^{ds} + K_O^s)) \tag{A19}$$

$$AnaRem\_SedP_{org}{}^{ds} = (rm_r \cdot fe_{an}) \cdot SedP_{org}{}^s/\Delta Z_{ds} \cdot (1 - O^{ds}/(O^{ds} + K_O^s)) \tag{A20}$$

$$\frac{dP^{ds}}{dt} = (Upw \cdot (P^{do} - P^{ds}) + Mix_{ld} \cdot (P^{do} - P^{ds}) + Mix_{vs} \cdot (P^{ss} - P^{ds})) \cdot spy/V^{ds}$$
$$+ \left(Rem\_SP_{org}{}^{ds} + Rem\_LP_{org}{}^{ds} + AerRem\_SedP_{org}{}^{ds} + AnaRem\_SedP_{org}{}^{ds}\right)$$

$$\tag{A21}$$

$$AerRemWcO^{ds} = OP_{Red} \cdot (Rem\_SP_{org}{}^{ds} + Rem\_LP_{org}{}^{ds}) \cdot (O^{ds}/(O^{ds} + K_O^w)) \tag{A22}$$

$$AerRemSedO^{ds} = OP_{Red} \cdot AerRem\_SedP_{org}{}^{ds} \tag{A23}$$

$$\frac{dO^{ds}}{dt} = (Upw \cdot (O^{do} - O^{ds}) + Mix_{ld} \cdot (O^{do} - O^{ds}) + Mix_{vs} \cdot (O^{ss} - O^{ds})) \cdot spy/V^{ds} - AerRemWcO^{ds}$$
$$- AerRemSedO^{ds}$$

$$\tag{A24}$$

**A.4 Surface open ocean (so)**

$$V^{so} = \Delta Z_{eu} \cdot A_{ocean} \cdot (1 - \mathcal{P}_{shelf}) \tag{A25}$$

$$Prod^{so} = P_{eff} \cdot (P^{so}/(P^{so} + K_P)) \cdot P^{so} \tag{A26}$$

$$Flx\_SP_{org}{}^{so} = (Prod^{so} - cg_f \cdot Prod^{so}) \cdot \Delta Z_{eu} \tag{A27}$$

$$Flx\_LP_{org}{}^{so} = cg_f \cdot Prod^{so} \cdot \Delta Z_{eu} \tag{A28}$$

$$VExp\_SP_{org}{}^{so} = Flx\_SP_{org}{}^{so} \cdot exp(-(\Delta Z_{eu}/2)/z_{rem}^S) \tag{A29}$$

$$VExp\_LP_{org}{}^{so} = Flx\_LP_{org}{}^{so} \cdot exp(-(\Delta Z_{eu}/2)/z_{rem}^L) \tag{A30}$$

$$VExp\_TOTbyV^{so} = \left(Flx\_SP_{org}{}^{so} + Flx\_SP_{org}{}^{so}\right)/\Delta Z_{eu} \tag{A31}$$

$$\frac{dP^{so}}{dt} = P_{in} \cdot \mathcal{P}_{open}/V^{ss} + (Upw \cdot (P^{ss} - P^{so}) + Mix_{ls} \cdot (P^{ss} - P^{so}) + Mix_{vo} \cdot (P^{do} - P^{so})) \cdot spy/V^{so}$$
$$- VExp\_TOTbyV^{so}$$

(A32)

$$AirSea^{so} = K_W \cdot (O^{at} \cdot p_{at}/K_{Henri} - O^{so}) \cdot (A_{ocean} \cdot (1 - \mathcal{P}_{shelf}))/V^{so} \tag{A33}$$

$$OProd^{so} = OP_{Red} \cdot VExp\_TOTbyV^{so} \tag{A34}$$

$$\frac{dO^{so}}{dt} = \left(Upw \cdot (O^{ss} - O^{so}) + Mix_{ls} \cdot (O^{ss} - O^{so}) + Mix_{vo} \cdot (O^{do} - O^{so})\right) \cdot spy/V^{so} + AirSea^{so}$$
$$+ OProd^{so}$$

(A35)

### A.5 Deep open ocean (do)

$$V^{do} = \Delta Z_{do} \cdot A_{ocean} \cdot (1 - \mathcal{P}_{shelf}) \tag{A36}$$

$$VIn\_SP_{org}{}^{do} = VExp\_SP_{org}{}^{so} \tag{A37}$$

$$VIn\_LP_{org}{}^{do} = VExp\_LP_{org}{}^{so} \tag{A38}$$

$$Rem\_SP_{org}{}^{do} = VIn\_SP_{org}{}^{do} \cdot (1 - exp(-\Delta Z_{do}/z_{rem}^S))/\Delta Z_{do} \tag{A39}$$

$$Rem\_LP_{org}{}^{do} = VIn\_LP_{org}{}^{do} \cdot (1 - exp(-\Delta Z_{do}/z_{rem}^L))/\Delta Z_{do} \tag{A40}$$

$$AerRem\_SedP_{org}{}^{do} = rm_r \cdot SedP_{org}{}^o/\Delta Z_{do} \cdot (O^{do}/(O^{do} + K_O^s)) \tag{A41}$$

$$AnaRem\_SedP_{org}{}^{do} = (rm_r \cdot fe_{an}) \cdot SedP_{org}{}^o/\Delta Z_{do} \cdot (1 - O^{do}/(O^{do} + K_O^s)) \tag{A42}$$

$$\frac{dP^{do}}{dt} = (Upw \cdot (P^{so} - P^{do}) + Mix_{ld} \cdot (P^{ds} - P^{do}) + Mix_{vo} \cdot (P^{so} - P^{do})) \cdot spy/V^{do}$$
$$+ \left(Rem\_SP_{org}{}^{do} + Rem\_LP_{org}{}^{do} + AerRem\_SedP_{org}{}^{do} + AnaRem\_SedP_{org}{}^{do}\right)$$

(A43)

$$AerRemWcO^{do} = OP_{Red} \cdot (Rem\_SP_{org}{}^{do} + Rem\_LP_{org}{}^{do}) \cdot (O^{do}/(O^{do} + K_O^w)) \tag{A44}$$

$$AerRemSedO^{do} = OP_{Red} \cdot AerRem\_SedP_{org}{}^{do} \tag{A45}$$

$$\frac{dO^{do}}{dt} = (Upw \cdot (O^{so} - O^{do}) + Mix_{ld} \cdot (O^{ds} - O^{do}) + Mix_{vo} \cdot (O^{so} - O^{do})) \cdot spy/V^{do} - AerRemWcO^{do}$$
$$- AerRemSedO^{do}$$

(A46)

### A.6 Shelf sea sediments (s)

$$SedFlx^s = VIn\_SP_{org}{}^{ds} \cdot exp(-\Delta Z_{ds}/z_{rem}^S) + VIn\_LP_{org}{}^{ds} \cdot exp(-\Delta Z_{ds}/z_{rem}^L) \tag{A47}$$

$$CaPform^s = CaP_r \cdot (SedP_{org}{}^s)^2 \cdot (O^{ds}/(O^{ds} + K_O^w) + fs_{an} \cdot (1 - O^{ds}/(O^{ds} + K_O^w))) \tag{A48}$$

$$Rem\_SedP_{org}{}^{ds} = AerRem\_SedP_{org}{}^{ds} + AnaRem\_SedP_{org}{}^{ds} \tag{A49}$$

$$\frac{dSedP_{org}{}^s}{dt} = SedFlx^s - CaPform^s - Rem\_SedP_{org}{}^{ds} \cdot \Delta Z_{ds} \tag{A50}$$

### A.7 Open ocean sediments (o)

$$SedFlx^o = VIn\_SP_{org}{}^{do} \cdot exp(-\Delta Z_{do}/z_{rem}^S) + VIn\_LP_{org}{}^{do} \cdot exp(-\Delta Z_{do}/z_{rem}^L) \tag{A51}$$

$$CaPform^o = CaP_r \cdot (SedP_{org}{}^o)^2 \cdot (O^{do}/(O^{do} + K_O^w) + fs_{an} \cdot (1 - O^{do}/(O^{do} + K_O^w))) \tag{A52}$$

$$Rem\_SedP_{org}{}^{do} = AerRem\_SedP_{org}{}^{do} + AnaRem\_SedP_{org}{}^{do} \tag{A53}$$

$$\frac{dSedP_{org}{}^o}{dt} = SedFlx^o - CaPform^o - Rem\_SedP_{org}{}^{do} \cdot \Delta Z_{do} \tag{A54}$$

### A.8 Atmosphere (at)

$$AirSea^{at} = (AirSea^{ss} + AirSea^{so})/(Mol_{atmo} \cdot 10^3) \tag{A55}$$

$$AnaRemWc^{ds} = OP_{Red} \cdot (Rem\_SP_{org}{}^{ds} + Rem\_LP_{org}{}^{ds}) \cdot (1 - O^{ds}/(O^{ds} + K_O^w)) \tag{A56}$$

$$AnaRemWc^{do} = OP_{Red} \cdot (Rem\_SP_{org}{}^{do} + Rem\_LP_{org}{}^{do}) \cdot (1 - O^{do}/(O^{do} + K_O^w)) \tag{A57}$$

$$AnaRemWc^{at} = (AnaRemWc^{ds} \cdot V^{ds} + AnaRemWc^{do} \cdot V^{do})/(Mol_{atmo} \cdot 10^3) \tag{A58}$$

$$AnaRemSed^{at} = (AnaRem\_SedP_{org}{}^{ds} \cdot V^{ds} + AnaRem\_SedP_{org}{}^{do} \cdot V^{do})/Mol_{atmo} \tag{A59}$$

$$OxyWeath = W_0 \cdot \sqrt{O^{at}/Omix_0}/Mol_{atmo} \tag{A60}$$

$$\frac{dO^{at}}{dt} = -AirSea^{at} - AnaRemWc^{at} - OxyWeath \tag{A61}$$

## A.9 Diagnostics: Total budgets of P and O

$$P_{sources} = P_{in} \cdot 10^{-15} \tag{A62}$$

$$P_{sinks} = \left(CaPform^s \cdot A_{ocean} \cdot \mathcal{P}_{shelf} + CaPform^o \cdot A_{ocean} \cdot \left(1 - \mathcal{P}_{shelf}\right)\right) \cdot 10^{-15} \tag{A63}$$

$$\frac{dP^{TOT}}{dt} = P_{sources} - P_{sinks}$$

$$\tag{A64}$$

$$O_{sources} = (OProd^{ss} \cdot V^{ss} + OProd^{so} \cdot V^{so}) \cdot 10^{-18} \tag{A65}$$

$$O_{sinks} = \left((SedRemO^{ds} + AerRem^{ds}) \cdot V^{ds} + (SedRemO^{do} + AerRem^{do}) \cdot V^{do} + OxyWeath \cdot Mol_{atmo}\right) \cdot$$
$$10^{-18} \tag{A66}$$

$$\frac{dO^{TOT}}{dt} = O_{sources} - O_{sinks}$$

$$\tag{A67}$$

35

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
