# Peer review of "BPOP-v1 model: exploring the impact of changes in the biological pump on the shelf sea and ocean nutrient and redox state"

_Geoscientific Model Development, 2019_

## Referee Comment (RC1) · Anonymous Referee #1 · 26 Aug 2019

**Review of BPOP-v1 model: exploring the impact of changes in the biological pump on the shelf sea and ocean nutrient and redox state (Lovecchio & Lenton)**

The authors present a new box model for the simulation of the marine oxygen and phosphorus cycles on geological time scales. They extend previous box models by introducing two different types of particulate organic matter (small and large) characterized by different sinking speeds. The model results presented by the authors confirm that the depth of the oxycline is to a large degree controlled by the remineralization length (size/settling velocity) of the sinking particles.

My major comments are related to the benthic model employed by the authors

- The model for phosphorus (P) degradation in marine sediments considers aerobic respiration (Eq. 11) but seems to ignore anaerobic degradation. As a result the burial efficiency increases when oxygen is deleted in ambient bottom waters whereas the available observations show that P burial efficiency actually decreases under low-oxygen conditions (Slomp et al., 2002; Van Cappellen and Ingall, 1994; Wallmann, 2010). The authors should try to change their benthic model (i.e. include anaerobic degradation and enhanced P release under anoxia) or explain why they apparently ignore the strong evidence for enhanced benthic P release under low oxygen conditions.
- The shelf model ignores P burial in shallow-water shelf sediments even though observations in the modern ocean indicate that most burial of particulate organic matter (POM) occurs in the inner shelf region at <50 m water depth (Dunne et al., 2007). The authors should try to change their benthic model to include shallow shelf burial or explain why they ignore burial in shallow shelf regions.
- Small (slowly sinking) particles are mostly degraded in the water column whereas a substantial fraction of the large (rapidly sinking) particles is not degraded but deposited at the seafloor. Consequently, large POM particles reaching the seabed are more reactive (fresher) than small (older) particles and the kinetic constant for benthic degradation should increase with increasing particle size (Stolpovsky et al., 2018). Since particle size (sinking speed, mineralization length) is the major parameter varied in the modeling, the authors should try to consider this effect in their benthic model.

Considering these model limitations, I do not know whether the authors' conclusion: "shelf ocean anoxia can coexist with an oxygenated deep ocean" (abstract, line 19) is really valid. Moreover, this conclusion depends on the model assumption that deep water formation takes place in the open ocean. This assumption is questionable since much of the modern deep water formation happens at continental margins. If these margin sites are oxygen depleted the resulting deep water would also be oxygen depleted. The authors should discuss this possibility and critically assess the validity of their model assumption.

**References**

Dunne, J.P., Sarmiento, J.L., Gnanadesikan, A., 2007. A synthesis of global particle export from the surface ocean and cycling through the ocean interior and on the seafloor. Global Biogeochemical Cycles 21, 16.

Slomp, C.P., Thompson, J., De Lange, G., 2002. Enhanced regeneration of phosphorus during formation of the most recent eastern Mediterranean sapropel (S1). Geochimica et Cosmochimica Acta 66, 1171-1184.

Stolpovsky, K., Dale, A.W., Wallmann, K., 2018. A new look at the multi-G model for organic carbon degradation in surface marine sediments for coupled benthic-pelagic simulations of the global ocean. Biogeosciences 15, 3391-3407.

Van Cappellen, P., Ingall, E.D., 1994. Benthic phosphorus regeneration, net primary production, and ocean anoxia: A model of the coupled marine biogeochemical cycles of carbon and phosphorus. Paleoceanography 9, 677-692.

Wallmann, K., 2010. Phosphorus imbalance in the global ocean? Global Biogeochemical Cycles 24, GB4030.

---

## Referee Comment (RC2) · Anonymous Referee #2 · 7 Oct 2019

This paper describes a new box model of phosphorus and oxygen cycling in the ocean and its sediments, with the oxygen in the ocean coupled to oxygen in the atmosphere through air-sea gas exchange. The model resolves shelf and open ocean environments separately. A novel aspect of the model is the separation of the particle export flux into export by small and export by large particles. This separation makes the model a potentially useful tool for investigating the impacts of the evolution of eukaryotic phytoplankton, which generally produce larger particles which remineralise deeper in the ocean on average. The transition back in geological time from an ocean dominated by

the microbial loop (intense recycling) to an ocean with more efficient export can thus be studied.

This paper is well written and the motivation for its construction is clearly articulated in the introduction to the paper. The scientific rationale for this study is convincing and the overall design of the model is appropriate for the scientific questions that it hopes to address. However, there appear to be some deficiencies in the mechanics of the model such that it is not at all clear that the model, as presently formulated, is working correctly. This review therefore concentrates on providing comments on the details of the model and its equations, because it is premature to give detailed consideration of the model results at a time when the model needs further development.

It is stated on line 10 of page 4 that any organic matter that does not reach the sediments is instantaneously remineralised. This decision seems quite reasonable, but it is contradicted by the equations, in which organic matter is not only remineralised but is also advected and mixed. I recommend to do either one thing or the other but not both. If suspended particles are going to be mixed around in the model then they should have their own separate ordinary differential equations and state variables. Alternatively, if particle flux and remineralisation are made instantaneous, as stated in the manuscript, then there should be no mixing or advection of particulate organic matter. Whichever way it is done, the descriptions in the text need to be made consistent with the equations.

Tables 1 to 3 are very helpful. Another table needs to be added, listing the state variables in the model and stating their units. In addition, the units of all equations (the left-hand side) should also be stated, if they are not already given in the tables.

Too many of the equations in the model are dimensionally inconsistent. That is to say, the units on the left-hand side of the equation do not match the units on the right-hand side of the equation when the different terms are combined together. As an example, equation 4 on page 4 is an equation for the rate of organic matter production in units of

phosphorus. This is a flux (rate of transfer), and therefore has to be in units of Moles y-1 or mmol m-3 y-1 or similar. Because this is an ongoing flux rather than a one-off transfer, it must be expressed as a rate of transfer per unit time. However, none of the terms on the right-hand side of the equation have time anywhere in their units. The equation is formulated in such a way that it appears to be aimed at converting a fraction of the surface phosphorus concentration into production at each timestep, but the way it is actually formulated means that the rate of conversion of surface phosphate into production (organic matter) will depend on the timestep used. Shorter time steps will convert phosphate to organic matter more rapidly than longer timesteps whereas ODE equations should be timestep-independent. Equation 5 is another example, where, according to the equation, 'Coag' must have units of organic matter concentration squared per year, which makes no sense. Before resubmission, I recommend that every equation in the model is checked for dimensional (units) consistency: multiplying through the units of the terms on the right-hand side should produce the units of the terms on the left-hand side.

I did not notice a statement anywhere that conservation of mass (or, more properly, conservation of total inventories of elements) has been checked and found to be stable. This is easy to do for a box model, whether it is closed or open. Obviously for a closed model, if there are no errors in the equations, the total sum of atoms of a given element should be constant over time. For an open model, the changes in the total inventory over time should exactly match the sum of external inputs over time minus the sum of the outputs from the system over time (i.e.  $\Delta Inv= -$ ). This can easily be checked by adding two extra differential equations to the model: one to track the sum of the system and another to keep track of the sum of the losses from the system as a whole.

From the model equations, I suspect that the model does not properly conserve phosphorus and oxygen but rather there is some (unintended) cumulative creation/destruction over time. This will interfere with the ability of the model to be run

СЗ

over long timescales to address the geological timescale questions of interest. It is a little bit unclear, but it appears that the amount of phosphate removed per unit time from the surface box as particle export is not identical to the amounts of phosphate added per unit time to the deep box and sediments combined. The euphotic zone depth appears in the equation for the former but not the latter, for instance, whereas if it appears in one then it should also appear in the other. Again, checks can be made by adding extra (book-keeping) ODEs to the model. For this example, one extra ODE could tally up the cumulative export from the surface box and another extra ODE could tally up the sum of the cumulative inputs to the deep and sediment boxes. At the end of each model run, a quick numerical check can be made to ensure that the tallies are identical within the precision of numerical rounding errors.

These checks should be made before the manuscript is resubmitted, and a statement added to the manuscript to confirm that mass balance checks have been made, with satisfactory results.

Specific comments:

Key paper not cited: Reinhard CT., Planavsky NJ et al. "Evolution of the global phosphorus cycle." Nature 541, no. 7637 (2017): 386.

Equations 14 & 17: AirSea should appear the same in both

Equation 17: why does anaerobic remineralisation remove oxygen?

Line 5 of page 7: anaerobic remineralisation of organic matter also releases phosphorus (in fact oxygen-depleted sediments are stronger sources of phosphorus to the overlying water column).

Table 1: moles of air or moles of oxygen in the atmospheric box?

Table 2: the Redfield ratio of oxygen to phosphorus (-O2:P) is  $\sim$ 150:1 not 106:1 (see for instance: Anderson, L.A. and Sarmiento, J.L., 1994. Redfield ratios of remineralization determined by nutrient data analysis. Global biogeochemical cycles, 8(1),

pp.65-80; Thomas, H., 2002. Remineralization ratios of carbon, nutrients, and oxygen in the North Atlantic Ocean: A field databased assessment. Global biogeochemical cycles, 16(3)). W0 is a baseline flux (line 3 of page 4), hence cannot have units of mmol if equation 3 is to be dimensionally plausible.

---

## Author Comment (AC3) · 7 Nov 2019

Author's response

**"BPOP-v1 model: exploring the impact of changes in the biological pump on the shelf sea and ocean nutrient and redox state"**

Elisa Lovecchio[1] and Timothy M. Lenton[1]
[1]Global Systems Institute, University of Exeter, Exeter, EX4 4QE, United Kingdom

Dear Editor,

Thank you very much for taking our manuscript into consideration for publication on GMD.
We have carefully revised our manuscript following your and the Referees' suggestions, and include our extended answers in the following pages. We summarize here our main changes.

In line with the suggestions of Referee nr.1, we have modified our benthic model including anaerobic remineralization and a modulation of remineralization and burial rates as a function of oxygen. Following the suggestion of Referee nr.2, we have switched off the advection of Porg in the model. We have also included two further diagnostic variables, to easily check mass conservation. Overall, none of the modifications to the model has drastically changed the model's behaviour, confirming the robustness of our results and conclusions. We have also run a sensitivity experiment, discussed in the answer to Referee nr.2.

We have carefully revised our manuscript according to the changes to the model, extended the model evaluation, and double-checked all of the equations and tables. We have added a figure showing the time evolution of the variables and added two further tables: one listing the variables and their dimensions and one summarizing the evaluation.

We include below our extended answers to all of the comments and questions, followed by a version of the manuscript with track changes.

We look forward to your response.
Sincerely,
Elisa Lovecchio

**Answer to Editor's Comments**

We thank the Editor for his comments and suggestions, which we copy below (in blue), and explain in detail our related changes to the manuscript (in red) and model.

*EC 1) Evaluation of your model (page 7) might benefit from a table which compares each of the quantities to the corresponding observational estimate (e.g. see Table 2 of Yool & Tyrrell, GBC, 2002)*

In agreement with this suggestion, we have added an evaluation table (**Table 5** of the revised manuscript) with the updated results of our model evaluation after revisions. We have updated our comments on the model evaluation in the related section 3 of the manuscript.

The table reads as follows:

| Quantity | Model | Modern values or estimates | Units | Source |
|---|---|---|---|---|
| Total ocean P | 3200 - 3400 | 3100 | TmolP | Watson et al. (2017) |
| Total ocean $O_2$ | 100 - 150 | 220 (deep ocean) | $PmolO_2$ | Slomp and Van Cappellen (2006) |
| $P^{ss}$ | 1.5 – 1.8 | 1 – 1.5 | mmol m$^{-3}$ | Garcia et al. (2018b);Sarmiento and Gruber (2006) |
| $P^{ds}$ | 4.5 – 6 | 2.2 | mmol m$^{-3}$ | Garcia et al. (2018b);Watson et al. (2017) |
| $P^{so}$ | 0.5 - 1 | 0.2 - 2 | mmol m$^{-3}$ | Garcia et al. (2018b);Sarmiento and Gruber (2006) |
| $P^{do}$ | 2.7 - 3 | 1 - 3 | mmol m$^{-3}$ | Garcia et al. (2018b);Sarmiento and Gruber (2006) |
| $O^{ss}$ | 273 | 200 - 350 | mmol m$^{-3}$ | Garcia et al. (2018a) |
| $O^{ds}$ | 4 - 20 | 0-80 | mmol m$^{-3}$ | Garcia et al. (2018a) |
| $O^{so}$ | 273 | 200 - 350 | mmol m$^{-3}$ | Garcia et al. (2018a) |
| $O^{do}$ | 75 - 120 | 40-200 | mmol m$^{-3}$ | Garcia et al. (2018a) |
| Production (Prod) | 1400 - 3000 | 3300 - 9000 | TmolC yr$^{-1}$ | Carr et al. (2006) |
| Export | 300 - 430 | 415 – 1660 | TmolC yr$^{-1}$ | Henson et al. (2011) |
| Export production | 10 % - 32 % | 2 % - 20 % | of total Prod | Boyd and Trull (2007) |
| Burial | 0.3 % - 0.7 % | 0.4 % | of total Prod | Sarmiento and Gruber (2006) |
| Shelf sea production | 12 % - 20 % | 20 % | of total Prod | Barrón and Duarte (2015);Wollast (1998) |
| Shelf sea export | 20 % - 23 % | 29 % | of total Export | Sarmiento and Gruber (2006) |
| Shelf sea burial | 100 % | 91 % | of total Burial | Sarmiento and Gruber (2006) |

**Table 5: Summary of the model evaluation provided in section 3. Modern observations and estimates are compared to model results obtained for $z_{rem}{}^{L}$ in the range of measured values for a modern shelf sea (Cavan et al., 2017).**

*EC 2) I'm a little surprised that you haven't tuned your productivity to more realistic values, and that you lay the blame for your model's low productivity on its implicit microbial loop and missing distinction of new / regenerated production; it should be possible to achieve more realistic levels of productivity through tuning, and other models (again, Yool & Tyrrell, 2002) have no such difficulties while having the same limitations*

Following the modifications to the model's equations in agreement with the suggestions of Referee nr.1 and Referee nr.2, we have further retuned the model towards more realistic values of production and export. This was done by slightly modifying a few biogeochemical parameters (coagulation rate, Ca-P formation rate) and one physical parameter (vertical mixing in the shelf) in the range of plausible values. Further changes to the model, following recent literature, consisted in distinguishing between a water column and a sediment half saturation constant for aerobic remineralization. The combination of these changes in the equations and in a few parameter values allowed us to increase production and export (bringing them closer to the lower end of modern estimates), while still maintaining the values of export production and burial to export ratio in the range of current estimates. We modified the discussion of the model evaluation (subsection 3.2) accordingly.

*EC 3&4) Although you mention the timescales over which you model can easily be run (as well as alluding to simulating it across the Phanerozoic), none of your figures show the time evolution of your model; it might be informative for readers were you to include some indication of the model's time-evolution from, say, the arbitrary initial conditions that you list in Table 2 (e.g. see Figure 5 of Yool & Tyrrell, 2002)*
*On a related point, how does your model compare in terms of residence times of the elements it represents?; given that it includes river inputs of P and reasonably represents ocean concentrations, I would expect it to do a good job here, but it might still be worth drawing attention to this*

To address this comment, we have included in the manuscript an additional figure (revised paper's **Figure 3**) to show an example of modelled tracer evolution. Furthermore, we have divided the Evaluation section in two parts with an initial subsection 3.1 focusing on the model dynamics and a second subsection 3.2 focusing on the comparison to model ocean budgets and fluxes (which corresponds to the old Evaluation before revisions).

Subsection 3.1 of the revised manuscript reads as follows:

[revised manuscript text omitted]
~~However, the model allows to include lateral advective and vertical mixing fluxes of organic phosphorus by setting to one the value of the parameter , which acts as a switch in the equations. This may be necessary when working with suspended $SP_{org}$ pools ($z_{rem}^{S} = 0$), and therefore that modelled $SP_{org}$ and $LP_{org}$ can only be transported by physical fluxes from regions of production to regions of remineralization. For this reason, physical fluxes affect the two organic matter~~

10  , as  these physical fluxes  is essential to account for  vertical  export by  downwelling and mixing (Stukel and Ducklow, 2017), and the lateral organic matter redistribution  (Lovecchio et al., 2017;Inthorn et al., 2006) may be important when working with suspended $SP_{org}$ ($z_{rem}^{S} = 0$), these fluxes are not currently accounted for in

15  the model. ~~When advected and mixed, implicitly modelled $SP_{org}$ and $LP_{org}$ can only be transported by physical fluxes from regions of production to regions of remineralization. For this reason, if included, these fluxes affect the two organic matter species only in a single direction (Figure 1b). Due to the wide extension of the modelled ocean boxes, lateral fluxes are assumed to only affect $SP_{org}$. The lateral export of $SP_{org}$ reduces its availability for export and burial in the shelf sea, i.e., at each time step lateral fluxes out of the shelf happen before the $SP_{org}$~~

[revised manuscript text omitted]

$$\cancel{LatExp^{ss} = SP_{org}{}^{ss} \cdot (Upw + Mix_{ls})/V^{ss}} \tag{A7}$$

$$VExp\_SP_{org}{}^{ss} = (SP_{org}{}^{ss} \cancel{- LatExp^{ss}}) \cdot (exp(-(\Delta Z_{eu}/2) /z_{rem}^S) \cancel{+ Mix_{vs}/V^{ss}})$$
$$\text{(A}\underline{8}\cancel{7}\text{)}$$

$$VExp\_LP_{org}{}^{ss} = LP_{org}{}^{ss} \cdot (exp(-(\Delta Z_{eu}/2) /z_{rem}^L) \cancel{+ Mix_{vs}/V^{ss}})$$
$$\text{(A}\underline{9}\cancel{8}\text{)}$$

$$\frac{dP^{ss}}{dt} = P_{in} \cdot \left(1 - \mathcal{P}_{open}\right)/V^{ss} + \left(Upw \cdot \left(P^{ds} - P^{ss}\right) + Mix_{ls} \cdot \left(P^{so} - P^{ss}\right) + Mix_{vs} \cdot \left(P^{ds} - P^{ss}\right)\right) \cdot spy/V^{ss}$$
$$- \left(\text{VExp}_{\text{SP}_{org}}{}^{ss} + \text{VExp}_{\text{LP}_{org}}{}^{ss}\right)$$

(A9̶1̶0̶)

$$AirSea^{ss} = K_W \cdot \left(O^{at}/K_{Henri} - O^{ss}\right) \cdot \left(A_{ocean} \cdot \mathcal{P}_{shelf}\right)/V^{ss} \tag{A10̶1̶}$$

$$OProd^{ss} = OP_{Red} \cdot \left(VExp\_SP_{org}{}^{ss} + VExp\_LP_{org}{}^{ss}\right) \tag{A11̶2̶}$$

$$\frac{dO^{ss}}{dt} = \left(Upw \cdot \left(O^{ds} - O^{ss}\right) + \text{Mix}_{ls} \cdot \left(O^{so} - O^{ss}\right) + \text{Mix}_{vs} \cdot \left(O^{ds} - O^{ss}\right)\right) \cdot spy/V^{ss} + AirSea^{ss}$$
$$+ OProd^{ss}$$

(A12̶3̶)

**A.3 Deep shelf sea (ds)**

$$V^{ds} = \Delta Z_{ds} \cdot A_{ocean} \cdot \mathcal{P}_{shelf}$$

(A13̶4̶)

$$VInp\_SP_{org}{}^{ds} = VExp\_SP_{org}{}^{ss} \cdot \left(V^{ss}/V^{ds}\right)$$

20 (A14̶5̶)

$$SP_{org}{}^{ds} = VInp\_SP_{org}{}^{ds} - cg_r \cdot \left(VInp\_SP_{org}{}^{ds}\right)^2$$

(A15̶6̶)

$$LP_{org}{}^{ds} = VExp\_LP_{org}{}^{ss} \cdot \left(V^{ss}/V^{ds}\right) + cg_r \cdot \left(VInp\_SP_{org}{}^{ds}\right)^2$$

(A16̶7̶)

25 $$\cancel{LatExp^{ds} = SP_{org}{}^{ds} \cdot Mix_{la}/V^{ds}} \tag{A18}$$

$$Rem\_SP_{org}{}^{ds} = \left(SP_{org}{}^{ds} \cancel{- LatExp^{ds}}\right) \cdot \left(1 - exp\left(-\Delta Z_{ds}/z_{rem}^S\right)\right)$$

(A17̶9̶)

$$Rem\_LP_{org}{}^{ds} = LP_{org}{}^{ds} \cdot \left(1 - exp\left(-\Delta Z_{ds}/z_{rem}^L\right)\right) \tag{A18̶2̶0̶}$$

$$AerRem\_SedP_{org}{}^{ds} = rm_r \cdot SedP_{org}{}^{s}/\Delta Z_{ds} \cdot \left(O^{ds}/\left(O^{ds} + K_O^{\cancel{s}K_U}\right)\right)$$

30 (A19̶2̶1̶)

$$AnaRem\_SedP_{org}{}^{ds} = (rm_r \cdot fe_{an}) \cdot SedP_{org}{}^{s}/\Delta Z_{ds} \cdot (1 - O^{ds}/(O^{ds} + K_O^{s})) \underline{\phantom{xxxxxxxxxxxxx}} \quad (A20)$$

$$\frac{dP^{ds}}{dt} = (Upw \cdot (P^{do} - P^{ds}) + Mix_{ld} \cdot (P^{do} - P^{ds}) + Mix_{vs} \cdot (P^{ss} - P^{ds})) \cdot spy/V^{ds}$$

$$+\!\!+ \left(Rem\_SP_{org}{}^{ds} + Rem\_LP_{org}{}^{ds} + AerRem\_SedP_{org}{}^{ds} + AnaRem\_SedP_{org}{}^{ds}\right)$$

$$\underline{\phantom{xxx} + \left(\cancel{Rem\_SP_{org}{}^{ds} + Rem\_LP_{org}{}^{ds} + Rem\_SedP_{org}{}^{ds}}\right)}$$

(A21<s>2</s>)

$$AerRem\underline{Wc}O^{ds} = OP_{Red} \cdot (Rem\_SP_{org}{}^{ds} + Rem\_LP_{org}{}^{ds}) \cdot (O^{ds}/(O^{ds} + K_O^{w}\cancel{K_O}))$$

(A22<s>3</s>)

$$Aer\cancel{Sed}Rem\underline{Sed}O^{ds} = OP_{Red} \cdot AerRem\_SedP_{org}{}^{ds} \quad\quad\quad (A2\underline{3}\cancel{4})$$

$$\frac{dO^{ds}}{dt} = (Upw \cdot (O^{do} - O^{ds}) + Mix_{ld} \cdot (O^{do} - O^{ds}) + Mix_{vs} \cdot (O^{ss} - O^{ds})) \cdot spy/V^{ds} - AerRem\underline{Wc}O^{ds}$$

$$- Aer\cancel{Sed}Rem\underline{Sed}O^{ds}$$

(A24<s>5</s>)

**A.4 Surface open ocean (so)**

$$V^{so} = \Delta Z_{eu} \cdot A_{ocean} \cdot (1 - \mathcal{P}_{shelf})$$

(<s>A26</s>A25)

$$Prod^{so} = P_{eff} \cdot (P^{so}/(P^{so} + K_P)) \cdot P^{so}$$

(A26<s>7</s>)

$$\cancel{LatInp^{so} = SP_{org}{}^{ss} \cdot (Upw + Mix_{ls})/V^{so}} \quad\quad\quad\quad\quad\quad (A28)$$

$$SP_{org}{}^{so} = \left(Prod^{so} \cancel{+ LatInp^{so}}\right) - cg_r \cdot \left(\left(Prod^{so}\right) \cancel{+ LatInp^{so}}\right)^2$$

$$\quad\quad (A\underline{27}\cancel{29})$$

$$LP_{org}{}^{so} = cg_r \cdot \left(Prod^{so} \cancel{+ LatInp^{so}}\right)^2$$

(A28<s>30</s>)

$$VExp\_SP_{org}{}^{so} = SP_{org}{}^{so} \cdot \left(exp(-(\Delta Z_{eu}/2)/z_{rem}^{S}) \cancel{+ Mix_{vo}/V^{so}}\right)$$

(A29<s>31</s>)

$$VExp\_LP_{org}{}^{so} = LP_{org}{}^{so} \cdot \left(exp(-(\Delta Z_{eu}/2)/z_{rem}^{L}) \cancel{+ Mix_{vo}/V^{so}}\right)$$

(A30<s>2</s>)

$$\frac{dP^{so}}{dt} = P_{in} \cdot \mathcal{P}_{open}/V^{ss} + (Upw \cdot (P^{ss} - P^{so}) + Mix_{ls} \cdot (P^{ss} - P^{so}) + Mix_{vo} \cdot (P^{do} - P^{so})) \cdot spy/V^{so}$$
$$- \left(VExp\_SP_{org}^{\ so} + VExp\_LP_{org}^{\ so}\right) +$$

$$\cancel{\left(VExp\_SP_{org}^{\ so} + VExp\_LP_{org}^{\ so}\right)}$$

(A3$\underline{1}$$\cancel{3}$)

$$AirSea^{so} = K_W \cdot (O^{at}/K_{Henri} - O^{so}) \cdot (A_{ocean} \cdot (1 - \mathcal{P}_{shelf}))/V^{so}$$

(A3$\underline{2}$$\cancel{4}$)

$$OProd^{so} = OP_{Red} \cdot \left(VExp\_SP_{org}^{\ so} + VExp\_LP_{org}^{\ so}\right)$$

(A3$\underline{3}$$\cancel{5}$)

$$\frac{dO^{so}}{dt} = \left(Upw \cdot (O^{ss} - O^{so}) + Mix_{ls} \cdot (O^{ss} - O^{so}) + Mix_{vo} \cdot (O^{do} - O^{so})\right) \cdot spy/V^{so} + AirSea^{so}$$
$$+ OProd^{so}$$

(A3$\underline{4}$$\cancel{6}$)

**A.5 Deep open ocean (do)**

$$V^{do} = \Delta Z_{do} \cdot A_{ocean} \cdot (1 - \mathcal{P}_{shelf})$$

(A3$\underline{5}$$\cancel{7}$)

$$VInp\_SP_{org}^{\ do} = VExp\_SP_{org}^{\ so} \cdot (V^{so}/V^{do})$$

(A3$\underline{6}$$\cancel{8}$)

$$\cancel{LatInp^{do} = SP_{org}^{\ ds} \cdot Mix_{ta}/V^{do}} \hspace{4cm} \cancel{(A39)}$$

$$SP_{org}^{\ do} = (VInp\_SP_{org}^{\ do} \cancel{+ LatInp^{do}}) - cg_r \cdot (VInp\_SP_{org}^{\ do} \cancel{+ LatInp^{do}})^2$$

(A3$\underline{7}$$\cancel{40}$)

$$LP_{org}^{\ do} = VExp\_LP_{org}^{\ so} \cdot (V^{so}/V^{do}) + cg_r \cdot (VInp_{SP_{org}}^{\ do} \cancel{+ LatInp^{do}})^2$$

(A3$\underline{8}$$\cancel{41}$)

$$Rem\_SP_{org}^{\ do} = SP_{org}^{\ do} \cdot (1 - exp(-\Delta Z_{do}/z_{rem}^S)) \hspace{3cm} (A3\underline{9}\cancel{42})$$

$$Rem\_LP_{org}^{\ do} = LP_{org}^{\ do} \cdot (1 - exp(-\Delta Z_{do}/z_{rem}^L)) \hspace{3cm} (A4\underline{0}\cancel{3})$$

$$AerRem\_SedP_{org}^{\ do} = rm_r \cdot SedP_{org}^{\ o}/\Delta Z_{do} \cdot (O^{do}/(O^{do} + K_O^S \cancel{K_O})) $$

(A4$\underline{1}$$\cancel{4}$)

$$AnaRem\_SedP_{org}^{\ do} = (rm_r \cdot fe_{an}) \cdot SedP_{org}^{\ o}/\Delta Z_{do} \cdot (1 - O^{do}/(O^{do} + K_O^S)) \hspace{1.5cm} (A42)$$

$$\frac{dP^{do}}{dt} = (Upw \cdot (P^{so} - P^{do}) + Mix_{ld} \cdot (P^{ds} - P^{do}) + Mix_{vo} \cdot (P^{so} - P^{do})) \cdot spy/V^{do}$$
$$+ \left(Rem\_SP_{org}{}^{do} + Rem\_LP_{org}{}^{do} + AerRem\_SedP_{org}{}^{do} + AnaRem\_SedP_{org}{}^{do}\right) +$$
$$+ \left(Rem\_SP_{org}{}^{do} + Rem\_LP_{org}{}^{do} + Rem\_SedP_{org}{}^{do}\right)$$

(A43)

$$AerRem\underline{Wc}O^{do} = OP_{Red} \cdot (Rem\_SP_{org}{}^{do} + Rem\_LP_{org}{}^{do}) \cdot (O^{do}/(O^{do} + K_O^w \overline{K_o}))$$

(A44)

$$Aer\overline{Sed}Rem\underline{Sed}O^{do} = OP_{Red} \cdot AerRem\_SedP_{org}{}^{do} \qquad\qquad (A4\underline{5})$$

$$\frac{dO^{do}}{dt} = (Upw \cdot (O^{so} - O^{do}) + Mix_{ld} \cdot (O^{ds} - O^{do}) + Mix_{vo} \cdot (O^{so} - O^{do})) \cdot spy/V^{do} - AerRem\underline{Wc}O^{do}$$
$$- Aer\overline{Sed}Rem\underline{Sed}O^{do}$$

(A46)

**A.6 Shelf sea sediments (s)**

$$SedFlx^s = (\,(SP_{org}{}^{ds} \overline{- LatExp^{ds}}) \cdot exp(-\Delta Z_{ds}/z_{rem}^S) + LP_{org}{}^{ds} \cdot exp(-\Delta Z_{ds}/z_{rem}^L)\,) \cdot \Delta Z_{ds}$$

(A47)

$$CaPform^s = CaP_r \cdot (SedP_{org}{}^s)^2 \cdot (\,O^{ds}/(O^{ds} + K_O^w) + fs_{an} \cdot (1 - O^{ds}/(O^{ds} + K_O^w))\,)$$

(A48)

$$Rem\_SedP_{org}{}^{ds} = AerRem\_SedP_{org}{}^{ds} + AnaRem\_SedP_{org}{}^{ds} \qquad\qquad (A49)$$

$$\frac{dSedP_{org}{}^s}{dt} = Sed\overline{im}Flx^s - CaPform^s - Rem\_SedP_{org}{}^{ds} \cdot \Delta Z_{ds}$$

(A50)

25 ## A.7 Open ocean sediments (o)

$$SedFlx^o = (\,(SP_{org}{}^{do} \overline{- LatExp^{do}}) \cdot exp(-\Delta Z_{dos}/z_{rem}^S) + LP_{org}{}^{do} \cdot exp(-\Delta Z_{dos}/z_{rem}^L)\,) \cdot \Delta Z_{do}$$

(A51)

$$CaPform^o = CaP_r \cdot (SedP_{org}{}^o)^2 \cdot (\,O^{do}/(O^{do} + K_O^w) + fs_{an} \cdot (1 - O^{do}/(O^{do} + K_O^w))\,)$$

(A52)

$$Rem\_SedP_{org}{}^{do} = AerRem\_SedP_{org}{}^{do} + AnaRem\_SedP_{org}{}^{do} \qquad\qquad (A53)$$

$$\frac{dSedP_{org}{}^{o}}{dt} = Sed\underline{im}Flx^{o} - CaPform^{o} - Rem\_SedP_{org}{}^{do} \cdot \Delta Z_{do}$$

(A544)

5    **A.8 Atmosphere (at)**

$$AirSea^{at} = (AirSea^{ss} + AirSea^{so})/(Mol_{atmo} \cdot 10^{3})$$    (A555)

$$AnaRem\textcolor{red}{Wc}^{ds} = OP_{Red} \cdot (Rem\_SP_{org}{}^{ds} + Rem\_LP_{org}{}^{ds}) \cdot (1 - O^{ds}/(O^{ds} + K_{O}^{w}\cancel{K_{O}}))$$

(A5656)

$$AnaRem\textcolor{red}{Wc}^{do} = OP_{Red} \cdot (Rem\_SP_{org}{}^{do} + Rem\_LP_{org}{}^{do}) \cdot (1 - O^{do}/(O^{do} + K_{O}^{w}\cancel{K_{O}}))$$

10    (A5757)

$$AnaRem\textcolor{red}{Wc}^{at} = (AnaRem\textcolor{red}{Wc}^{ds} \cdot V^{ds} + AnaRem\textcolor{red}{Wc}^{do} \cdot V^{do})/(Mol_{atmo} \cdot 10^{3})$$

(A5858)

$$\textcolor{red}{AnaRemSed^{at} = (AnaRem\_SedP_{org}{}^{ds} \cdot V^{ds} + AnaRem\_SedP_{org}{}^{do} \cdot V^{do})/Mol_{atmo}} \quad \text{(A59)}$$

$$OxyWeath = W_{0} \cdot \sqrt{O^{at}/Omix_{0}} /\textcolor{red}{(}Mol_{atmo} \textcolor{red}{\cdot 10^{3})}$$    (A6059)

15    $$\frac{dO^{at}}{dt} = -AirSea^{at} - AnaRem\textcolor{red}{Wc}^{at} - OxyWeath$$

(A610)

[revised manuscript text omitted]

---

## Author Response (AR1)

**"BPOP-v1 model: exploring the impact of changes in the biological pump on the shelf sea and ocean nutrient and redox state"**

Elisa Lovecchio[1] and Timothy M. Lenton[1]
[1]Global Systems Institute, University of Exeter, Exeter, EX4 4QE, United Kingdom
* * *
Dear Editor,

Thank you very much for taking our manuscript into consideration for publication on GMD.
We have carefully revised our manuscript following your and the Referees' suggestions, and include our extended answers in the following pages. We summarize here our main changes.

In line with the suggestions of Referee nr.1, we have modified our benthic model including anaerobic remineralization and a modulation of remineralization and burial rates as a function of oxygen. Following the suggestion of Referee nr.2, we have switched off the advection of Porg in the model. We have also included two further diagnostic variables, to easily check mass conservation. Overall, none of the modifications to the model has drastically changed the model's behaviour, confirming the robustness of our results and conclusions. We have also run a sensitivity experiment, discussed in the answer to Referee nr.2.

We have carefully revised our manuscript according to the changes to the model, extended the model evaluation, and double-checked all of the equations and tables. We have added a figure showing the time evolution of the variables and added two further tables: one listing the variables and their dimensions and one summarizing the evaluation.

We include below our extended answers to all of the comments and questions, followed by a version of the manuscript with track changes.

We look forward to your response.
Sincerely,
Elisa Lovecchio

**Answer to Editor's Comments**

We thank the Editor for his comments and suggestions, which we copy below (in blue), and explain in detail our related changes to the manuscript (in red) and model.

*EC 1) Evaluation of your model (page 7) might benefit from a table which compares each of the quantities to the corresponding observational estimate (e.g. see Table 2 of Yool & Tyrrell, GBC, 2002)*

In agreement with this suggestion, we have added an evaluation table (**Table 5** of the revised manuscript) with the updated results of our model evaluation after revisions. We have updated our comments on the model evaluation in the related section 3.2 of the manuscript.

The table reads as follows:

| Quantity | Model | Modern values or estimates | Units | Source |
|---|---|---|---|---|
| Total ocean P | 3200 - 3400 | 3100 | TmolP | Watson et al. (2017) |
| Total ocean $O_2$ | 100 - 150 | 220 (deep ocean) | $PmolO_2$ | Slomp and Van Cappellen (2006) |
| $P^{ss}$ | 1.5 – 1.8 | 1 – 1.5 | mmol m$^{-3}$ | Garcia et al. (2018b);Sarmiento and Gruber (2006) |
| $P^{ds}$ | 4.5 – 6 | 2.2 | mmol m$^{-3}$ | Garcia et al. (2018b);Watson et al. (2017) |
| $P^{so}$ | 0.5 - 1 | 0.2 - 2 | mmol m$^{-3}$ | Garcia et al. (2018b);Sarmiento and Gruber (2006) |
| $P^{do}$ | 2.7 - 3 | 1 - 3 | mmol m$^{-3}$ | Garcia et al. (2018b);Sarmiento and Gruber (2006) |
| $O^{ss}$ | 273 | 200 - 350 | mmol m$^{-3}$ | Garcia et al. (2018a) |
| $O^{ds}$ | 4 - 20 | 0-80 | mmol m$^{-3}$ | Garcia et al. (2018a) |
| $O^{so}$ | 273 | 200 - 350 | mmol m$^{-3}$ | Garcia et al. (2018a) |
| $O^{do}$ | 75 - 120 | 40-200 | mmol m$^{-3}$ | Garcia et al. (2018a) |
| Production (Prod) | 1400 - 3000 | 3300 - 9000 | TmolC yr$^{-1}$ | Carr et al. (2006) |
| Export | 300 - 430 | 415 – 1660 | TmolC yr$^{-1}$ | Henson et al. (2011) |
| Export production | 10 % - 32 % | 2 % - 20 % | of total Prod | Boyd and Trull (2007) |
| Burial | 0.3 % - 0.7 % | 0.4 % | of total Prod | Sarmiento and Gruber (2006) |
| Shelf sea production | 12 % - 20 % | 20 % | of total Prod | Barrón and Duarte (2015);Wollast (1998) |
| Shelf sea export | 20 % - 23 % | 29 % | of total Export | Sarmiento and Gruber (2006) |
| Shelf sea burial | 100 % | 91 % | of total Burial | Sarmiento and Gruber (2006) |

Table 5: Summary of the model evaluation provided in section 3. Modern observations and estimates are compared to model results obtained for $z_{rem}{}^L$ in the range of measured values for a modern shelf sea (Cavan et al., 2017).

*EC 2) I'm a little surprised that you haven't tuned your productivity to more realistic values, and that you lay the blame for your model's low productivity on its implicit microbial loop and missing distinction of new / regenerated production; it should be possible to achieve more realistic levels of productivity through tuning, and other models (again, Yool & Tyrrell, 2002) have no such difficulties while having the same limitations*

Following the modifications to the model's equations in agreement with the suggestions of Referee nr.1 and Referee nr.2, we have further retuned the model towards more realistic values of production and export. This was done by slightly modifying a few biogeochemical parameters (coagulation rate, Ca-P formation rate) and one physical parameter (vertical mixing in the shelf) in the range of plausible values. Further changes to the model, following recent literature, consisted in distinguishing between a water column and a sediment half saturation constant for aerobic remineralization. The combination of these changes in the equations and in a few parameter values allowed us to increase production and export (bringing them closer to the lower end of modern estimates), while still maintaining the values of export production and burial to export ratio in the range of current estimates. We modified the discussion of the model evaluation (subsection 3.2) accordingly.

*EC 3&4) Although you mention the timescales over which you model can easily be run (as well as alluding to simulating it across the Phanerozoic), none of your figures show the time evolution of your model; it might be informative for readers were you to include some indication of the model's time-evolution from, say, the arbitrary initial conditions that you list in Table 2 (e.g. see Figure 5 of Yool & Tyrrell, 2002)*
*On a related point, how does your model compare in terms of residence times of the elements it represents?; given that it includes river inputs of P and reasonably represents ocean concentrations, I would expect it to do a good job here, but it might still be worth drawing attention to this*

To address this comment, we have included in the manuscript an additional figure (revised paper's **Figure 3**) to show an example of modelled tracer evolution. Furthermore, we have divided the Evaluation section in two parts with an initial subsection 3.1 focusing on the model dynamics and a second subsection 3.2 focusing on the comparison to model ocean budgets and fluxes (which corresponds to the old Evaluation before revisions).

Subsection 3.1 of the revised manuscript reads as follows:

**3.1 Timescales**

Starting from the initial values listed in Table 3, the modelled state variables evolve towards equilibrium for any couple of values of $z_{rem}^S$ and $z_{rem}^L$ in the explored interval. Simple mass conservation checks show no hidden source or sink of tracers in the model's boxes. Figure 3 illustrates an example of evolution of the variables for $z_{rem}^S$ and $z_{rem}^L$ in the middle of the interval of explored values for both particle types. In all the ocean boxes, P shows an initial oscillation that evolves on timescales of tens of thousands of years (Figure 3a,b), as expected by the typical timescale of evolution of the tracer (Lenton and Watson, 2000). This is followed by a slower drift which depends on the dynamics of the deep water oxygen content, as the release and burial of P in the sediments depends on the level of oxygenation of the deep ocean and especially of the deep shelf sea. P reaches complete equilibrium as soon as the deep ocean boxes become stably oxygenated. The timescales of evolution of O are slower and lay on the order of tens of millions of

years (Lenton and Watson, 2000). Oxygen in the deep shelf overcomes hypoxia after the first few millions of years and then slowly evolves towards equilibrium on the same timescale of O in the other ocean boxes. The dynamics of SedPorg is also strongly driven by level of oxygenation of the deep shelf sea.

Figure 3 of the revised manuscript is shown below:

[Figure]

**Figure 3: Evolution of the state variables from the initial conditions listed in Table 2 and remineralization lengths roughly in the middle of the interval of explored values: $z_{rem}^S = 20$ m, $z_{rem}^L = 250$ m. (a) Evolution of inorganic phosphorus P in the water column (left axis) and of organic phosphorus in the sediments $SedP_{org}$ (right axis); (b) zoom on the dynamics of P in the two hundred thousand years; (c) Evolution of oxygen in the water column (left axis) and atmosphere (right axis). In subplot (c) the two lines $O^{ss}$ and $O^{so}$ are overlapping: the two variables evolve closely due to the couplingof the surface ocean with the atmosphere via air-sea gas exchange.**

**Answer to Anonymous Referee nr.1**

We thank Anonymous Referee nr.1 for their thoughtful comments which allowed us to improve the model and the manuscript. We especially appreciated the Referee's corrections regarding the benthic model and describe below our related changes to the code. We also provide a point by point answer to all of their comments. The Referee's comments are highlighted in blue, actual changes to the manuscript are highlighted in red.

*R1C1)*
*The model for phosphorus (P) degradation in marine sediments considers aerobic respiration (Eq. 11) but seems to ignore anaerobic degradation. As a result the burial efficiency increases when oxygen is deleted in ambient bottom waters whereas the available observations show that P burial efficiency actually decreases under low-oxygen conditions (Slomp et al., 2002; Van Cappellen and Ingall, 1994; Wallmann, 2010). The authors should try to change their benthic model (i.e. include anaerobic degradation and enhanced P release under anoxia) or explain why they apparently ignore the strong evidence for enhanced benthic P release under low oxygen conditions.*

We agree with Referee nr.1 that our model was failing at correctly representing the dependence of P degradation and burial on $O_2$ concentration, and have therefore modified the benthic model accordingly.

Our new model representation of the sediments includes an enhancement of the rate of $P_{org}$ remineralization (+25%) and decrease of Ca-P burial (-50%) under anoxic conditions compared to oxic conditions, in line with Slomp & Van Cappellen 2007. The new equations are described in detail in subsections 2.2.4 and 2.2.5 of the revised paper, and can be read in full in the paper's appendix. The two new parameters determining changes in the rates of remineralization and Ca-P formation under anoxic conditions have been added to Table 4.

Despite these changes, our model results and conclusions regarding P and $O_2$ concentrations as a function of the particle remineralization lengths (Figure 4 and following ones) remain the same.

- New formulation of sediment $P_{org}$ remineralization for the sediment box i overlaid by the deep ocean box j:

$$SedRem\_ox^i = rm_r \cdot SedP_{org}{}^i \cdot (O^j/(O^j + K_O^s))$$

$$SedRem\_an^i = (rm_r \cdot fe_{an}) \cdot SedP_{org}{}^i \cdot (1 - O^j/(O^j + K_O^s))$$

where $fe_{an}$=1.25 is the remineralization rate enhancement factor in anoxic conditions.

- New formulation of CaP formation for the sediment box i overlaid by the deep ocean box j:

$$CaPform\_ox^i = CaP_r \cdot (SedP_{org}{}^i)^2 \cdot (O^j/(O^j + K_O^s))$$

$$CaPform\_an^i = (CaP_r \cdot fs_{an}) \cdot (SedP_{org}{}^i)^2 \cdot (1 - O^j/(O^j + K_O^s))$$

where $fs_{an}$=0.5 is the Ca-P formation rate suppression factor in anoxic conditions.

The revised subsection 2.2.5 on sediment remineralization reads as follows:

In each sediment box $i$, remineralization of $SedP_{org}$ happens in a similar way to remineralization in the water column, with an aerobic and an anaerobic component. The first takes up oxygen from the overlaying deep-water box $j$ and happens at a constant rate $rm_r$, while being limited by a Michaelis-Menten coefficient. Anaerobic remineralization releases its product to the atmosphere and happens at a faster rate $rm_r{}^* = rm_r \cdot fe_{an}$ with $fe_{an} > 1$, in agreement with recent observations and previous models (Slomp and Van Cappellen, 2006). Total sediment remineralization is therefore the sum of the two terms as in:

$$SedRem^i = rm_r \cdot SedP_{org}{}^i \cdot (O^j/(O^j + K_O^s)) + (rm_r \cdot fe_{an}) \cdot SedP_{org}{}^i \cdot (1 - O^j/(O^j + K_O^s)) \tag{11}$$

The revised subsection 2.2.4 on Ca-P formation reads as follows:

Ca-P formation happens at a lower rate under low oxygen conditions ($CaP_r{}^* = CaP_r \cdot fs_{an}$ with $fs_{an} < 1$), in agreement with observations and previous models (Slomp and Van Cappellen, 2006). The transition from aerobic and anaerobic conditions is controlled by a Michaelis-Menten type of function of the oxygen concentration in the deep ocean box j overlaying the sediment box i. The oxic and anoxic terms sum to the total formation term as in:

$$CaPform^i = \left(SedP_{org}{}^i\right)^2 \cdot \left[CaP_r \cdot O^j/(O^j + K_O^s) + (CaP_r \cdot fs_{an}) \cdot (1 - O^j/(O^j + K_O^s))\right] \tag{8}$$

*R1C2)*
*The shelf model ignores P burial in shallow-water shelf sediments even though observations in the modern ocean indicate that most burial of particulate organic matter (POM) occurs in the inner shelf region at < 50 m water depth (Dunne et al., 2007). The authors should try to change their benthic model to include shallow shelf burial or explain why they ignore burial in shallow shelf regions.*

We agree with Referee nr.1 that a significant fraction of the burial of POM happens in the shallowest regions of the shelf. Some of this is heavily influenced by the input of sediments and POM from the land. This shallowest portion of the shelf is also the most influenced by the details of the local bathymetry, tides, sediment resuspension and bottom layer mixing (Hill et al., 2008; Simpson and Pingree, 1978), which have important consequences in terms of physical-biogeochemical interactions. Differently from other models which distinguish between shallow (< 50 m deep) shelf and slope region (e.g., Slomp & Van Cappellen, 2007), BPOP calculates the sedimentation and burial fluxes dynamically, and would therefore require accounting for extra physical processes and flux parameterizations to justify going to such a higher level of detail. As our principal focus is on the first-order response to changes in the biological pump, we think that a 4-box representation of the ocean and of its circulation is a reasonable simplification, allowing us to keep the number of parameters small. We therefore keep this suggestion in mind for future developments of the model.

*R1C3)*
*Small (slowly sinking) particles are mostly degraded in the water column whereas a substantial fraction of the large (rapidly sinking) particles is not degraded but deposited at the seafloor. Consequently, large POM particles reaching the seabed are more reactive (fresher) than small (older) particles and the kinetic constant for benthic degradation should increase with increasing particle size (Stolpovsky et al., 2018). Since particle size (sinking speed, mineralization length) is the major parameter varied in the modeling, the authors should try to consider this effect in their benthic model.*

We thank Referee nr.1 for this interesting suggestion regarding the relation that may exist between the particles' remineralization rates in the sediments and their size or sinking speed. As BPOP allows to explore a variety of different hypothesis, we have decided to test the possibility of having sediment remineralization rates that depend on the particle properties, and we show our results below.

However, we highlight that the remineralization length $z_{rem}$ (around which the model is built) does not only depend on the particles' sinking speeds, but also on the particles' remineralization rates in the water column. By this definition, it may not be given that particles that have larger remineralization lengths (our "large particles") have also larger sinking speeds. They may instead just be more refractory to water column remineralization, which also allows the particles to be remineralized on average more at depth. Other factors, such as the fact that large particles are defined as a secondary product of the coagulation of smaller particles can also influence the liability the $LP_{org}$ pool. For these reasons we take this suggestion as a sensitivity experiment, but do not include it in the revised baseline model.

As a simple hypothesis, we have assumed that the sediment remineralization rate $rm_r$ increases linearly with $z_{rem}$ by 40% of our baseline value going from $z_{rem} = 0$ to $z_{rem} = 450$ m, with the baseline value of $rm_r$ being found at $z_{rem}=225$m, which is in the middle of the full range of explored values. In each run, the remineralization rate is calculated separately for small and large particles according to their respective $z_{rem}$. The two types of particles in the sediments are then solved separately and remineralized according to their respective rate. We have re-run the model under this assumption and include below a few significant comparison plots of the results of both our baseline run (the one adopted in the model) and the sensitivity run.

Our results show that there is no significant change in the model results that affects our primary conclusions from the baseline run. The main difference between the two sets of results consists in the facts that the sensitivity study shows a slightly stronger decoupling between the influence of the small and large particle remineralization lengths (respectively, $z_{rem}^S$ and $z_{rem}^L$) on the equilibrium budgets and fluxes. The value of $z_{rem}^S$ becomes even more influential for high $z_{rem}^L$, where the latter seems to make little difference in determining the equilibrium state of the model.

[Figure]

*Figure R1-1: Ocean P and O₂ budgets for 2 model configurations: constant remineralization rate (subplots (a) and (b), new baseline); remineralization rate dependent on the particles' remineralization length (subplots (c) and (d)).*

[Figure]

*Figure R1-2: Oxygen concentrations in the deep shelf (ds) and in the deep open ocean (do) for 2 model configurations: constant remineralization rate (subplots (a) and (b), new baseline); remineralization rate dependent on the particles' remineralization length (subplots (c) and (d)).*

[Figure]

*Figure R1-3: Total ocean production for 2 model configurations: constant remineralization rate (subplots (a) and (b), new baseline); remineralization rate dependent on the particles' remineralization length (subplots (c) and (d)).*

*R1C4)*
*Considering these model limitations, I do not know whether the authors' conclusion: "shelf ocean anoxia can coexist with an oxygenated deep ocean" (abstract, line 19) is really valid. Moreover, this conclusion depends on the model assumption that deep water formation takes place in the open ocean. This assumption is questionable since much of the modern deep water formation happens at continental margins. If these margin sites are oxygen depleted the resulting deep water would also be oxygen depleted.*

We have changed the wording in the abstract to say that the results "suggest" this can happen (rather than "highlight" that it can) – the use of the word "can" should anyway have been taken to imply this is one of several possibilities. Regarding  the impact of our representation of ocean circulation on the modelled oxygen distribution, in the modern ocean, deep water formation happens mostly at very high latitudes i.e., in the subpolar North Atlantic (Labrador Sea and Greenland Sea), in the Southern Ocean (Weddel Sea and Ross Sea). Both regions are characterized by physical and biogeochemical properties that differ substantially from our representation of a shelf sea, which is not intended to include polar and subpolar regions of sea-ice cover, continental ice shelves, deep convection etc. Therefore we don't think that attributing deep water formation to the shelf sea in our model would be the correct way to represent ocean circulation. Furthermore, our estimate of open ocean mixing and vertical exchange of water refers mostly to open ocean fluxes both at high and at low latitudes, and is in line with current estimates (Sarmiento and Gruber, 2006; Ganachaud and Wunsch, 2000). Instead it is our assumption of a circulation flux from the deep open ocean to the shelves, which thus boosts shelf nutrient concentration, which is more critical for determining the modelled oxygen distribution.

Bibliography (Answers to Anonymous Referee nr.1)

Hill, A. E., Brown, J., Fernand, L., Holt, J., Horsburgh, K. J., Proctor, R., Raine, R., and Turrell, W. R. ( 2008), Thermohaline circulation of shallow tidal seas, Geophys. Res. Lett., 35, L11605, doi:10.1029/2008GL033459.

Schlünz, B., and R. R. Schneider. "Transport of terrestrial organic carbon to the oceans by rivers: re-estimating flux-and burial rates." *International Journal of Earth Sciences* 88.4 (2000): 599-606.

Simpson J.H., Pingree R.D. (1978) Shallow Sea Fronts Produced by Tidal Stirring. In: Bowman M.J., Esaias W.E. (eds) Oceanic Fronts in Coastal Processes. Springer, Berlin, Heidelberg

Slomp, C. P. and Van Cappellen, P.: The global marine phosphorus cycle: sensitivity to oceanic circulation, Biogeosciences, 4, 155–171, https://doi.org/10.5194/bg-4-155-2007, 2007.

**Answer to Anonymous Referee nr.2**

We thank Anonymous Referee nr.2 for their thoughtful comments and corrections which allowed us to fix some important typos in the manuscript and encouraged us to rethink about the way in which the model is implemented. We provide below a point by point answer to all of their comments. The referee's comments are highlighted in blue, while changes to the manuscript are highlighted in red.

*R2C1)*
*It is stated on line 10 of page 4 that any organic matter that does not reach the sediments is instantaneously remineralised. This decision seems quite reasonable, but it is contradicted by the equations, in which organic matter is not only remineralized but is also advected and mixed. I recommend to do either one thing or the other but not both. If suspended particles are going to be mixed around in the model then they should have their own separate ordinary differential equations and state variables. Alternatively, if particle flux and remineralisation are made instantaneous, as stated in the manuscript, then there should be no mixing or advection of particulate organic matter. Whichever way it is done, the descriptions in the text need to be made consistent with the equations.*

We thank Referee nr.2 for their observation, and we agree with them on the fact that our choice of combining an implicit sinking scheme with lateral advection and mixing can be troublesome. For this reason, we have modified the code in order to switch off the lateral advective and vertical mixing fluxes of POC. In our new baseline run, POC is exclusively redistributed throughout the boxes by vertical sinking, in line with the Referee's suggestion. In order to account for these changes in the model, we have modified the manuscript as described below.

The description of the $P_{org}$ physical fluxes in section 2.2.2 has been reformulated as:

The implicit representation of the organic matter in the water column implies that no organic matter is accumulated in the ocean. In our baseline version of the model, corresponding to the results presented in this manuscript, SPorg and LPorg are redistributed throughout the watercolumn exclusively by implicitly modelled gravitational sinking before being either buried, accumulated in the sediments or remineralized. Even though the vertical export by downwelling and mixing (Stukel and Ducklow, 2017), and the lateral organic matter redistribution (Lovecchio et al., 2017;Inthorn et al., 2006) may be important when working with suspended SPorg (zremS = 0), these fluxes are not currently accounted for in the model.

We have modified Figure 1 and 2 in order to account for this change in the model.

We have run the revised model both with and without the lateral advective and vertical mixing fluxes in order to understand the impact of removing these fluxes onto our model results.
We include below a few significant plots which show overall higher oxygen levels and production in the absence of mixing/advective fluxes of $P_{org}$. Our main conclusions remain unchanged.

[Figure]

*Figure R2-1: Ocean P and O₂ budgets for 2 model configurations: without POC lateral advection and vertical mixing (subplots (a) and (b), new baseline); with POC lateral advection and vertical mixing (subplots (c) and (d)).*

[Figure]

*Figure R2-2: Oxygen concentrations in the deep shelf (ds) and in the deep open ocean (do) for 2 model configurations: without POC lateral advection and vertical mixing (subplots (a) and (b), new baseline); with POC lateral advection and vertical mixing (subplots (c) and (d)).*

[Figure]

*Figure R2-3: Total ocean production for 2 model configurations: without POC lateral advection and vertical mixing (subplots (a) and (b), new baseline); with POC lateral advection and vertical mixing (subplots (c) and (d)).*

*R2C2)*
*Tables 1 to 3 are very helpful. Another table needs to be added, listing the state variables in the model and stating their units. In addition, the units of all equations (the left-hand side) should also be stated, if they are not already given in the tables.*

As suggested by Referee nr.2, we have added a table (**Table 1** of the revised manuscript) which indicates names and units of the state variables of the model.

| Name | Description | Units |
|---|---|---|
| $P^{ss}$ | Inorganic phosphorus in surface shelf sea box | mmol m$^{-3}$ |
| $P^{ds}$ | Inorganic phosphorus in deep shelf sea box | mmol m$^{-3}$ |
| $P^{so}$ | Inorganic phosphorus in surface open ocean box | mmol m$^{-3}$ |
| $P^{do}$ | Inorganic phosphorus in deep open ocean box | mmol m$^{-3}$ |
| $O^{ss}$ | Molecular oxygen in surface shelf box | mmol m$^{-3}$ |
| $O^{ds}$ | Molecular oxygen in deep shelf box | mmol m$^{-3}$ |
| $O^{so}$ | Molecular oxygen in surface open ocean box | mmol m$^{-3}$ |
| $O^{do}$ | Molecular oxygen in deep open ocean box | mmol m$^{-3}$ |
| $O^{at}$ | Oxygen mixing ratio in atmosphere (mol mol$^{-1}$) | - |
| $SedP_{org}^{s}$ | Organic phosphorus in the sediments of the shelf sea | mmol m$^{-2}$ |
| $SedP_{org}^{o}$ | Organic phosphorus in the sediments of the open ocean | mmol m$^{-2}$ |
| $P^{TOT}$ | Diagnostic variable: total P budget from sources and sinks only | Tmol P |
| $O^{TOT}$ | Diagnostic variable: total O budget from sources and sinks only | Pmol O$_2$ |

**Table 1: List of the model's state variables and of their units**

We mention the new table in subsection 2.1 as follows:

The entire set of the model's state and diagnostic variables and their units are listed in Table 1.

*R2C3)*
*Too many of the equations in the model are dimensionally inconsistent. That is to say, the units on the left-hand side of the equation do not match the units on the right-hand side of the equation when the different terms are combined together. As an example, equation 4 on page 4 is an equation for the rate of organic matter production in units of C2 GMDD Interactive comment Printer-friendly version Discussion paper phosphorus. This is a flux (rate of transfer), and therefore has to be in units of Moles y-1 or mmol m-3 y-1 or similar. Because this is an ongoing flux rather than a one-off transfer, it must be expressed as a rate of transfer per unit time. However, none of the terms on the right-hand side of the equation have time anywhere in their units. The equation is formulated in such a way that it appears to be aimed at converting a fraction of the surface phosphorus concentration into production at each timestep, but the way it is actually formulated means that the rate of*

*conversion of surface phosphate into production (organic matter) will depend on the timestep used. Shorter time steps will convert phosphate to organic matter more rapidly than longer timesteps whereas ODE equations should be timestep-independent. Equation 5 is another example, where, according to the equation, 'Coag' must have units of organic matter concentration squared per year, which makes no sense. Before resubmission, I recommend that every equation in the model is checked for dimensional (units) consistency: multiplying through the units of the terms on the right-hand side should produce the units of the term on the left-hand side.*

We thank Referee nr.2 for having spotted a few mistakes in the manuscript's equations. We have double checked the entire set of equations, the parameters' units and we have corrected a few typos.

As highlighted by the Referee, the uptake rate $P_{eff}$ (Equation 4) as well as the coagulation factor $cg_r$ (Equation 5) were missing part of their units (see Table 4 of the revised manuscript). We have now corrected their units accordingly (see also Gruber et al., 2006). We have explained better the values and magnitudes of the parameters in the table's caption. We have also spotted another few inconsistencies in the units and fixed them both in the tables and in the equations.

*R2C4)*

*I did not notice a statement anywhere that conservation of mass (or, more properly, conservation of total inventories of elements) has been checked and found to be stable. This is easy to do for a box model, whether it is closed or open. Obviously for a closed model, if there are no errors in the equations, the total sum of atoms of a given element should be constant over time. For an open model, the changes in the total inventory over time should exactly match the sum of external inputs over time minus the sum of the outputs from the system over time (i.e. ΔInv= - ). This can easily be checked by adding two extra differential equations to the model: one to track the sum of the external inputs to the system and another to keep track of the sum of the losses from the system as a whole. From the model equations, I suspect that the model does not properly conserve phosphorus and oxygen but rather there is some (unintended) cumulative creation/destruction over time. This will interfere with the ability of the model to be run over long timescales to address the geological timescale questions of interest. It is a little bit unclear, but it appears that the amount of phosphate removed per unit time from the surface box as particle export is not identical to the amounts of phosphate added per unit time to the deep box and sediments combined. The euphotic zone depth appears in the equation for the former but not the latter, for instance, whereas if it appears in one then it should also appear in the other. Again, checks can be made by adding extra (book-keeping) ODEs to the model. For this example, one extra ODE could tally up the cumulative export from the surface box and another extra ODE could tally up the sum of the cumulative inputs to the deep and sediment boxes. At the end of each model run, a quick numerical check can be made to ensure that the tallies are identical within the precision of numerical rounding errors.*

We agree with Referee nr.2 that a check on conservation of mass is essential for the model's evaluation. Even though we did not discuss it explicitly, our model does go to equilibrium on the long term, as now visible from **Figure 3** of the revised manuscript.

As a further check we have now added 2 extra variables to the model, $P^{TOT}$ and $O^{TOT}$, which represent the total P and O budgets. These two variables (in units of TmolP and $PmolO_2$, respectively) are initialized to the total sum of the two tracers across all boxes at time zero.

At each time step, P$^{TOT}$ and O$^{TOT}$ evolve according to the sum of all sources and sinks in the model. Their derivatives are as follows:

$$\frac{dP^{TOT}}{dt} = \left[ P_{in} - \left( CaPform^s \cdot A_{ocean} \cdot \mathcal{P}_{shelf} + CaPform^o \cdot A_{ocean} \cdot \left( 1 - \mathcal{P}_{shelf} \right) \right) \right] \cdot 10^{-15}$$

$$\frac{dO^{TOT}}{dt} = \Big[ OProd^{ss} \cdot V^{ss} + OProd^{so} \cdot V^{so}$$
$$- \left( \left( SedRemO^{ds} + AerRem^{ds} \right) \cdot V^{ds} + \left( SedRemO^{do} + AerRem^{do} \right) \cdot V^{do} \right.$$
$$\left. + OxyWeath \cdot \left( Mol_{atmo} \cdot 10^3 \right) \right) \Big] \cdot 10^{-18}$$

At the end of each model run we plot the sum of the total P and O content in the model at each time step by multiplying the concentrations in each box by the box volume or area (the latter for the sediments), and check that this corresponds to the time evolution of P$^{TOT}$ and O$^{TOT}$ calculated at run time regardless of the partition of the tracers among the boxes.

In order to include these new diagnostic variables in the model's description, we have modified the first paragraph of subsection 2.1 as follows:

The box model resolves explicitly for each relevant box the local concentrations of three types of tracers: molecular oxygen O$_2$ (O), inorganic dissolved phosphorus (P) and sediment organic phosphorus (SedP$_{org}$). The total budgets of P and O, respectively P$^{TOT}$ and O$^{TOT}$, are also independently integrated from the net sources and sinks of the two tracers over the entire model domain, for the purpose of checking mass conservation.

We have also added the following statement to subsection 3.1 of the model evaluation:

Starting from the initial values listed in Table 3, the modelled state variables evolve towards equilibrium for any couple of values of z$_{rem}$$^S$ and z$_{rem}$$^L$ in the explored interval. Simple mass conservation checks show no hidden source or sink of tracers in the model's boxes.

We include below the result for the choice of z$_{rem}$$^S$ and z$_{rem}$$^L$ used for Figure 3 of the revised paper.

[Figure]

*Figure R2-4: Comparison between the evolution of the total tracer budgets ($P^{TOT}$ and $O^{TOT}$) calculated at run time from the tracer's net input fluxes (total sources + total sinks), and the tracer budgets calculated integrating the local concentrations in the boxes at each time step from the model's output.*

[revised manuscript text omitted]
 = \left(SP_{org}^{\ j} \cdot exp\left(-\Delta Z_j\ /z_{rem}^S\right) + LP_{org}^{\ j} \cdot exp\left(-\Delta Z_j\ /z_{rem}^L\right)\right) \cdot \Delta Z_j \tag{7}$$

The accumulated SedP$_{org}$ is partially slowly remineralized and partially irreversibly buried in a mineral form. Phosphorus burial as mineral Ca-P is modelled as a function of the square of SedP$_{org}$ that accumulates in the sediments, in a way that is analogous to the dynamics of particle coagulation, and is regulated by a constant rate coefficient CaP$_r$. Ca-P formation happens at a lower rate under low oxygen conditions (CaP$_r^*$ = CaP$_r \cdot$ fs$_{an}$ with fs$_{an}$ < 1), in agreement with observations and previous models (Slomp and Van Cappellen, 2006). The transition from aerobic and anaerobic conditions is controlled by a Michaelis-Menten type of function of the oxygen concentration in the deep ocean box j overlaying the sediment box i. The oxic and anoxic terms sum to the total formation term as in:

$$CaPform^i = \left(SedP_{org}^{\ i}\right)^2 \cdot \left[CaP_r \cdot O^j/(O^j + K_O^s)\ +\ (CaP_r \cdot fs_{an})\ \cdot\ \left(1 - O^j/(O^j + K_O^s)\right)\right]\ \ \sout{CaP_r \cdot}$$
$$\sout{(SedP_{org}^{\ i})^2}\ \ \rule{3cm}{0.4pt} \tag{8}$$

This flux is essential to balance the continuous P river input, therefore preventing the ocean from overflowing with nutrients.

**2.2.5 Remineralization in the water column and sediments**

At each time step, remineralization in the water column completely depletes the P$_{org}$ that has not reached the sediments. In the two surface boxes, remineralization of P$_{org}$ that is not exported below the euphotic layer uses up part of the oxygen that was released by production. For this reason, net oxygen production in each surface box is proportional to the export of P$_{org}$ below the euphotic layer. Export from a surface box $i$ to a deep box $j$ happens both via gravitational sinking and via mixing, as in:

$$VExp^i = SP_{org}^{\ i} \cdot exp\left(-(\Delta Z_{eu}/2)\ /z_{rem}^S\right) + LP_{org}^{\ i} \cdot exp\left(-(\Delta Z_{eu}/2)\ /z_{rem}^L\right)\underline{\hspace{3cm}} \tag{9}$$

$$\sout{VExp^i = SP_{org}^{\ i} \cdot exp\left(-(\Delta Z_{eu}/2)\ /z_{rem}^S\right) + LP_{org}^{\ i} \cdot exp\left(-(\Delta Z_{eu}/2)\ /z_{rem}^L\right) +}$$
$$\sout{(SP_{org}^{\ i} + LP_{org}^{\ i}) \cdot Mix_{ij}/V^i} \tag{9}$$

At depth, the remineralization of P$_{org}$ that does not reach the sediments happens through both aerobic and anaerobic processes, completely depleting the remaining P$_{org}$. Water-column remineralization of P$_{org}$ into inorganic P in the deep box $j$ is therefore calculated as:

$$WcRem^j =\ SP_{org}^{\ j} \cdot \left(1 - exp\left(-\Delta Z_j\ /z_{rem}^S\right)\right) + LP_{org}^{\ i} \cdot \left(1 - exp\left(-\Delta Z_j\ /z_{rem}^L\right)\right) \
[revised manuscript text omitted]

$$\cancel{LatExp^{ss} = SP_{org}{}^{ss} \cdot (Upw + Mix_{ls})/V^{ss}} \tag{A7}$$

$$VExp\_SP_{org}{}^{ss} = (SP_{org}{}^{ss} \cancel{- LatExp^{ss}}) \cdot (exp(-(\Delta Z_{eu}/2) /z_{rem}^S) \cancel{+ Mix_{vs}/V^{ss}})$$

(A$\underline{8}$$\cancel{7}$)

$$VExp\_LP_{org}{}^{ss} = LP_{org}{}^{ss} \cdot (exp(-(\Delta Z_{eu}/2) /z_{rem}^L) \cancel{+ Mix_{vs}/V^{ss}})$$

(A$\underline{9}$$\cancel{8}$)

$$\frac{dP^{ss}}{dt} = P_{in} \cdot \left(1 - \mathcal{P}_{open}\right)/V^{ss} + \left(Upw \cdot (P^{ds} - P^{ss}) + Mix_{ls} \cdot (P^{so} - P^{ss}) + Mix_{vs} \cdot (P^{ds} - P^{ss})\right) \cdot spy/V^{ss}$$
$$- \left(\text{VExp}_{SP_{org}}{}^{ss} + \text{VExp}_{LP_{org}}{}^{ss}\right)$$

(A9̶1̶0̶)

$$AirSea^{ss} = K_W \cdot (O^{at}/K_{Henri} - O^{ss}) \cdot (A_{ocean} \cdot \mathcal{P}_{shelf})/V^{ss} \tag{A10̶1̶}$$

$$OProd^{ss} = OP_{Red} \cdot \left(VExp_{SP_{org}}{}^{ss} + VExp_{LP_{org}}{}^{ss}\right) \tag{A11̶2̶}$$

$$\frac{dO^{ss}}{dt} = \left(Upw \cdot (O^{ds} - O^{ss}) + \text{Mix}_{ls} \cdot (O^{so} - O^{ss}) + \text{Mix}_{vs} \cdot (O^{ds} - O^{ss})\right) \cdot spy/V^{ss} + AirSea^{ss}$$
$$+ OProd^{ss}$$

$$\rule{9cm}{0.4pt} \tag{A12̶3̶}$$

**A.3 Deep shelf sea (ds)**

$$V^{ds} = \Delta Z_{ds} \cdot A_{ocean} \cdot \mathcal{P}_{shelf}$$

(A13̶4̶)

$$VInp\_SP_{org}{}^{ds} = VExp\_SP_{org}{}^{ss} \cdot (V^{ss}/V^{ds})$$

(A14̶5̶)

$$SP_{org}{}^{ds} = VInp\_SP_{org}{}^{ds} - cg_r \cdot (VInp\_SP_{org}{}^{ds})^2$$

(A15̶6̶)

$$LP_{org}{}^{ds} = VExp\_LP_{org}{}^{ss} \cdot (V^{ss}/V^{ds}) + cg_r \cdot (VInp\_SP_{org}{}^{ds})^2$$

(A16̶7̶)

$$\cancel{LatExp^{ds} = SP_{org}{}^{ds} \cdot Mix_{ta}/V^{ds}} \tag{A18}$$

$$Rem\_SP_{org}{}^{ds} = \cancel{(}SP_{org}{}^{ds} \cancel{- LatExp^{ds})} \cdot (1 - exp(-\Delta Z_{ds}/z_{rem}^S))$$

(A17̶9̶)

$$Rem\_LP_{org}{}^{ds} = LP_{org}{}^{ds} \cdot (1 - exp(-\Delta Z_{ds}/z_{rem}^L)) \tag{A18̶2̶0̶}$$

$$AerRem\_SedP_{org}{}^{ds} = rm_r \cdot SedP_{org}{}^s/\Delta Z_{ds} \cdot (O^{ds}/(O^{ds} + K_O^s\cancel{K_o}))$$

(A19̶2̶1̶)

$$AnaRem\_SedP_{org}{}^{ds} = (rm_r \cdot fe_{an}) \cdot SedP_{org}{}^{S}/\Delta Z_{ds} \cdot (1 - O^{ds}/(O^{ds} + K_O^s)) \quad\text{(A20)}$$

$$\frac{dP^{ds}}{dt} = (Upw \cdot (P^{do} - P^{ds}) + Mix_{ld} \cdot (P^{do} - P^{ds}) + Mix_{vs} \cdot (P^{ss} - P^{ds})) \cdot spy/V^{ds}$$
$$+ \left(Rem\_SP_{org}{}^{ds} + Rem\_LP_{org}{}^{ds} + AerRem\_SedP_{org}{}^{ds} + AnaRem\_SedP_{org}{}^{ds}\right)$$
$$+ \left(Rem\_SP_{org}{}^{ds} + Rem\_LP_{org}{}^{ds} + Rem\_SedP_{org}{}^{ds}\right)$$

5    (A21)

$$AerRemWcO^{ds} = OP_{Red} \cdot (Rem\_SP_{org}{}^{ds} + Rem\_LP_{org}{}^{ds}) \cdot (O^{ds}/(O^{ds} + K_O^w))$$

(A22)

$$AerRemSedO^{ds} = OP_{Red} \cdot AerRem\_SedP_{org}{}^{ds} \quad\text{(A23)}$$

10    $$\frac{dO^{ds}}{dt} = (Upw \cdot (O^{do} - O^{ds}) + Mix_{ld} \cdot (O^{do} - O^{ds}) + Mix_{vs} \cdot (O^{ss} - O^{ds})) \cdot spy/V^{ds} - AerRemWcO^{ds}$$
$$- AerRemSedO^{ds}$$

(A24)

**A.4 Surface open ocean (so)**

$$V^{so} = \Delta Z_{eu} \cdot A_{ocean} \cdot (1 - \mathcal{P}_{shelf})$$

(A25)

$$Prod^{so} = P_{eff} \cdot (P^{so}/(P^{so} + K_P)) \cdot P^{so}$$

20    (A26)

$$LatInp^{so} = SP_{org}{}^{ss} \cdot (Upw + Mix_{ls})/V^{so} \quad\text{(A28)}$$

$$SP_{org}{}^{so} = (Prod^{so} + LatInp^{so}) - cg_r \cdot ((Prod^{so}) + LatInp^{so})^2$$
        (A27)

$$LP_{org}{}^{so} = cg_r \cdot (Prod^{so} + LatInp^{so})^2$$

25    (A28)

$$VExp\_SP_{org}{}^{so} = SP_{org}{}^{so} \cdot (exp(-(\Delta Z_{eu}/2)/z_{rem}^S) + Mix_{vo}/V^{so})$$

(A29)

$$VExp\_LP_{org}{}^{so} = LP_{org}{}^{so} \cdot (exp(-(\Delta Z_{eu}/2)/z_{rem}^L) + Mix_{vo}/V^{so})$$

(A30)

$$\frac{dP^{so}}{dt} = P_{in} \cdot \mathcal{P}_{open}/V^{ss} + (Upw \cdot (P^{ss} - P^{so}) + Mix_{ls} \cdot (P^{ss} - P^{so}) + Mix_{vo} \cdot (P^{do} - P^{so})) \cdot spy/V^{so}$$
$$- \left(VExp\_SP_{org}{}^{so} + VExp\_LP_{org}{}^{so}\right) +$$

$$\sout{\left(VExp\_SP_{org}{}^{so} + VExp\_LP_{org}{}^{so}\right)}$$

(A31)

$$AirSea^{so} = K_W \cdot (O^{at}/K_{Henri} - O^{so}) \cdot (A_{ocean} \cdot (1 - \mathcal{P}_{shelf}))/V^{so}$$

(A32)

$$OProd^{so} = OP_{Red} \cdot \left(VExp\_SP_{org}{}^{so} + VExp\_LP_{org}{}^{so}\right)$$

(A33)

$$\frac{dO^{so}}{dt} = \left(Upw \cdot (O^{ss} - O^{so}) + Mix_{ls} \cdot (O^{ss} - O^{so}) + Mix_{vo} \cdot (O^{do} - O^{so})\right) \cdot spy/V^{so} + AirSea^{so}$$
$$+ OProd^{so}$$

(A34)

**A.5 Deep open ocean (do)**

$$V^{do} = \Delta Z_{do} \cdot A_{ocean} \cdot (1 - \mathcal{P}_{shelf})$$

(A35)

$$VInp\_SP_{org}{}^{do} = VExp\_SP_{org}{}^{so} \cdot (V^{so}/V^{do})$$

(A36)

$$\sout{LatInp^{do} = SP_{org}{}^{ds} \cdot Mix_{la}/V^{do}} \tag{A39}$$

$$SP_{org}{}^{do} = (VInp\_SP_{org}{}^{do} \sout{+ LatInp^{do}}) - cg_r \cdot (VInp\_SP_{org}{}^{do} \sout{+ LatInp^{do}})^2$$

(A37)

$$LP_{org}{}^{do} = VExp\_LP_{org}{}^{so} \cdot (V^{so}/V^{do}) + cg_r \cdot (VInp_{SP_{org}}{}^{do} \sout{+ LatInp^{do}})^2$$

(A38)

$$Rem\_SP_{org}{}^{do} = SP_{org}{}^{do} \cdot (1 - exp(-\Delta Z_{do}/z_{rem}^S)) \tag{A39}$$

$$Rem\_LP_{org}{}^{do} = LP_{org}{}^{do} \cdot (1 - exp(-\Delta Z_{do}/z_{rem}^L)) \tag{A40}$$

$$AerRem\_SedP_{org}{}^{do} = rm_r \cdot SedP_{org}{}^o/\Delta Z_{do} \cdot (O^{do}/(O^{do} + K_O^S \sout{K_O}))$$

(A41)

$$AnaRem\_SedP_{org}{}^{do} = (rm_r \cdot fe_{an}) \cdot SedP_{org}{}^o/\Delta Z_{do} \cdot (1 - O^{do}/(O^{do} + K_O^S)) \tag{A42}$$

$$\frac{dP^{do}}{dt} = (Upw \cdot (P^{so} - P^{do}) + Mix_{ld} \cdot (P^{ds} - P^{do}) + Mix_{vo} \cdot (P^{so} - P^{do})) \cdot spy/V^{do}$$
$$+ \left(Rem\_SP_{org}{}^{do} + Rem\_LP_{org}{}^{do} + AerRem\_SedP_{org}{}^{do} + AnaRem\_SedP_{org}{}^{do}\right) + $$
$$+ \left(Rem\_SP_{org}{}^{do} + Rem\_LP_{org}{}^{do} + Rem\_SedP_{org}{}^{do}\right)$$

(A43)

$$AerRem\underline{Wc}O^{do} = OP_{Red} \cdot (Rem\_SP_{org}{}^{do} + Rem\_LP_{org}{}^{do}) \cdot (O^{do}/(O^{do} + K_O^{\underline{w}}{}^{\sim\sim O\sim\sim}))$$

(A44)

[revised manuscript text omitted]

---

## Referee Report (RR1)

**Review of Lovecchio & Lenton**

The simple model presented by the authors confirms that the remineralization length has a strong impact on ocean productivity and oxygenation. Due to the simplicity of their model, the authors are able to fully explore the parameter space. The main conclusions of the manuscript are, hence, valid within the limits given by the model architecture. The revised manuscript is in much better shape than the original version. Nevertheless, there are a few remaining issues that should be addressed in a further revision of the manuscript:

The local concentrations of organic particles suspended in the water column (SPorg and LPorg) are employed to calculate rates of coagulation, export, organic matter degradation and organic matter deposition at the seafloor. However, SPorg and LPorg do not appear as state variables in Tab. 1. On page 5, lines 1 -8, the authors explain that SPorg and LPorg are redistributed in the water column by gravitational sinking and calculated implicitly but they do not fully explain how SPorg and LPorg are constrained. The authors should better explain how these important variables are derived and specify the equations that they employ to calculate SPorg and LPorg in their model.

The authors assume that organic matter is converted into methane when oxygen is depleted. They deliberately neglect denitrification and the reduction of iron and sulfate in their model. This is a major limitation because the system behavior (e.g. changes in ocean productivity, dissolved and atmospheric oxygen) may change drastically when these processes are considered (Wallmann, Flogel et al. 2019). The authors should add a section/sentence to chapter 5.1 (Model limitations) to explain that the model outcomes would change drastically when other redox pathways and nutrients (nitrogen, iron) would be considered in the modeling.

The authors assume that anaerobic degradation is faster than aerobic degradation (page 6, line 30,  $fe_{an} > 1$ ). This assumption is valid for organic P ( $P_{org}$ ) but studies on the degradation kinetics of particulate organic carbon (POC) show that POC degradation either declines under anoxic conditions or proceeds at a rate similar to that observed in the presence of oxygen (Hedges, Hu et al. 1999, Burdige 2007, Dale, Sommer et al. 2015). The authors should explain and clearly specify that  $fe_{an} > 1$  is valid only for  $P_{org}$  degradation but not for POC degradation. Since the product of anaerobic POC degradation (methane) is assumed to contribute to the consumption of oxygen in the atmosphere (Eq. 17), the authors should separate  $P_{org}$  and POC degradation in their model and employ  $fe_{an} \le 1$  to simulate POC degradation.

**References**

- Burdige, D. J. (2007). "Preservation of organic matter in marine sediments: Controls, mechanisms, and an imbalance in sediment organic carbon budgets?" Chem. Rev. **107**: 467-485.
- Dale, A. W., S. Sommer, U. Lomnitz, I. Montes, T. Treude, V. Liebetrau, J. Gier, C. Hensen, M. Dengler,
  K. Stolpovsky, L. D. Bryant and K. Wallmann (2015). "Organic carbon production,
  mineralisation and preservation on the Peruvian margin." Biogeosciences 12: 1537-1559.
- Hedges, J. I., F. S. Hu, A. H. Devol, H. E. Hartnett, E. Tsamakis and R. G. Keil (1999). "Sedimentary organic matter preservation: A test for selective degradation under oxic conditions." American Journal of Science 299(7-9): 529-555.
- Wallmann, K., S. Flogel, F. Scholz, A. W. Dale, T. P. Kemena, S. Steinig and W. Kuhnt (2019). "Periodic changes in the Cretaceous ocean and climate caused by marine redox see-saw." Nature Geoscience 12(6): 456- 462.

---

## Author Response (AR2)

Author's response

**"BPOP-v1 model: exploring the impact of changes in the biological pump on the shelf sea and ocean nutrient and redox state"**

Elisa Lovecchio[1] and Timothy M. Lenton[1]
[1]Global Systems Institute, University of Exeter, Exeter, EX4 4QE, United Kingdom

Dear Editor,

Thank you for your answer. We have gone through a new round of revisions and address below your new questions and comments. We apologize for having forgotten to address the specific comments of Referee nr.2. We are including our answers below. We have also uploaded these answers in a separate file as an Author's comment in the public discussion page.

We include our new answers below in the following order:
1) Revisions round 2: comments and answers;
2) Revisions round 1: answers to previous comments, updated according to the new changes in the model;
3) revised manuscript with track changes.

We look forward to your response.
Sincerely,
Elisa Lovecchio

**Revisions Round 2**

**Answer to Editor's Comments**

We thank the Editor for his comments and suggestions, which we copy below (in blue), and explain in detail our related changes to the manuscript (in red) and model.

*EC1) Table 5: is there a reason why "deep ocean" is mentioned in the "total ocean O2" row?; is it not more straightforward to just use total ocean O2?*

We are now using the total ocean $O_2$ reservoir estimates.

*EC2) Table 5: regarding "production (prod)", I think that the upper span of the range here may have been miscalculated; the Carr et al. (2006) reference confusingly refers to estimates based on 6 months of data, but the units used suggest a maximum PP of ~60 Pg C / y (5000 Tmol C / y in your units); Figure 5a of the paper has a single GCM with a PP close to ~80 Pg C / y (6667 Tmol C / y in your units); also, why are you using Tmol C / y when Pg C / y is the more common unit, and its non-SI equivalent, Gt C / y is used in Carr et al., 2006?*

Thank you very much for this correction, we have updated Table 5 and changed the production units accordingly. The full Table 5 can be found in the updated answer to the Editor's comments from round 1 (below).

*EC3) Figure 3b: you could dispense with this blow-up by plotting both a and c here on a log(time) scale; this would expand the most interesting period of the model's evolution; however, you may have already tried this and decided against it*

Thank you for your suggestion, we have tried to use a log(time) scale, but prefer to maintain the current plotting style for Figure 3.

*EC4) Figure 3b caption: "in the \*first\* two hundred thousand years"*

Thank you.

*EC5) Page 6 of your response: you say "Anaerobic remineralization releases its product to the atmosphere" but this is unclear to me; releases its product to the atmosphere?; this is the sediments we're talking about, no?; one would assume they were quite isolated from the atmosphere*

In our model, the reducing agent produced by anaerobic remineralisation is methane gas and it is only produced when the sediments and the deep shelf water column have gone anoxic. As we do not track other oxidising agents such as $SO_4$ there is nothing for the methane to be oxidised by until it reaches the surface ocean, and as the surface ocean is equilibrated with the atmosphere, the fact that we assume oxidation in the atmosphere is a reasonable approximation. A more elaborate

model would recognise other oxidants, notably $SO_4$, and this would transfer some of the oxygen consumption to deep waters, tending to make them more anoxic. Hence our results can be seen as minimum estimates of deep shelf anoxia. (Anaerobic sediment remineralization only matters in the deep shelf, as the deep ocean doesn't really accumulate any organic matter in the sediments and never reaches suboxic levels.)

We have further explained this in our manuscript by adding the following sentences to subparagraph 2.2.5:

In our model, the reducing agent produced by anaerobic remineralisation is methane gas and it is only produced when the sediments and the deep shelf water column have gone anoxic. As we do not track other oxidising agents such as SO4 there is nothing for the methane to be oxidised by until it reaches the surface ocean, and as the surface ocean is equilibrated with the atmosphere, the fact that we assume oxidation in the atmosphere is a reasonable approximation.

*EC6) Page 7 of your response: has any of this made it into the manuscript or its supplementary material; this would be a reasonable addition to the latter if it's not already there; and it's would be of interest / value to readers in any case*

We have decided to refer to this sensitivity experiment in subsection 5.1.3 of the Discussion section of the paper, adding the following sentences:

Furthermore, we have tested the impact of having sediment remineralization rates that vary with the particles' $z_{rem}$, under the assumption that the liability of small and large particles may be different. In our experiment, we increased the remineralization $rm_r$ rate linearly with $z_{rem}$ by 40 % of our baseline value ($rm_r^0$), with $rm_r^0$ being found at the centre of the interval of explored values of $z_{rem}$ = [0 m, 450 m]. Under these conditions, we obtained a higher decoupling between the influence of $z_{rem}^S$ and $z_{rem}^L$ on budgets and fluxes, both being more strongly driven by the small particle properties for large values of $z_{rem}^L$.

*EC7) Table 3 in the revised manuscript: this O2:P ratio was mentioned by referee 2 but the point was not addressed in the authors' response, and does not appear to have been changed either*
*EC8) More generally, it's not completely clear whether you have addressed the specific comments from referee 2; the preceding point would rather suggest not*

Unfortunately, while revising the manuscript and preparing our answers we have forgotten to address the specific comments of Referee nr.2. We have now answered them and included our changes to the model below. Due to this latter changed we have further retuned the model around modern ocean values. We refer to the answer to EC2 of the revision round one for a more detailed discussion of the applied changes to the model.

**Answer to Anonymous Referee nr.2**

We thank Anonymous Referee nr.2 for their specific comments, and apologize for having forgotten to address them in our initial answer. We include our answers below, highlighting the Referee's comments in blue and our changes to the manuscript in red.

*SC 1) Key paper not cited: Reinhard CT., Planavsky NJ et al. "Evolution of the global phosphorus cycle." Nature 541, no. 7637 (2017): 386.*

Thank you for your suggestion, we have added the reference in the manuscript's discussion.

*SC 2) Equations 14 & 17: AirSea should appear the same in both*

Thank you, we have corrected it.

*SC3) Equation 17: why does anaerobic remineralisation remove oxygen?*

In our model, the reducing agent produced by anaerobic remineralisation is methane gas and it is only produced when the sediments and the deep shelf water column have gone anoxic. As we do not track other oxidising agents such as $SO_4$ there is nothing for the methane to be oxidised by until it reaches the surface ocean, and as the surface ocean is equilibrated with the atmosphere, the fact that we assume oxidation in the atmosphere is a reasonable approximation.

We have added this discussion to our manuscript in subparagraph 2.2.5 in order to clarify our equations.

*SC 4) Line 5 of page 7: anaerobic remineralisation of organic matter also releases phosphorus (in fact oxygen-depleted sediments are stronger sources of phosphorus to the overlying water column).*

We agree with Referee nr.2 and we have modified our model accordingly, following the suggestions of one of the major comments of Referee nr.1. We refer to our answer to the comment R1C1 (Referee nr.1, comment nr.1) for a detailed description of the changes to the model.

*SC 5) Table 1: moles of air or moles of oxygen in the atmospheric box?*

Oxygen in the atmospheric box is expressed in terms of its mixing ratio [mol mol$^{-1}$]. We have added a table (Table 1) with the list of the variables and of their units.

*SC 6) Table 2: the Redfield ratio of oxygen to phosphorus (-O2:P ) is ∼150:1 not 106:1 (see for instance: Anderson, L.A. and Sarmiento, J.L., 1994. Redfield ratios of remineralization determined by nutrient data analysis. Global biogeochemical cycles, 8(1), pp.65-80; Thomas, H., 2002. Remineralization ratios of carbon, nutrients, and oxygen in the North Atlantic Ocean: A field databased assessment. Global biogeochemical cycles, 16(3)).*

We thank Referee nr.2 for their comment. Our latest version of the model adopts the suggested value O2:P=150:1, and we have updated this value in Table 3.

*SC 7) W0 is a baseline flux (line 3 of page 4), hence cannot have units of mmol if equation 3 is to be dimensionally plausible.*

Thank you, we have corrected it. The correct unit is: mmol yr$^{-1}$ (see Table 3).

**Revisions Round 1 (updated)**

**Answer to Editor's Comments**

We thank the Editor for his comments and suggestions, which we copy below (in blue), and explain in detail our related changes to the manuscript (in red) and model.

*EC 1) Evaluation of your model (page 7) might benefit from a table which compares each of the quantities to the corresponding observational estimate (e.g. see Table 2 of Yool & Tyrrell, GBC, 2002)*

In agreement with this suggestion, we have added an evaluation table (**Table 5** of the revised manuscript) with the updated results of our model evaluation after revisions. We have updated our comments on the model evaluation in the related section 3.2 of the manuscript.

The table reads as follows:

| Quantity | Model | Modern values or estimates | Units | Source |
|---|---|---|---|---|
| Total ocean P | 2500 - 2800 | 3100 | TmolP | Watson et al. (2017) |
| Total ocean $O_2$ | 100 - 140 | 225-310 | $PmolO_2$ | Duursma and Boisson (1994);Keeling et al. (1993) |
| $P^{ss}$ | 1.5 – 1.9 | 1 – 1.5 | mmol m$^{-3}$ | Garcia et al. (2018b);Sarmiento and Gruber (2006) |
| $P^{ds}$ | 3.9 – 5.4 | 2.2 | mmol m$^{-3}$ | Garcia et al. (2018b);Watson et al. (2017) |
| $P^{so}$ | 0.4 – 1.1 | 0.2 - 2 | mmol m$^{-3}$ | Garcia et al. (2018b);Sarmiento and Gruber (2006) |
| $P^{do}$ | 2.0 – 2.3 | 1 - 3 | mmol m$^{-3}$ | Garcia et al. (2018b);Sarmiento and Gruber (2006) |
| $O^{ss}$ | 273 - 274 | 200 - 350 | mmol m$^{-3}$ | Garcia et al. (2018a) |
| $O^{ds}$ | 2.8 – 12.3 | 0 - 80 | mmol m$^{-3}$ | Garcia et al. (2018a) |
| $O^{so}$ | 273 | 200 - 350 | mmol m$^{-3}$ | Garcia et al. (2018a) |
| $O^{do}$ | 72 - 110 | 40 - 200 | mmol m$^{-3}$ | Garcia et al. (2018a) |
| Production (Prod) | 18.8 – 51.5 | 35 - 80 | GtC yr$^{-1}$ | Carr et al. (2006) |
| Export | 2.8 – 4.0 | 4 – 20 | GtC yr$^{-1}$ | Henson et al. (2011) |
| Export production | 8 % - 30 % | 2 % - 20 % | of total Prod | Boyd and Trull (2007) |
| Burial | 0.3 % - 0.8 % | 0.4 % | of total Prod | Sarmiento and Gruber (2006) |
| Shelf sea production | 12 % - 25 % | 20 % | of total Prod | Barrón and Duarte (2015);Wollast (1998) |
| Shelf sea export | 21 % - 27 % | 29 % | of total Export | Sarmiento and Gruber (2006) |
| Shelf sea burial | 100 % | 91 % | of total Burial | Sarmiento and Gruber (2006) |

Table 5: Summary of the model evaluation provided in section 3. Modern observations and estimates are compared to model results obtained for $z_{rem}^L$ in the range of measured values for a modern shelf sea (Cavan et al., 2017).

*EC 2) I'm a little surprised that you haven't tuned your productivity to more realistic values, and that you lay the blame for your model's low productivity on its implicit microbial loop and missing distinction of new / regenerated production; it should be possible to achieve more realistic levels of productivity through tuning, and other models (again, Yool & Tyrrell, 2002) have no such difficulties while having the same limitations*

Following the modifications to the model's equations in agreement with the suggestions of Referee nr.1 and Referee nr.2, we have further retuned the model towards more realistic values of production and export. This was done by slightly modifying a few biogeochemical parameters (coagulation rate, remineralization rate, Ca-P formation rate) and mixing in the shelf. Further changes to the model, following recent literature, consisted in distinguishing between a water column and a sediment half saturation constant for aerobic remineralization. The combination of these changes in the equations and in a few parameter values allowed us to increase production and export (bringing them closer to the lower end of modern estimates), while still maintaining the values of export production and burial to export ratio in the range of current estimates. We modified the discussion of the model evaluation (subsection 3.2) accordingly.

*EC 3&4) Although you mention the timescales over which you model can easily be run (as well as alluding to simulating it across the Phanerozoic), none of your figures show the time evolution of your model; it might be informative for readers were you to include some indication of the model's time-evolution from, say, the arbitrary initial conditions that you list in Table 2 (e.g. see Figure 5 of Yool & Tyrrell, 2002)*
*On a related point, how does your model compare in terms of residence times of the elements it represents?; given that it includes river inputs of P and reasonably represents ocean concentrations, I would expect it to do a good job here, but it might still be worth drawing attention to this*

To address this comment, we have included in the manuscript an additional figure (revised paper's **Figure 3**) to show an example of modelled tracer evolution. Furthermore, we have divided the Evaluation section in two parts with an initial subsection 3.1 focusing on the model dynamics and a second subsection 3.2 focusing on the comparison to model ocean budgets and fluxes (which corresponds to the old Evaluation before revisions).

Subsection 3.1 of the revised manuscript reads as follows:

**3.1 Timescales**

Starting from the initial values listed in Table 3, the modelled state variables evolve towards equilibrium for any couple of values of $z_{rem}^S$ and $z_{rem}^L$ in the explored interval. Simple mass conservation checks show no hidden source or sink of tracers in the model's boxes. Figure 3 illustrates an example of evolution of the variables for $z_{rem}^S$ and $z_{rem}^L$ in the middle of the interval of explored values for both particle types. In all the ocean boxes, P shows an initial oscillation that evolves on timescales of tens of thousands of years (Figure 3a,b), as expected by the typical timescale of evolution of the tracer (Lenton and Watson, 2000). This is followed by a slower drift which depends on the dynamics of the deep water oxygen content, as the release and burial of P in the sediments depends on the level of oxygenation of the deep ocean and especially of the deep shelf sea. P reaches complete equilibrium as soon as the deep ocean boxes become stably oxygenated. The timescales of evolution of O are slower and lay on the order of tens of millions of years (Lenton and Watson, 2000). Oxygen in the deep shelf overcomes hypoxia after the first few

millions of years and then slowly evolves towards equilibrium on the same timescale of O in the other ocean boxes. The dynamics of SedPorg is also strongly driven by level of oxygenation of the deep shelf sea.

Figure 3 of the revised manuscript is shown below:

[Figure]

**Figure 3: Evolution of the state variables from the initial conditions listed in Table 2 and remineralization lengths roughly in the middle of the interval of explored values: $z_{rem}^S = 20$ m, $z_{rem}^L = 250$ m. (a) Evolution of inorganic phosphorus P in the water column (left axis) and of organic phosphorus in the sediments SedP$_{org}$ (right axis); (b) zoom on the dynamics of P in the two hundred thousand years; (c) Evolution of oxygen in the water column (left axis) and atmosphere (right axis). In subplot (c) the two lines O$^{ss}$ and O$^{so}$ are overlapping: the two variables evolve closely due to the couplingof the surface ocean with the atmosphere via air-sea gas exchange.**

**Answer to Anonymous Referee nr.1**

We thank Anonymous Referee nr.1 for their thoughtful comments which allowed us to improve the model and the manuscript. We especially appreciated the Referee's corrections regarding the benthic model and describe below our related changes to the code. We also provide a point by point answer to all of their comments. The Referee's comments are highlighted in blue, actual changes to the manuscript are highlighted in red.

*R1C1)*
*The model for phosphorus (P) degradation in marine sediments considers aerobic respiration (Eq. 11) but seems to ignore anaerobic degradation. As a result the burial efficiency increases when oxygen is deleted in ambient bottom waters whereas the available observations show that P burial efficiency actually decreases under low-oxygen conditions (Slomp et al., 2002; Van Cappellen and Ingall, 1994; Wallmann, 2010). The authors should try to change their benthic model (i.e. include anaerobic degradation and enhanced P release under anoxia) or explain why they apparently ignore the strong evidence for enhanced benthic P release under low oxygen conditions.*

We agree with Referee nr.1 that our model was failing at correctly representing the dependence of P degradation and burial on $O_2$ concentration, and have therefore modified the benthic model accordingly.

Our new model representation of the sediments includes an enhancement of the rate of $P_{org}$ remineralization (+25%) and decrease of Ca-P burial (-50%) under anoxic conditions compared to oxic conditions, in line with Slomp & Van Cappellen 2007. The new equations are described in detail in subsections 2.2.4 and 2.2.5 of the revised paper, and can be read in full in the paper's appendix. The two new parameters determining changes in the rates of remineralization and Ca-P formation under anoxic conditions have been added to Table 4.

Despite these changes, our model results and conclusions regarding P and $O_2$ concentrations as a function of the particle remineralization lengths (Figure 4 and following ones) remain the same.

- New formulation of sediment $P_{org}$ remineralization for the sediment box i overlaid by the deep ocean box j:

$$SedRem\_ox^i = rm_r \cdot SedP_{org}^{\phantom{org}i} \cdot (O^j/(O^j + K_O^s))$$

$$SedRem\_an^i = (rm_r \cdot fe_{an}) \cdot SedP_{org}^{\phantom{org}i} \cdot (1 - O^j/(O^j + K_O^s))$$

where $fe_{an}$=1.25 is the remineralization rate enhancement factor in anoxic conditions.

- New formulation of CaP formation for the sediment box i overlaid by the deep ocean box j:

$$CaPform\_ox^i = CaP_r \cdot (SedP_{org}^{\phantom{org}i})^2 \cdot (O^j/(O^j + K_O^s))$$

$$CaPform\_an^i = (CaP_r \cdot fs_{an}) \cdot (SedP_{org}^{\phantom{org}i})^2 \cdot (1 - O^j/(O^j + K_O^s))$$

where $fs_{an}$=0.5 is the Ca-P formation rate suppression factor in anoxic conditions.

The revised subsection 2.2.5 on sediment remineralization reads as follows:

In each sediment box $i$, remineralization of $SedP_{org}$ happens in a similar way to remineralization in the water column, with an aerobic and an anaerobic component. The first takes up oxygen from the overlaying deep-water box $j$ and happens at a constant rate $rm_r$, while being limited by a Michaelis-Menten coefficient. Anaerobic remineralization releases its product to the atmosphere and happens at a faster rate $rm_r^* = rm_r \cdot fe_{an}$ with $fe_{an} > 1$, in agreement with recent observations and previous models (Slomp and Van Cappellen, 2006). Total sediment remineralization is therefore the sum of the two terms as in:

$$SedRem^i = rm_r \cdot SedP_{org}^{\ i} \cdot (O^j/(O^j + K_O^s)) + (rm_r \cdot fe_{an}) \cdot SedP_{org}^{\ i} \cdot (1 - O^j/(O^j + K_O^s)) \tag{11}$$

The revised subsection 2.2.4 on Ca-P formation reads as follows:

Ca-P formation happens at a lower rate under low oxygen conditions ($CaP_r^* = CaP_r \cdot fs_{an}$ with $fs_{an} < 1$), in agreement with observations and previous models (Slomp and Van Cappellen, 2006). The transition from aerobic and anaerobic conditions is controlled by a Michaelis-Menten type of function of the oxygen concentration in the deep ocean box j overlaying the sediment box i. The oxic and anoxic terms sum to the total formation term as in:

$$CaPform^i = \left(SedP_{org}^{\ i}\right)^2 \cdot \left[CaP_r \cdot O^j/(O^j + K_O^s) + (CaP_r \cdot fs_{an}) \cdot (1 - O^j/(O^j + K_O^s))\right] \tag{8}$$

*R1C2)*
*The shelf model ignores P burial in shallow-water shelf sediments even though observations in the modern ocean indicate that most burial of particulate organic matter (POM) occurs in the inner shelf region at < 50 m water depth (Dunne et al., 2007). The authors should try to change their benthic model to include shallow shelf burial or explain why they ignore burial in shallow shelf regions.*

We agree with Referee nr.1 that a significant fraction of the burial of POM happens in the shallowest regions of the shelf. Some of this is heavily influenced by the input of sediments and POM from the land. This shallowest portion of the shelf is also the most influenced by the details of the local bathymetry, tides, sediment resuspension and bottom layer mixing (Hill et al., 2008; Simpson and Pingree, 1978), which have important consequences in terms of physical-biogeochemical interactions. Differently from other models which distinguish between shallow (< 50 m deep) shelf and slope region (e.g., Slomp & Van Cappellen, 2007), BPOP calculates the sedimentation and burial fluxes dynamically, and would therefore require accounting for extra physical processes and flux parameterizations to justify going to such a higher level of detail. As our principal focus is on the first-order response to changes in the biological pump, we think that a 4-box representation of the ocean and of its circulation is a reasonable simplification, allowing us to keep the number of parameters small. We therefore keep this suggestion in mind for future developments of the model.

*R1C3)*
*Small (slowly sinking) particles are mostly degraded in the water column whereas a substantial fraction of the large (rapidly sinking) particles is not degraded but deposited at the seafloor. Consequently, large POM particles reaching the seabed are more reactive (fresher) than small (older) particles and the kinetic constant for benthic degradation should increase with increasing particle size (Stolpovsky et al., 2018). Since particle size (sinking speed, mineralization length) is the major parameter varied in the modeling, the authors should try to consider this effect in their benthic model.*

We thank Referee nr.1 for this interesting suggestion regarding the relation that may exist between the particles' remineralization rates in the sediments and their size or sinking speed. As BPOP allows to explore a variety of different hypothesis, we have decided to test the possibility of having sediment remineralization rates that depend on the particle properties, and we show our results below.

However, we highlight that the remineralization length $z_{rem}$ (around which the model is built) does not only depend on the particles' sinking speeds, but also on the particles' remineralization rates in the water column. By this definition, it may not be given that particles that have larger remineralization lengths (our "large particles") have also larger sinking speeds. They may instead just be more refractory to water column remineralization, which also allows the particles to be remineralized on average more at depth. Other factors, such as the fact that large particles are defined as a secondary product of the coagulation of smaller particles can also influence the liability the $LP_{org}$ pool. For these reasons we take this suggestion as a sensitivity experiment, but do not include it in the revised baseline model.

As a simple hypothesis, we have assumed that the sediment remineralization rate $rm_r$ increases linearly with $z_{rem}$ by 40% of our baseline value going from $z_{rem}$ = 0 to $z_{rem}$ = 450 m, with the baseline value of $rm_r$ being found at $z_{rem}$=225m, which is in the middle of the full range of explored values. In each run, the remineralization rate is calculated separately for small and large particles according to their respective $z_{rem}$. The two types of particles in the sediments are then solved separately and remineralized according to their respective rate. We have re-run the model under this assumption and include below a few significant comparison plots of the results of both our baseline run (the one adopted in the model) and the sensitivity run.

Our results show that there is no significant change in the model results that affects our primary conclusions from the baseline run. The main difference between the two sets of results consists in the facts that the sensitivity study shows a slightly stronger decoupling between the influence of the small and large particle remineralization lengths (respectively, $z_{rem}^S$ and $z_{rem}^L$) on the equilibrium budgets and fluxes. The value of $z_{rem}^S$ becomes even more influential for high $z_{rem}^L$, where the latter seems to make little difference in determining the equilibrium state of the model.

[Figure]

*Figure R1-1: Ocean P and O₂ budgets for 2 model configurations: constant remineralization rate (subplots (a) and (b), new baseline); remineralization rate dependent on the particles' remineralization length (subplots (c) and (d)).*

[Figure]

*Figure R1-2: Oxygen concentrations in the deep shelf (ds) and in the deep open ocean (do) for 2 model configurations: constant remineralization rate (subplots (a) and (b), new baseline); remineralization rate dependent on the particles' remineralization length (subplots (c) and (d)).*

[Figure]

Figure R1-3: Total ocean production for 2 model configurations: constant remineralization rate (subplots (a) and (b), new baseline); remineralization rate dependent on the particles' remineralization length (subplots (c) and (d)).

We have also added the following paragraph to the discussion section, in subsection 5.1.3:

Furthermore, we have tested the impact of having sediment remineralization rates that vary with the particles' $z_{rem}$, under the assumption that the liability of small and large particles may be different. In our experiment, we increased the remineralization $rm_r$ rate linearly with $z_{rem}$ by 40 % of our baseline value ($rm_r^0$), with $rm_r^0$ being found at the centre of the interval of explored values of $z_{rem}$ = [0 m, 450 m]. Under these conditions, we obtained a higher decoupling between the influence of $z_{rem}^S$ and $z_{rem}^L$ on budgets and fluxes, both being more strongly driven by the small particle properties for large values of $z_{rem}^L$.

*R1C4)*
*Considering these model limitations, I do not know whether the authors' conclusion: "shelf ocean anoxia can coexist with an oxygenated deep ocean" (abstract, line 19) is really valid. Moreover, this conclusion depends on the model assumption that deep water formation takes place in the open ocean. This assumption is questionable since much of the modern deep water formation happens at continental margins. If these margin sites are oxygen depleted the resulting deep water would also be oxygen depleted.*

We have changed the wording in the abstract to say that the results "suggest" this can happen (rather than "highlight" that it can) – the use of the word "can" should anyway have been taken to imply this is one of several possibilities. Regarding the impact of our representation of ocean circulation on the modelled oxygen distribution, in the modern ocean, deep water formation happens mostly at very high latitudes i.e., in the subpolar North Atlantic (Labrador Sea and Greenland Sea), in the Southern Ocean (Weddel Sea and Ross Sea). Both regions are characterized by physical and biogeochemical properties that differ substantially from our representation of a shelf sea, which is not intended to include polar and subpolar regions of sea-ice cover, continental ice shelves, deep convection etc. Therefore we don't think that attributing deep water formation to the shelf sea in our model would be the correct way to represent ocean circulation. Furthermore, our estimate of open ocean mixing and vertical exchange of water refers mostly to open ocean fluxes both at high and at low latitudes, and is in line with current estimates (Sarmiento and Gruber, 2006; Ganachaud and Wunsch, 2000). Instead it is our assumption of a circulation flux from the deep open ocean to the shelves, which thus boosts shelf nutrient concentration, which is more critical for determining the modelled oxygen distribution.

Bibliography (Answers to Anonymous Referee nr.1)

Hill, A. E., Brown, J., Fernand, L., Holt, J., Horsburgh, K. J., Proctor, R., Raine, R., and Turrell, W. R. (2008), Thermohaline circulation of shallow tidal seas, Geophys. Res. Lett., 35, L11605, doi:10.1029/2008GL033459.

Schlünz, B., and R. R. Schneider. "Transport of terrestrial organic carbon to the oceans by rivers: re-estimating flux-and burial rates." *International Journal of Earth Sciences* 88.4 (2000): 599-606.

Simpson J.H., Pingree R.D. (1978) Shallow Sea Fronts Produced by Tidal Stirring. In: Bowman M.J., Esaias W.E. (eds) Oceanic Fronts in Coastal Processes. Springer, Berlin, Heidelberg

Slomp, C. P. and Van Cappellen, P.: The global marine phosphorus cycle: sensitivity to oceanic circulation, Biogeosciences, 4, 155–171, https://doi.org/10.5194/bg-4-155-2007, 2007.

**Answer to Anonymous Referee nr.2**

We thank Anonymous Referee nr.2 for their thoughtful comments and corrections which allowed us to fix some important typos in the manuscript and encouraged us to rethink about the way in which the model is implemented. We provide below a point by point answer to all of their comments. The referee's comments are highlighted in blue, while changes to the manuscript are highlighted in red.

*R2C1)*
*It is stated on line 10 of page 4 that any organic matter that does not reach the sediments is instantaneously remineralised. This decision seems quite reasonable, but it is contradicted by the equations, in which organic matter is not only remineralized but is also advected and mixed. I recommend to do either one thing or the other but not both. If suspended particles are going to be mixed around in the model then they should have their own separate ordinary differential equations and state variables. Alternatively, if particle flux and remineralisation are made instantaneous, as stated in the manuscript, then there should be no mixing or advection of particulate organic matter. Whichever way it is done, the descriptions in the text need to be made consistent with the equations.*

We thank Referee nr.2 for their observation, and we agree with them on the fact that our choice of combining an implicit sinking scheme with lateral advection and mixing can be troublesome. For this reason, we have modified the code in order to switch off the lateral advective and vertical mixing fluxes of POC. In our new baseline run, POC is exclusively redistributed throughout the boxes by vertical sinking, in line with the Referee's suggestion. In order to account for these changes in the model, we have modified the manuscript as described below.

The description of the $P_{org}$ physical fluxes in section 2.2.2 has been reformulated as:

The implicit representation of the organic matter in the water column implies that no organic matter is accumulated in the ocean. In our baseline version of the model, corresponding to the results presented in this manuscript, SPorg and LPorg are redistributed throughout the watercolumn exclusively by implicitly modelled gravitational sinking before being either buried, accumulated in the sediments or remineralized. Even though the vertical export by downwelling and mixing (Stukel and Ducklow, 2017), and the lateral organic matter redistribution (Lovecchio et al., 2017;Inthorn et al., 2006) may be important when working with suspended SPorg (zremS = 0), these fluxes are not currently accounted for in the model.

We have modified Figure 1 and 2 in order to account for this change in the model.

We have run the revised model both with and without the lateral advective and vertical mixing fluxes in order to understand the impact of removing these fluxes onto our model results.
We include below a few significant plots which show overall higher oxygen levels and production in the absence of mixing/advective fluxes of $P_{org}$. Our main conclusions remain unchanged.

[Figure]

*Figure R2-1: Ocean P and O₂ budgets for 2 model configurations: without POC lateral advection and vertical mixing (subplots (a) and (b), new baseline); with POC lateral advection and vertical mixing (subplots (c) and (d)).*

[Figure]

*Figure R2-2: Oxygen concentrations in the deep shelf (ds) and in the deep open ocean (do) for 2 model configurations: without POC lateral advection and vertical mixing (subplots (a) and (b), new baseline); with POC lateral advection and vertical mixing (subplots (c) and (d)).*

[Figure]

*Figure R2-3: Total ocean production for 2 model configurations: without POC lateral advection and vertical mixing (subplots (a) and (b), new baseline); with POC lateral advection and vertical mixing (subplots (c) and (d)).*

*R2C2)*
*Tables 1 to 3 are very helpful. Another table needs to be added, listing the state variables in the model and stating their units. In addition, the units of all equations (the left-hand side) should also be stated, if they are not already given in the tables.*

As suggested by Referee nr.2, we have added a table (**Table 1** of the revised manuscript) which indicates names and units of the state variables of the model.

| Name | Description | Units |
|---|---|---|
| $P^{ss}$ | Inorganic phosphorus in surface shelf sea box | mmol m$^{-3}$ |
| $P^{ds}$ | Inorganic phosphorus in deep shelf sea box | mmol m$^{-3}$ |
| $P^{so}$ | Inorganic phosphorus in surface open ocean box | mmol m$^{-3}$ |
| $P^{do}$ | Inorganic phosphorus in deep open ocean box | mmol m$^{-3}$ |
| $O^{ss}$ | Molecular oxygen in surface shelf box | mmol m$^{-3}$ |
| $O^{ds}$ | Molecular oxygen in deep shelf box | mmol m$^{-3}$ |
| $O^{so}$ | Molecular oxygen in surface open ocean box | mmol m$^{-3}$ |
| $O^{do}$ | Molecular oxygen in deep open ocean box | mmol m$^{-3}$ |
| $O^{at}$ | Oxygen mixing ratio in atmosphere (mol mol$^{-1}$) | - |
| $SedP_{org}^{s}$ | Organic phosphorus in the sediments of the shelf sea | mmol m$^{-2}$ |
| $SedP_{org}^{o}$ | Organic phosphorus in the sediments of the open ocean | mmol m$^{-2}$ |
| $P^{TOT}$ | Diagnostic variable: total P budget from sources and sinks only | Tmol P |
| $O^{TOT}$ | Diagnostic variable: total O budget from sources and sinks only | Pmol O$_2$ |

**Table 1: List of the model's state variables and of their units**

We mention the new table in subsection 2.1 as follows:

The entire set of the model's state and diagnostic variables and their units are listed in Table 1.

*R2C3)*
*Too many of the equations in the model are dimensionally inconsistent. That is to say, the units on the left-hand side of the equation do not match the units on the right-hand side of the equation when the different terms are combined together. As an example, equation 4 on page 4 is an equation for the rate of organic matter production in units of C2 GMDD Interactive comment Printer-friendly version Discussion paper phosphorus. This is a flux (rate of transfer), and therefore has to be in units of Moles y-1 or mmol m-3 y-1 or similar. Because this is an ongoing flux rather than a one-off transfer, it must be expressed as a rate of transfer per unit time. However, none of the terms on the right-hand side of the equation have time anywhere in their units. The equation is formulated in such a way that it appears to be aimed at converting a fraction of the surface phosphorus concentration into production at each timestep, but the way it is actually formulated means that the rate of*

*conversion of surface phosphate into production (organic matter) will depend on the timestep used. Shorter time steps will convert phosphate to organic matter more rapidly than longer timesteps whereas ODE equations should be timestep-independent. Equation 5 is another example, where, according to the equation, 'Coag' must have units of organic matter concentration squared per year, which makes no sense. Before resubmission, I recommend that every equation in the model is checked for dimensional (units) consistency: multiplying through the units of the terms on the right-hand side should produce the units of the term on the left-hand side.*

We thank Referee nr.2 for having spotted a few mistakes in the manuscript's equations. We have double checked the entire set of equations, the parameters' units and we have corrected a few typos.

As highlighted by the Referee, the uptake rate $P_{eff}$ (Equation 4) as well as the coagulation factor $cg_r$ (Equation 5) were missing part of their units (see Table 4 of the revised manuscript). We have now corrected their units accordingly (see also Gruber et al., 2006). We have explained better the values and magnitudes of the parameters in the table's caption. We have also spotted another few inconsistencies in the units and fixed them both in the tables and in the equations.

*R2C4)*

*I did not notice a statement anywhere that conservation of mass (or, more properly, conservation of total inventories of elements) has been checked and found to be stable. This is easy to do for a box model, whether it is closed or open. Obviously for a closed model, if there are no errors in the equations, the total sum of atoms of a given element should be constant over time. For an open model, the changes in the total inventory over time should exactly match the sum of external inputs over time minus the sum of the outputs from the system over time (i.e. ΔInv= - ). This can easily be checked by adding two extra differential equations to the model: one to track the sum of the external inputs to the system and another to keep track of the sum of the losses from the system as a whole. From the model equations, I suspect that the model does not properly conserve phosphorus and oxygen but rather there is some (unintended) cumulative creation/destruction over time. This will interfere with the ability of the model to be run over long timescales to address the geological timescale questions of interest. It is a little bit unclear, but it appears that the amount of phosphate removed per unit time from the surface box as particle export is not identical to the amounts of phosphate added per unit time to the deep box and sediments combined. The euphotic zone depth appears in the equation for the former but not the latter, for instance, whereas if it appears in one then it should also appear in the other. Again, checks can be made by adding extra (book-keeping) ODEs to the model. For this example, one extra ODE could tally up the cumulative export from the surface box and another extra ODE could tally up the sum of the cumulative inputs to the deep and sediment boxes. At the end of each model run, a quick numerical check can be made to ensure that the tallies are identical within the precision of numerical rounding errors.*

We agree with Referee nr.2 that a check on conservation of mass is essential for the model's evaluation. Even though we did not discuss it explicitly, our model does go to equilibrium on the long term, as now visible from **Figure 3** of the revised manuscript.

As a further check we have now added 2 extra variables to the model, $P^{TOT}$ and $O^{TOT}$, which represent the total P and O budgets. These two variables (in units of TmolP and $PmolO_2$, respectively) are initialized to the total sum of the two tracers across all boxes at time zero.

At each time step, $P^{TOT}$ and $O^{TOT}$ evolve according to the sum of all sources and sinks in the model. Their derivatives are as follows:

$$\frac{dP^{TOT}}{dt} = \left[ P_{in} - \left( CaPform^s \cdot A_{ocean} \cdot \mathcal{P}_{shelf} + CaPform^o \cdot A_{ocean} \cdot \left(1 - \mathcal{P}_{shelf}\right) \right) \right] \cdot 10^{-15}$$

$$\frac{dO^{TOT}}{dt} = \left[ OProd^{ss} \cdot V^{ss} + OProd^{so} \cdot V^{so} \right.$$
$$- \left( \left( SedRemO^{ds} + AerRem^{ds} \right) \cdot V^{ds} + \left( SedRemO^{do} + AerRem^{do} \right) \cdot V^{do} \right.$$
$$\left. \left. + OxyWeath \cdot \left( Mol_{atmo} \cdot 10^3 \right) \right) \right] \cdot 10^{-18}$$

At the end of each model run we plot the sum of the total P and O content in the model at each time step by multiplying the concentrations in each box by the box volume or area (the latter for the sediments), and check that this corresponds to the time evolution of $P^{TOT}$ and $O^{TOT}$ calculated at run time regardless of the partition of the tracers among the boxes.

In order to include these new diagnostic variables in the model's description, we have modified the first paragraph of subsection 2.1 as follows:

The box model resolves explicitly for each relevant box the local concentrations of three types of tracers: molecular oxygen $O_2$ (O), inorganic dissolved phosphorus (P) and sediment organic phosphorus ($SedP_{org}$). The total budgets of P and O, respectively $P^{TOT}$ and $O^{TOT}$, are also independently integrated from the net sources and sinks of the two tracers over the entire model domain, for the purpose of checking mass conservation.

We have also added the following statement to subsection 3.1 of the model evaluation:

Starting from the initial values listed in Table 3, the modelled state variables evolve towards equilibrium for any couple of values of $z_{rem}^S$ and $z_{rem}^L$ in the explored interval. Simple mass conservation checks show no hidden source or sink of tracers in the model's boxes.

We include below the result for the choice of $z_{rem}^S$ and $z_{rem}^L$ used for Figure 3 of the revised paper.

[Figure]

*Figure R2-4: Comparison between the evolution of the total tracer budgets ($P^{TOT}$ and $O^{TOT}$) calculated at run time from the tracer's net input fluxes (total sources + total sinks), and the tracer budgets calculated integrating the local concentrations in the boxes at each time step from the model's output.*

[revised manuscript text omitted]
~~However, the model allows to include lateral advective and vertical mixing fluxes of organic phosphorus by setting to one the value of the parameter , which acts as a switch in the equations. This may be necessary when working with suspended $SP_{org}$ pools ($z_{rem}^S = 0$), and therefore that modelled $SP_{org}$ and $LP_{org}$ can only be transported by physical fluxes from regions of production to regions of remineralization. For this reason, physical fluxes affect the two organic matter species only in a single direction (Figure 1b). Despite this limitation, we believe thattheseon the water column $P_{org}$ is essential to account for both theorganic matteradvectivefrom the coast to the open waters both at the surface and at depth.When advected and mixed, implicitly modelled $SP_{org}$ and $LP_{org}$ can only be transported by physical fluxes from regions of production to regions of remineralization. For this reason, if included, these fluxes affect the two organic matter species only in a single direction (Figure 1b). Due to the wide extension of the modelled ocean boxes, lateral fluxes are assumed to only affect $SP_{org}$. The lateral export of $SP_{org}$ reduces its availability for export and burial in the shelf sea, i.e., at each time step lateral fluxes out of the shelf happen before the $SP_{org}$ vertical export and sedimentation fluxes are calculated.~~

**2.2.3 Remineralization length scheme**

The export and sedimentation fluxes of $P_{org}$ through the water column are represented by a remineralization length scheme. In this representation, the vertical fluxes of organic matter f(z) vary exponentially with depth. The shape of the exponential depends on the value of the remineralization length ($z_{rem}$) of each organic matter species:

$$f^k(z) = f_0^k \cdot e^{-\frac{z-z_0}{z_{rem}^k}} , \tag{6}$$

where $f_0^k$ is the flux at the reference depth $z_0$, and the index $k$ indicates the organic matter pool of reference, either small (S) or large (L). This representation of the export flux is convenient, as it does not depend on the specific choice of $z_0$ (Boyd and Trull, 2007).

The remineralization length $z_{rem}$ indicates the distance through which the particle flux becomes 1/e times (about 36 %) the flux at the reference depth (Buesseler and Boyd, 2009;Marsay et al., 2015). This quantity is expressed in metres and can be calculated as the ratio between the particle sinking speed and the particle's remineralization rate (Cavan et al., 2017). Consequently, $z_{rem}$ implicitly contains information on several particle inherent properties (among which density, size, shape, organic matter liability) as well as information about the surrounding environment, e.g., the type of heterotrophs which feed upon the organic material (McDonnell and Buesseler, 2010;Baker et al., 2017). For simplicity, we assume that the remineralization length of small and large particles does not vary between shelf sea and open ocean boxes. We examine the potential impact of this limitation in the discussion section of the paper.

**2.2.4 Sediments and burial**

SPorg and LPorg accumulate in the sediments as SedP$_{org}$, which is calculated as a density per unit of area. The sediment flux into the sediment box $i$ depends on the organic matter concentration in the overlaying deep ocean box $j$ and on the remineralization length of the two pools as in:

$$SedFlx^i = \left(SP_{org}{}^j \cdot exp\left(-\Delta Z_j / z_{rem}^S\right) + LP_{org}{}^j \cdot exp\left(-\Delta Z_j / z_{rem}^L\right)\right) \cdot \Delta Z_j \tag{7}$$

The accumulated SedP$_{org}$ is partially slowly remineralized and partially irreversibly buried in a mineral form. Phosphorus burial as mineral Ca-P- is modelled as a function of the square of SedP$_{org}$ that accumulates in the sediments, in a way that is analogous to the dynamics of particle coagulation, and is regulated by a constant rate coefficient CaP$_r$. Ca-P formation happens at a lower rate under low oxygen conditions (CaP$_r^*$ = CaP$_r$ · fs$_{an}$ with fs$_{an}$ < 1), in agreement with observations and previous models (Slomp and Van Cappellen, 2006). The transition from aerobic and anaerobic conditions is controlled by a Michaelis-Menten type of function of the oxygen concentration in the deep ocean box j overlaying the sediment box i. The oxic and anoxic terms sum to the total formation term as in:

$$CaPform^i = \left(SedP_{org}{}^i\right)^2 \cdot \left[CaP_r \cdot O^j/(O^j + K_O^s) + (CaP_r \cdot fs_{an}) \cdot (1 - O^j/(O^j + K_O^s))\right] \quad \cancel{CaP_r \cdot (SedP_{org}{}^i)^2} \tag{8}$$

This flux is essential to balance the continuous P river input, therefore preventing the ocean from overflowing with nutrients.

**2.2.5 Remineralization in the water column and sediments**

At each time step, remineralization in the water column completely depletes the P$_{org}$ that has not reached the sediments. In the two surface boxes, remineralization of P$_{org}$ that is not exported below the euphotic layer uses up part of the oxygen that was released by production. For this reason, net oxygen production in each surface box is proportional to the export of P$_{org}$ below the euphotic layer. Export from a surface box $i$ to a deep box $j$ happens both via gravitational sinking and via mixing, as in:

$$VExp^i = SP_{org}{}^i \cdot exp\left(-(\Delta Z_{eu}/2) / z_{rem}^S\right) + LP_{org}{}^i \cdot exp\left(-(\Delta Z_{eu}/2) / z_{rem}^L\right) \tag{9}$$

$$\cancel{VExp^i = SP_{org}{}^i \cdot exp\left(-(\Delta Z_{eu}/2) / z_{rem}^S\right) + LP_{org}{}^i \cdot exp\left(-(\Delta Z_{eu}/2) / z_{rem}^L\right) +}$$
$$\cancel{(SP_{org}{}^i + LP_{org}{}^i) \cdot Mix_{ij}/V^i} \tag{9}$$

At depth, the remineralization of P$_{org}$ that does not reach the sediments happens through both aerobic and anaerobic processes, completely depleting the remaining P$_{org}$. Water-column remineralization of P$_{org}$ into inorganic P in the deep box $j$ is therefore calculated as:

$$WcRem^j = SP_{org}{}^j \cdot \left(1 - exp\left(-\Delta Z_j / z_{rem}^S\right)\right) + LP_{org}{}^i \cdot \left(1 - exp\left(-\Delta Z_j / z_{rem}^L\right)\right) \
[revised manuscript text omitted]

$$\cancel{LatExp^{ss} = SP_{org}{}^{ss} \cdot (Upw + Mix_{ls})/V^{ss}} \tag{A7}$$

$$VExp\_SP_{org}{}^{ss} = (SP_{org}{}^{ss} \cancel{- LatExp^{ss}}) \cdot (exp(-(\Delta Z_{eu}/2) /z_{rem}^S) \cancel{+ Mix_{vs}/V^{ss}})$$
$$\tag{A7\underline{8}}$$

$$VExp\_LP_{org}{}^{ss} = LP_{org}{}^{ss} \cdot (exp(-(\Delta Z_{eu}/2) /z_{rem}^L) \cancel{+ Mix_{vs}/V^{ss}})$$
$$\tag{A8\underline{9}}$$

$$\frac{dP^{ss}}{dt} = P_{in} \cdot (1 - \mathcal{P}_{open})/V^{ss} + (Upw \cdot (P^{ds} - P^{ss}) + Mix_{ls} \cdot (P^{so} - P^{ss}) + Mix_{vs} \cdot (P^{ds} - P^{ss})) \cdot spy/V^{ss}$$
$$- \left(VExp_{SP_{org}}{}^{ss} + VExp_{LP_{org}}{}^{ss}\right)$$

$$\tag{A9\underline{10}}$$

$$AirSea^{ss} = K_W \cdot (O^{at}/K_{Henri} - O^{ss}) \cdot (A_{ocean} \cdot \mathcal{P}_{shelf})/V^{ss} \qquad (A10)$$

$$OProd^{ss} = OP_{Red} \cdot \left(VExp_{SP_{org}}{}^{ss} + VExp_{LP_{org}}{}^{ss}\right) \qquad (A11)$$

$$\frac{dO^{ss}}{dt} = \left(Upw \cdot (O^{ds} - O^{ss}) + \mathrm{Mix}_{ls} \cdot (O^{so} - O^{ss}) + \mathrm{Mix}_{vs} \cdot (O^{ds} - O^{ss})\right) \cdot spy/V^{ss} + AirSea^{ss}$$
$$+ OProd^{ss}$$

$$\rule{8cm}{0.4pt} \qquad (A12)$$

**A.3 Deep shelf sea (ds)**

$$V^{ds} = \Delta Z_{ds} \cdot A_{ocean} \cdot \mathcal{P}_{shelf}$$
$$(A13)$$

$$VInp\_SP_{org}{}^{ds} = VExp\_SP_{org}{}^{ss} \cdot (V^{ss}/V^{ds})$$
$$(A14)$$

$$SP_{org}{}^{ds} = VInp\_SP_{org}{}^{ds} - cg_r \cdot (VInp\_SP_{org}{}^{ds})^2$$
$$(A15)$$

$$LP_{org}{}^{ds} = VExp\_LP_{org}{}^{ss} \cdot (V^{ss}/V^{ds}) + cg_r \cdot (VInp\_SP_{org}{}^{ds})^2$$
20
$$(A16)$$

$$\cancel{LatExp^{ds} = SP_{org}{}^{ds} \cdot Mix_{td}/V^{ds}} \qquad \cancel{(A18)}$$

$$Rem\_SP_{org}{}^{ds} = (SP_{org}{}^{ds} - \cancel{LatExp^{ds}}) \cdot (1 - exp(-\Delta Z_{ds}/z_{rem}^S))$$
$$(A17)$$

$$Rem\_LP_{org}{}^{ds} = LP_{org}{}^{ds} \cdot (1 - exp(-\Delta Z_{ds}/z_{rem}^L)) \qquad (A18)$$

$$AerRem\_SedP_{org}{}^{ds} = rm_r \cdot SedP_{org}{}^s/\Delta Z_{ds} \cdot (O^{ds}/(O^{ds} + K_O^s \cancel{K_O}))$$
$$(A19)$$

$$AnaRem\_SedP_{org}{}^{ds} = (rm_r \cdot fe_{an}) \cdot SedP_{org}{}^s/\Delta Z_{ds} \cdot (1 - O^{ds}/(O^{ds} + K_O^s)) \qquad (A20)$$

$$\frac{dP^{ds}}{dt} = (Upw \cdot (P^{do} - P^{ds}) + Mix_{ld} \cdot (P^{do} - P^{ds}) + Mix_{vs} \cdot (P^{ss} - P^{ds})) \cdot spy/V^{ds}$$
$$++ \left(Rem\_SP_{org}{}^{ds} + Rem\_LP_{org}{}^{ds} + AerRem\_SedP_{org}{}^{ds} + AnaRem\_SedP_{org}{}^{ds}\right)$$

$$+ \left(Rem\_SP_{org}{}^{ds} + Rem\_LP_{org}{}^{ds} + Rem\_SedP_{org}{}^{ds}\right)$$

(A21)

$$AerRem\underline{Wc}O^{ds} = OP_{Red} \cdot \left(Rem\_SP_{org}{}^{ds} + Rem\_LP_{org}{}^{ds}\right) \cdot \left(O^{ds}/\left(O^{ds} + K_O^w\,\underline{K_O}\right)\right)$$

(A22)

$$Aer\underline{Sed}RemSedO^{ds} = OP_{Red} \cdot AerRem\_SedP_{org}{}^{ds}$$ (A23)

$$\frac{dO^{ds}}{dt} = \left(Upw \cdot (O^{do} - O^{ds}) + Mix_{ld} \cdot (O^{do} - O^{ds}) + Mix_{vs} \cdot (O^{ss} - O^{ds})\right) \cdot spy/V^{ds} - AerRem\underline{Wc}O^{ds}$$
$$- Aer\underline{Sed}RemSedO^{ds}$$

(A24)

**A.4 Surface open ocean (so)**

$$V^{so} = \Delta Z_{eu} \cdot A_{ocean} \cdot (1 - \mathcal{P}_{shelf})$$

(A25)

$$Prod^{so} = P_{eff} \cdot (P^{so}/(P^{so} + K_P)) \cdot P^{so}$$

(A26)

$$SP_{org}{}^{so} = \left(Prod^{so} \right) - cg_r \cdot \left(\left(Prod^{so}\right) \right)^2$$

(A27)

$$LP_{org}{}^{so} = cg_r \cdot \left(Prod^{so} \right)^2$$

[revised manuscript text omitted]

---

## Author Response (AR4)

Author's response

**"BPOP-v1 model: exploring the impact of changes in the biological pump on the shelf sea and ocean nutrient and redox state"**

Elisa Lovecchio[1] and Timothy M. Lenton[1]
[1]Global Systems Institute, University of Exeter, Exeter, EX4 4QE, United Kingdom
* * *
Dear Editor,

We are very glad to read that our manuscript has been accepted for publication on GMD.
We thank you very much for your support. We include all the necessary files in the final upload.

As the re-uploaded manuscript and supplement correspond to the files of the last upload of February 12[th], we are not including any further track changes.

Best regards,
Elisa Lovecchio